



# Last interglacial (MIS 5e) sea level proxies in the glaciated Northern Hemisphere

**April S. Dalton[1], Evan J. Gowan[2,3,4], Jan Mangerud[5], Per Möller[6], Juha P. Lunkka[7], Valery Astakhov[8,9]**

[1]Department of Physical Geography and Geoecology, Charles University, Prague, Czech Republic

[2]Department of Earth and Environmental Sciences, Kumamoto University, Kumamoto, Japan

[3]Alfred Wegener Institute, Helmholtz Center for Polar and Marine Research, Bremerhaven, Germany

[4]MARUM, University of Bremen, Bremen, Germany

[5]Department of Earth Science, University of Bergen and Bjerknes Centre for Climate Research, Bergen, Norway

[6]Department of Geology, Quaternary Sciences, Lund University, Lund, Sweden

[7]Geology Research Group, Oulu Mining School, University of Oulu, P.O. Box 9000, Oulu FI-90014 Finland

[8]Institute of Earth Sciences, St. Petersburg University, Universitetskaya 7/9, 199034, St. Petersburg, Russia

[9]A. P. Karpinsky Russian Geological Research Institute (VSEGEI), Sredny pr. 74, 199178, St. Petersburg, Russia

*Correspondence to*: April S. Dalton (aprils.dalton@gmail.com) and Evan J. Gowan (evangowan@gmail.com)

**Abstract.** Because global sea level during the last interglacial (LIG; 130-115 ka) was higher than today, the LIG is a useful analogue for improving predictions of future sea level rise. Here, we synthesize sea level proxies for the LIG in the glaciated Northern Hemisphere for inclusion in the World Atlas of Last Interglacial Shorelines (WALIS) database. We describe 82 sites from Russia, northern Europe, Greenland and North America from a variety of settings, including boreholes, riverbank
exposures and along coastal cliffs. Marine sediments at these sites were constrained to the LIG using a variety of radiometric methods (radiocarbon, U-Series dating, K-Ar dating), non-radiometric methods (amino acid dating, luminescence methods, and electron spin resonance, tephrochronology) as well as various stratigraphic and palaeoenvironmental approaches. As the areas in this database were covered by ice sheets from the penultimate glaciation and were affected by glacial isostatic adjustment (GIA), most of the proxies show that sea level was much higher than present during the LIG. Many of the sites
show evidence of regression due sea level fall due to GIA uplift, and some also show fluctuations that may reflect regrowth of continental ice or increased influence of the global sea level signal. The database is available at https://doi.org/10.5281/zenodo.5602212 (Dalton et al., 2021).

## 1 INTRODUCTION

During the last interglacial (LIG) between 130 and 115 ka (peak interglacial at 123 ka; Lisiecki and Raymo, 2005),
temperatures were warmer than today by up to 5°C in some regions of the northern Hemisphere (Dahl-Jensen et al., 2013),



and global sea levels were 5 to 10 m higher (Dutton and Lambeck, 2012). Like today, Greenland and Antarctica were the predominant global ice stores, as large continental ice sheets that grew repeatedly during the Quaternary over North America and Eurasia were absent at that time (see Batchelor et al., 2019). The LIG therefore represents a useful analogue for understanding the behavior of large continental ice sheets in a warming world, which is key for improving prediction of

future melting of the Greenland and Antarctica ice sheets and concomitant sea level rise (Slater et al., 2021).

The World Atlas of Last Interglacial Shorelines (WALIS) is a standardized database that has been created to archive global sea level sites constrained to the LIG. Here, we contribute 82 sites from the glaciated Northern Hemisphere to WALIS. We focus on sites that were covered by ice during the MIS 6 glaciation (191 ka to 130 ka; Lisiecki and Raymo, 2005; Fig. 1),

including Russia, Finland, Estonia, Poland, Sweden, Norway, Svalbard, Iceland, Greenland, Canada, and the United States. Sea level proxies in the glaciated regions of the southern North Sea, Jutland Peninsula and Great Britain are the subject of separate studies in this issue (Cohen et al., *in preparation*). To standardize the presentation of LIG proxy sites, we use marine isotope stages (MIS), as presented in Lisiecki and Raymo (2005) and consider that LIG corresponds with MIS 5e (130 to 115 ka) and the penultimate glaciation to be MIS 6. Most of the sediments described herein were deposited in

isostatically depressed land immediately following the retreat of major continental ice sheets from the MIS 6 glaciation (Fig 2). Isostatic recovery is sometimes preserved as a sequence of deep water, shallowing and shoreline, deltaic, estuarine depositional settings. Such LIG sites were subject to considerable erosion from subsequent glaciations (notably, during MIS 4 and MIS 2; Batchelor et al., 2019) and are therefore only sporadically preserved. There are also many more sites preserved in Europe than in North America, likely a consequence of the smaller extent of the MIS 4 and MIS 2 glaciations relative to

MIS 6. In general, areas near the center of large ice sheets underwent more isostatic depression than more peripheral sites. Our database of LIG sites in the Northern Hemisphere is open access and available at https://doi.org/10.5281/zenodo.5602212 (Dalton et al., 2021). A detailed description of database fields in the WALIS database is available at https://doi.org/10.5281/zenodo.3961544 (Rovere et al., 2020).

The first part of this manuscript (Sections 2-5) we define the types of sea level proxies, elevations measurements, dating techniques and quality assessment, all of which are technical aspects of entering the LIG data into the WALIS database. Then, in Section 6, we describe each LIG site in detail, paying particular attention to the elevation of marine sediments and any geochronological constraints. In Section 7, we present sites containing LIG marine sediments that are not *in situ* and have been transported and/or glaciotectonised post-deposition. These sites are unsuitable as precise indicators of RSL and

are therefore excluded from the WALIS database. However, they contribute to the general picture of uplift and are therefore included here. In the Discussion (Section 8) we provide an overview of the LIG sites compiled for this manuscript, as well as provide examples of other marine deposits in the glaciated region, notably MIS 7 (191 ka to 130 ka), MIS 5c (peak 96 ka), MIS 5a (peak 82 ka) and MIS 3 (57 ka to 29 ka) and Holocene (11.5 ka to present-day). We conclude with suggestions for

future research. The LIG is known regionally as the Kazantsevo interglacial (Siberia; here however redefined as Karginsky,
per Astakhov 2013), the Mikulino interglacial (Russia), the Eemian (western, central and northern Europe), the Ipswichian
(United Kingdom), the Langelandselv interglaciation (Greenland) and the Sangamonian (North America).

## 2    SEA LEVEL PROXIES

Our approach to describing sea level proxies differs from the standard approach used in most of the studies in WALIS. Since
most of the LIG sites are located in places that were undergoing rapid sea level changes dominantly due to GIA rather than
global sea level change, it is essentially impossible to pinpoint when sea level was at a particular elevation (especially given
the large uncertainty in the dating methods). In many locations, there are indications of regression from an often
indeterminate highstand position, to a position below the elevation of the outcrop. For many sites, there is clear evidence of
coarse grained, wave influenced deposits that show that sea level was near the elevation of the the deposit. The indicative
meaning of these deposits, as defined by Rovere et al. (2016), is not sufficiently clear to deduce a precise sea level position.


We regard it as more useful to describe the sea level proxies in terms of how much information a site provides in terms of
changes in sea level during the LIG. The primary target for this compilation is for those interested in infering the size of the
MIS 6 ice sheets through GIA modelling (e.g. Lambeck et al., 2006). In the database descriptions, we have indicated the
relative water depth based on the geological descriptions at the sites (i.e. deep water, shallow water, near sea level, above sea
level). From this information, it should be possible to test the reliability of MIS 6 ice sheet reconstructions and deduce LIG
sea level.

In our database, the vast majority of the sea level proxies are denoted as "marine limiting". The entered elevation marks the
highest elevation of marine sediments at a site. This usually marks an unconformity between the marine sediments and
overlying sediments that date to between MIS 5d and MIS 1. In a few cases, the contact is a conformable transition from
marine to terrestrial (often fluvial or lacustrine) sedimentation. At sites where there is reasonably high confidence of the
indicative meaning (generally where there is confidence of the sea level highstand), we have defined them as sea level
indicators using the standard approach of WALIS (i.e. Rovere et al., 2016), and note that sea level likely never exceeded that
elevation. At sites where there is evidence of a regression and transgression within the LIG, we have created two entries in
the database. Based on the amount of information on sea level position and variations at a site, we have assigned a quality
score, which will be elaborated on in Section 5.



## 3  ELEVATION MEASUREMENTS

A summary of elevation measurement techniques and datums are found on Tables 1 and 2. The techniques used to measure the elevation are not stated in many of the studies covered in our database and were extracted from stated elevations and

section diagrams in the papers. In these cases, we applied a nominal uncertainty of 20% of the stated elevation, as recommended by Rovere et al. (2016), or 10 m, which ever was smaller. The later constraint is it is unlikely the elevation uncertainty will be worse than the contour intervals of typical topographic maps (10-20 m), provided the authors were precise in the pinpointing the location of their site. For studies that involved the authors of this paper, we were able to provide the details of the elevation measurements and provide narrower uncertainties. The datum used was not stated in most

of the studies presented here and are assumed to be referenced to present day mean sea level. The tidal range in most locations covered in our database is presently relatively small (*i.e.* < 1-2 m), so this is unlikely to add significant uncertainty.

## 4  OVERVIEW OF DATING TECHNIQUES

A large number of dating techniques have been applied to LIG marine deposits covered in our database. These dating techniques include absolute (luminescence, electron spin resonance, U/Th), minimum limiting (radiocarbon), relative (amino

acid racemization, stratigraphy, environmental conditions). The absolute dating techniques have relatively large uncertainties and cannot be used to give a precise timing of deposition within the LIG. When combined with paleo-environmental conditions, it can usually be concluded that the deposit has a LIG age, rather than being part of another period of high sea level (*e.g.* Holocene, MIS 3, MIS 5a/c, MIS 7). Consideration of quality of the age control in the database is elaborated in Section 5.

### 4.1.1  Amino acid racemization dating

Amino acid geochronology measures the racemisation of amino acids. For the LIG, the epimerization of allo-isoleucine to isoleucine is most used (Oldale et al., 1982; Miller and Mangerud, 1985). Older shells have a higher isoleucine epimerization ratio than younger shells. However, this is a relative dating technique, with the epimerization controlled by regional diagenetic temperature, among other factors (Andrews et al., 1983). Therefore, this technique can only be used for

correlation between sites or to differentiate between different marine incursions in each region, rather than give precise ages.

### 4.1.2  Radiocarbon dating

Radiocarbon dating measures the amount of radioactive carbon ($^{14}C$) remaining in organic material after death of the dated animal/plant. The time since death can be approximated by consideration of the mean half-life (5730 years) of $^{14}C$ (Stuiver and Polach, 1977) and then converted to calendar year via calibration (Reimer et al., 2020). However, this chronological

method is only useful for samples less than ~45,000 years old because the remaining $^{14}C$ in old samples is too scarce to be





reliably measured beyond that point, and the sample becomes increasingly susceptible to modern-day carbon contamination (Douka et al., 2010). Thus, for the purposes of identifying MIS 5e marine sites, radiocarbon ages offer only minimum constraint.

### 4.1.3 Stratigraphic inferences

In many cases, the stratigraphic position of a particular marine unit has provided evidence for its age of deposition. When the marine unit is overlain by tills that are independently assigned to MIS 5d/b, MIS 4 or MIS 2 glaciation, this is used as evidence to support an MIS 5e age assignment (e.g., at Põhja-Uhtju and Peski; Miettinen et al., 2002). Conversely, the presence of a till directly underneath the marine sediments that suggests significant isostatic depression (often related to the MIS 6 glaciation) and rapid marine inundation into the isostatically depressed landscape and shallowing of marine waters

owing to subsequent rebound (e.g., Ile aux Coudres; Occhietti et al., 1995).

### 4.1.4 Palaeoenvironmental inferences

Climate during MIS 5e was several degrees warmer than present-day temperatures in the Northern Hemisphere (Rasmussen et al., 2003; Sánchez Goñi et al., 2012). As a result, palaeo-indicators of warmer-than-present-day conditions are often used as support for a MIS 5e age assignment. Marine-based palaeoecological indicators commonly preserved in the stratigraphic

record include dinoflagellate cysts, foraminifera, Coelenterata, Bryozoa and Mollusca, diatoms, and marine gastropods (Bergsten et al., 1998; Mangerud et al., 1981). In the terrestrial sediments that often overlie the MIS 5e marine unit, pollen and Coleoptera are some of the most used markers for determining palaeo-temperature (Dredge et al., 1992; Miettinen et al., 2002). Some caution is needed when considering these as correlating to MIS 5e in the absence of other numerical dating methods, as it is possible that these deposits could be from an older interglacial period. In our database, we refer to any site

with environmental conditions supporting LIG assignment as an "Eemian interglacial deposit", as defined by Mangerud et al. (1979).

### 4.1.5 Thermoluminescence

A common dating method for MIS 5e marine sediments is thermoluminescence (TL), although in recent years it is largely replaced by optically stimulated luminescence (OSL; described below). This technique measures the last exposure of a

sediment to sunlight via the resetting of electrons (Huntley et al., 1985; Lamothe and Huntley, 1988). In a laboratory setting, these changes are measured by heat stimulation. Either quartz or feldspar can be used, although feldspar is susceptible to anomalous fading, which can lead to large uncertainties in the age estimation (Godfrey-Smith et al., 1988; Huntley et al., 1985).





### 4.1.6    Infrared stimulated luminescence (IRSL)

This method uses wavelengths in the infrared range to induce luminescence in feldspar, which in some cases has been shown to reduce uncertainties in age estimates compared with TL (Godfrey-Smith et al., 1988).

### 4.1.7    Optically stimulated luminescence (OSL)

Similar to TL dating, OSL measures the refilling of shallow electron traps in sediment grains that occur during burial following exposure of the sediment grain to sunlight; therefore, the time since burial can be obtained (Duller, 2008; Huntley
et al., 1985). In a laboratory setting, the release of these electrons is induced by light stimulation and measured escaping dose. A key consideration in OSL dating is the former depositional context (largely shallow marine settings in our dataset) and its impact on bleaching (zeroing) of the sediments by sunlight, as well as the burial history of the sediment (multiple ages in Svalbard; Alexanderson and Landvik, 2018; Alexanderson et al., 2011a). Generally, OSL is considered to be more reliable than TL and IRSL.

### 160    4.1.8    Electron spin resonance (ESR) dating

This method estimated the time since deposition of certain materials (largely molluscs in our database) by measuring the trapping of electrons within the material's crystal lattice. A key factor is the radiation occurring from the enclosing sediment, as well as the radioactivity of the sample (Grün, 1989). The precision of the age from ESR dates on mollusc shells is complicated, since the shells have an open system to uranium (Schellmann and Radkte, 1999). As a result, care must be
taken in interpreting ESR ages, and the uncertainty can be larger than the stated analytical uncertainty. The analytical techniques used to determine ESR ages for many of the sites in our database are described in detail in Molodkov (1988) and Molodkov et al. (1998).

### 4.1.9    U/Th dating

As with ESR dating, U/Th dates of mollusc shells are complicated to interpret since the shells have an open system to
uranium, and relies on assumptions on the exchange of the element with the surrounding environment (Radke et al., 1985). As a result, this technique is not commonly applied to LIG deposits. U/Th dating of mollusc shells was used to constrain the age of deposits at two sites in our database (Miller et al., 1977, Israelson et al., 1994). For the later, the ages provided only minimum ages.

### 4.1.10    Tephrochronology

At one site in our database (Galtalækur site in Iceland; Vliet-Lanoë et al., 2018), tephrochronology is used to support the MIS 5e age assignment. Contained within the marine sediments is a tephra layer that was linked via geochemical analyses to



a specific eruption (Grimsvötn volcano) which was, in turn, constrained to MIS 5e based on the position of the Grimsvötn tephra in North Atlantic marine sediment cores (Davies et al., 2014)

### 4.1.11    K-Ar dating

At one site in our database (Galtalækur site in Iceland; Vliet-Lanoë et al., 2018) potassium-argon dating is used to constrain the age of a glacio-volcanic unit that underlies the MIS 5e marine unit. This method measured the rate of decay from potassium to argon and was possible at this site given the presence of volcanic rocks and is otherwise not common to Quaternary research.

## 5    QUALITY ASSESSMENT

In the WALIS database, there is a quality assessment rating for the RSL proxies and the precision of the age. In the WALIS documentation, the standard RSL rating is designed for far-field sea level indicators, in which a precise assignment of sea level position can often be determined. For glaciated areas where sea level was rapidly changing due to glacial isostatic effects after the end of the MIS 6 glaciation, this type of quality assessment is not as useful, especially since the dating techniques are not precise enough to precisely pinpoint when the sediments were deposited. As one of the primary uses of

this database will be GIA modelling, we have devised a rating scheme to assess the usefulness for this purpose (Table 3). The best quality RSL proxies are sedimentary sequences that have well documented elevation measurements, and there is a clear transition from deep marine, shallow marine, marginal, and terrestrial environments. A high rating is also assigned if it can be proved that sea level position remained above a threshold for a long period of time, *i.e.* for most or all of the LIG. The rating decreases when there is less geological evidence sea level position changes and proximity to sea level.


The age rating follows the WALIS standard (Table 4) and is determined by how precise the age of the deposit can be determined. As a result of this, almost all of the proxies are set to have a score of 2 (not numerically age constrained and based on relative ages or environmental conditions) or 3 (numerically age constrained). Higher scores in WALIS are generally only assigned for U/Th dated corals that can have precision of less than a few thousand years, which is not the case

in the techniques applied to glaciated region sea level proxies. Despite this, based on stratigraphy and environmental proxies, there is reasonable confidence that all of the proxies documented in our database fall either in the LIG, the latest part of MIS 6 when the ice sheets were retreating, or during the early parts of MIS 5d when the ice sheets began to grow again. Without the environmental indicators, it can be difficult to distinguish deposits that might instead date to the sea level highstand that happened after the MIS 5d glaciation (*e.g.* Mangerud et al., 1998).





## 6    RELATIVE SEA LEVEL PROXIES

Here, we describe sites from the glaciated Northern Hemisphere that contain *in situ* LIG marine sediments, ordered roughly from east to west, and sorted by country. Sediments overlying or underlying the marine strata are generalized unless they provide additional context for the LIG marine event. We offer no interpretation of tectonics, glacio-isostatic adjustment or eustacy. Elevation measurements are in MASL (meters above sea level) or MBSL (meters below sea level). Owing to the global scale of this database, it is not possible to map all features/locations described in Figure 1 and the reader is referred to the original publications for additional site information. The elevation of marine sediments, along with quality scores for both RSL and age determinations are summarized in Table 5 and detailed in the database of Dalton et al., (2021).

### 6.1    Novorybnoye 2, Taimyr Peninsula, Russia (72.83, 105.79)

On the southernmost Taimyr Peninsula, ~30 m-high river bluffs close to the small settlement of Novorybnoye, on the south shore of the Khatanga River, expose a complex Mid- to Late Pleistocene stratigraphy. As described in Kind and Leonov (1982), this record encompasses three glacial till units on top of Cretaceous sandstone; the till beds are described as interbedded with two marine sediment successions, the lowermost to be, in their terminology, Kazantsevo (i.e. MIS 5e, LIG) in age and the upper one sandwiched between two Early Zyryanka tills (MIS 5d–5a). This chronostratigraphy was, however, not substantiated by any numerical ages. The Novorybnoye bluffs were reinvestigated by Möller et al. (2019a; 2019b), resulting in two more observed marine units and a very different chronology, in which the lowermost marine unit probably dates to MIS 9-11. Relevant to the present study, the LIG marine sediments span 14.8 to 24 MASL and are divided into two units: F1 and F2. Marine unit F1 (spanning 14.8 to 21.5 MASL) is at the base a glaciomarine clayey silt with numerous occurrences of ice-rafted debris. Above these marine sediments are stratified and normally graded sandy shoreface sediment (F2) with an abundance of *Hiatella arctica* and *Astarte sp*. shells from 21.5 to 24 MASL. An OSL age of 124 ka is supported by a mollusc ESR date at 131 ka; Möller et al. (2019a) thus concluded that this entire marine unit (F1–F2) represents marine inundation following deglaciation and sediment deposition during isostasy-driven shore regression at the transition from MIS 6 into the Karginsky interglacial (MIS 5e). The absolute height of unit F sediments is ≥25 MASL, which is a minimum deglacial sea level altitude.

### 6.2    Bol'shaya Balakhnya River (BBR 17), Taimyr Peninsula, Russia (73.62, 105.36)

In the lower reaches of the Bol'shaya Balakhnya River, a sediment succession shows marine silty clay with dispersed ice-rafted debris situated between 6.2 to 7 MASL at site BBR 17B and 7 to 13 MASL at site BBR 17A. At BBR 17 B, there is an upper erosional contact to fluvial sediment at 7 MASL (Möller et al., 2019a; Möller et al. 2019b; Der Sarkissian et al., 2020). The mollusc fauna in the marine sediments is dominated by arctic *Portlandia arctica*, but there is also an abundant occurrence of subartic taxa: *Buccinum undatum, Mytilus edulis* and *Macoma baltica*, suggesting higher than present influx of Atlantic water. ESR ages on *P. arctica* are 101 ka to 105 ka (n = 3) and indicate an Early Zyryanka age (MIS 5c). However,



two molluscs in the above-lying fluvial sediments (OSL-dated to a MIS 3 age), the molluscs redeposited from erosion of the marine sediment, yield ESR ages of 122 ka and 123 ka. Theses dates, together with the interglacial-type mollusc fauna composition clearly set the marine sediments into the Karginsky interglacial (MIS 5e; LIG), however with poor indication of sea level at deposition.

### 6.3    Kamennaya River, Taimyr Peninsula, Russia (76.53, 103.52)

Around the Kamennaya River, which is a tributary to the Leningradskaya River, Gudina et al. (1983) described several sites exposing marine and nearshore marine sediments forming regressive terraces from 133 down to 40 MASL (site with highest altitude, 133 MASL, site no 373, coordinates as above). The sediment successions are divided into a lower coarsening upwards member, from silts and clays to cross-bedded sand, and an upper member of sand and gravel. All sediments are rich in molluscs, predominantly *Astarte borealis, A. crenata, A. montuagi, Macoma calcarea, Hiatella arctica* and *Mya truncata*, (i.e. species that are arctic to non-conclusive in their biogeography). As opposed to the molluscs, the foraminifera association (48 species detected) has a dominance of subarctic species, thus suggesting warmer than present sea temperatures. Based on the latter, Gudina et al. (1983) suggested that the marine sediments in the Leningradskaya basin were deposited at shore regression during the Kazantsevo interglacial (Karginsky in terms of Astakhov (2013); i.e. MIS 5e). Gudina et al. (1983) do not present any numerical age data.

### 6.4    Kratnaya River sections, Taimyr Peninsula, Russia (77.5, 103.2)

Three river-cut sections along the Kratnaya River (denoted KR1, KR2 and KR3), expose a thick basal till on top of which are marine sediments with a slight upwards-coarsening trend with off-shore silt and sand grading into shoreface-deposited sand with gravel stringers (Möller et al., 2008; Möller et al., 2015). The base of these marine sediments is situated at 37 MASL and the uppermost logged sediments reach ~43 MASL, but the marine sediment succession can locally be followed upslope to at least 50 MASL. The sediments host a variable abundance of the molluscs with *Hiatella arctica, Mya truncata* and *Astarte borealis* as dominating taxa (all arctic to non-conclusive in their biogeography). Six ESR ages on molluscs from the marine sediment form an age cluster between 111 ka and 142 ka, while two OSL dates gave younger ages of 84 ka and 100 ka (Möller et al., 2008; Möller et al., 2015). The ESR ages firmly suggest a transition from Late Taz (MIS 6 glaciation) into the Karginsky (MIS 5e) interglacial for the emplacement of this marine sediment succession.

### 6.5    Anjeliko River sections, Taimyr Peninsula, Russia (77.35, 102.75)

Based on the composite stratigraphy of five river-cut sections (three of which have LIG sediments and are included in the database) along and in the vicinity of the Anjeliko River, Möller et al. (2008) and Möller et al. (2015) report three marine units intercalated with glacial tills and suggested that two of these marine events, together with underlying tills, represent two full glacial cycles coupled with marine inundation and regression following deglaciations. Relevant to the present study, the intermediate marine unit consists of glaciomarine debris flow sediments followed by an upwards coarsening sediment



succession of off-shore clay and silt into shore-face/fore-shore sand. These marine sediments are situated between 55 and 58 MASL. Outside of logged sediment successions, the marine sediments could be followed to ~80 MASL and, if observation from investigations by Mirošnikov (1959) and Šnejder (1989) from other locations on Cape Chelyuskin refer to the same sediments, they might have a highest altitude up to 140 MASL. The sediments host a variable abundance of the molluscs *Hiatella arctica, Mya truncata* and *Astarte borealis* as dominating taxa, all arctic to non-conclusive in their biogeography. Three ESR ages on molluscs from the marine sediment form an age cluster of 143 ka, 145 ka and 156 ka, while two OSL dated show age dispersal, 79 ka and 135 ka (see Möller et al., 2015). The age envelope (with disregarding the 79 ka OSL age) suggests a transition from Late Taz (MIS 6 glaciation) into Karginsky (MIS 5e) interglacial for this marine sediment succession. The uppermost glacial/marine sediment succession in the Anjeliko River area dates from a glaciation followed by marine inundation in the Early Zyryanka (MIS 5d–5c; Möller et al., 2008).

### 6.6 Ozernaya River, October Revolution Island, Russia (79.12, 96.92)

October Revolution Island is located offshore of the Taimyr Peninsula, in the Severnaya Zemlya archipelago, Arctic Russia. On this island, along the north to south-flowing Ozernaya River, are occasional exposures of up to 50-m thick paleo-valley fill sediment successions, ordered in a pancake-like stratigraphy. First briefly described by Bolshiyanov and Makeyev (1995), a more in-depth description was made in Möller et al. (2007), showing four till beds (Till I – IV) interbedded with three marine units (Marine I – III). Relevant to the present study, Marine unit III is situated between 75.5 and 80.5 MASL (sites Oz 1b and 1d). These marine sediments can laterally be followed to higher ground with no covering till (sites Oz 2 and 3; see Fig. 5A in Möller et al., 2015), eventually terminating in sets of beach ridges, the highest at ~140 MASL. In the database, we have treated the highest elevation beach ridge as a sea level indicator. The marine sediments show a general coarsening upward trend with off-shore to shore-face deposited silt and clay with varying frequency of ice-dropped clasts, continuing into massive or vaguely stratified sand (site Oz1) and finally beach-face gravels (sites Oz 2–3). Besides arctic molluscs as *Hiatella arctica, Mya truncata* and *Astarte borealis*, the marine sediments also host biogeographically subarctic such, as *Chlamys islandica* and *Buccinum undatum*. The foraminifera fauna is also mainly arctic, but a warmer-water indicator is *Trifarina* cf. *angulosa* and the also occurring *Elphidium ustulatum* and *Islandiella inflata* that are rarely found in deposits younger than MIS 5e in Europe (Möller et al., 2007).

The age of the Marine unit III is not straight forward, as discussed in Möller et al. (2007) and Möller et al. (2015); ESR ages on the mollusc fauna have and age envelope of 77–105 ka (n = 8), while ESR ages on nearby found *Chlamys islandica* is 105 and 120 ka (Bolshiyanov and Makeyev, 1995). GSL ages on the sediment show an age envelope of 143–176 ka (n = 11). Based on the stratigraphic position in a regional context, the interglacial fauna elements in the marine sediments and an evaluation of the numerical ages from performed datings, Möller et al. (2007) favoured an interpretation that the Marine unit III sediments in the Ozernaya River valley were deposited at the Taz/Karginsky (i.e. MIS 6/MIS 5e) transition into interglacial conditions and that the deglacial sea had a high-stand of at least 140 MASL.





### 6.7 Lower Agapa River, Taimyr Peninsula, Russia (71.6, 88.3)

In the lower reaches of the Agapa River, several river sections display three marine units (Gudina et al., 1968; later reinvestigated by Sukhorukova, 1998). The lowermost unit is interbedded fine sand and silt with the boreal bivalve *Cyprina islandica* (now *Arctica islandica*) followed by silt and clay (Unit 2), 30-35 m thick, spanning 30 to ≥63 MASL. Unit 2 hosts a rich mollusc fauna (16 subarctic (arcto-boreal) and 20 arctic species), however with no species list given. The foraminifera

fauna also suggests a component of Atlantic water inflow. Marine Unit 2 with interglacial-type marine fauna reaches ≥63 MASL, which is a minimum sea level altitude for these deeper marine sediments. No numerical age data are presented in Gudina et al. (1968), nor in Sukhorukova (1998). However, the presented biostratigraphy strongly suggests that Unit 2, with a warmer-water fauna, represents interglacial conditions, most probably MIS 5e (Karginsky).

### 6.8 Karginsky Cape, West Siberian Plain, Russia (69.95, 83.57)

At Karginsky Cape, Western Siberian Plain, Kind (1974) described a ~16-m thick sequence of marine sands and silts situated between 5 and 21 MASL, sandwiched between two tills. This is the marine stratotype of the Karginsky Formation (MIS 5e, LIG; see Figure 3). These marine sediments contain shells of the biogeographically arctic to non-conclusive molluscs *Astarte borealis*, *Macoma calcarea, Mya truncata* and *Ciliatocardium ciliatum,*as well as the subarctic *Mytilus edulis*. Shells of a typical boreal mollusc, *Arctica islandica,* were found only on the beach. The sequence also contains

remains of plants presently growing some 3-5° to the south, suggesting it was deposited during a warmer interval. Conventional radiocarbon dates obtained in this section are 42 ka, 42 ka, 46 ka and ≥52 ka BP. However, the first ESR date on an *Arctica islandica* shell yielded an age of 121.9 ka (Katzenberger and Grün, 1985; Arkhipov, 1989). The MIS 5e (LIG) age of this marine formation was later confirmed by 6 OSL dates in the range of 117 to 97 ka (Astakhov and Nazarov, 2010b; Nazarov, 2011; Nazarov et al., 2018, 2020).

### 6.9 Tanama River, Western Siberian Plain, Russia (2 sites)

Along the Tanama River, Nazarov et al., (2021) describe marine sediments overlying till from the Taz (MIS 6) glaciation and marine sediments from an earlier interglacial (interpreted as MIS 7 based on OSL dates). The marine sediments (known locally as the Payuta marine formation) consist of sands and silts, with numerous shells of *Arctica islandica*, which indicates warmer water conditions. The unit is associated with terraces that reach an elevation of 60 to 70 MASL. The change in slope

at 70 m took on the appearance of strandlines. Overlying the marine sediments are lacustrine and alluvial sediments that gave MIS 4-3 ages.

### 6.9.1 Tanama site 1 (70.24, 79.76)

The unit is associated with terraces that reach an elevation of between 60 and 70 MASL. The change in slope at 70 m took on the appearance of strandlines.





### 6.9.2    Tanama site 2 (69.83, 79.00)

The unit is associated with terraces that reach an elevation of between 60 and 70 MASL. The change in slope at 70 m took on the appearance of strandlines.

### 6.10    Bol'shaya Kheta, West Siberian Plain, Russia (4 sites)

In the Western Siberia Plain, there are LIG outcrops along the Bol'shaya Kheta River (Nazarov et al., 2020, 2021; Astakhov and Semionova, 2021). First described by Volkova (1958), these sites contain a similar identical stratigraphic record that can be traced along the river sections (Fig. 3), with LIG marine sediments generally ranging from 5 to 40 MASL. The upper marine formation was assigned to the Kazantsevo interglacial by Sachs (1953), in modern terminology corresponding to the Karginsky interglacial (MIS 5e; LIG). This marine unit contains only the extant species *Cyrtodaria siliqua* and *C. kurriana* and a rare occurrence of the boreal *Arctica islandica.*

### 6.10.1    Site 7251, Bol'shaya Kheta River (68.47, 83.12)

At this site, the upper marine sand, and clays (constrained to MIS 5e; LIG) are situated between 20 and 30 MASL (Fig. 3). Two OSL ages on these marine sediments yielded 124 ± 31 ka and 121 ± 11 ka (Nazarov et al., 2020, 2021; Astakhov and Semionova, 2021). These marine sediments directly overlie a till of MIS 6 age.

### 6.10.2    Site 7248, Bol'shaya Kheta River (67.97, 83.10)

Located 55 km south from the Northern River section, marine sand, and clays (constrained to MIS 5e; LIG) are located between 5 to 30 MASL (Fig. 3). Three OSL attempts on these marine sediments yielded 110 ± 16 ka, 127 ± 20 ka and 114 ± 12 ka (Nazarov et al., 2020, 2021; Astakhov and Semionova, 2021).

### 6.10.3    Site 7249, Bol'shaya Kheta River (68.00, 83.13)

At this site, marine sand, and clays (LIG) are located between 5 to 30 MASL (Fig. 3; Nazarov et al., 2021). No chronological constraints are available for this site, but the marine unit can be traced along the shoreline to the other sites dated to the LIG.

### 6.10.4    Site 7246, Bol'shaya Kheta River (67.96, 83.21)

Marine sand, and clays (LIG) are located between 5 to 30 MASL (Fig. 3; Nazarov et al., 2021). A single OSL age of 132 ± 11 ka suggests deposition during the LIG (Nazarov et al., 2021).

### 6.11    Observations Cape, West Siberian Plain, Russia (68.97, 76.10)

On the tip of the Taz Peninsula, Western Siberian Plain, the Observations Cape site contains parallel laminated sand, silt and clay with an abundance of boreal molluscs as *Modiolus* sp and *Zirphaea crispate* and subarctic species as *Buccinum*





*undatum, Macoma balthica* and *Mytilus.* Described and dated by Astakhov and Nazarov (2010b), these marine sediments are situated between 3 and 35 MASL and yielded 6 OSL dates with a mean age of 135 ka (Fig. 3), leading to a MIS 5e (LIG) age assignment. These marine sediments are underlain by Middle Pleistocene glacial till and overlain by fluvial sand with OSL ages at 77 and 74 ka BP.

### 6.12    Sula, Pechora Lowland, Russia (67, 50.34)

In the Pechora Lowland, the best dated MIS 5e (LIG) site is the succession of shoreline sands on the Sula River, a left tributary to the Pechora River (denoted sites 21/22). The sand formation was originally described as lying on top of marine clay with a cool-indicating mollusc fauna (Lavrova, 1949), the latter, however, not confirmed by later descriptions. The well-exposed marine sand spans the interval 31to 41 MASL, starting with thin foreshore gravel in tabular foresets, containing paired shells of subarctic *Mytilus edulis*. The fining upwards and bioturbated sand contains abundant shells of boreal molluscs *Arctica islandica* and rare shells of *Cerastoderma edule* and *Zirphaea crispata* – all species presently not living east of the Kola Peninsula. This mollusc fauna is typical for a shallow sea with positive bottom temperatures and occurs through the sand formation, topped by a cross-bedded beach gravel. The marine unit, attributed to the LIG, is overlain by fluvial sand and cryoturbated black silty clay of glaciolacustrine origin, topped by aeolian silt (Mangerud et al., 1999). An approximate MIS 5e age was later confirmed by numerous OSL dates; altogether 16 ages in the range from 90 to 128 ka with a mean age at $112 \pm 2$ ka (see Fig. 3, and Murray et al., 2007).

### 6.13    River Yangarei, Pechora Lowland, Russia (68.7, 61.83)

In addition to the Sula site, marine sediments of a MIS 5e age are also found at much higher elevations within the Pechora Lowland. As an example, marine sand with mollusc shells occur along the Yangarei River at 70 MASL, the sediments yielding OSL ages at $121.6 \pm 9.2$ ka and $114.8 \pm 8.9$ ka (Astakhov and Semionova, 2021).

### 6.14    Vorga-Yol Section, Pechora Lowland, Russia (66.70, 56.75)

Also in the Pechora Lowland, OSL ages of $126 \pm 8$ ka, $131 \pm 8$ ka and $149 \pm 10$ ka were obtained from sand with shell fragments  at 90 MASL, directly underlying the terminal glaciofluvial delta at Vorga-Yol section (Astakhov and Semionova, 2021).

### 6.15    Pyoza River, Arkhangelsk district, Russia (11 sites)

Marine sediments assigned to the Mikulinian (LIG) were first noted in stratigraphic records along the Pyoza River, Arkhangelsk district, Russia in the early-mid 20[th] century and examined more recently by Houmark-Nielsen et al., (2001) and Grøsfjeld et al., (2006). We compile 11 sites below.





### 6.15.1   Zaton site (65.58, 44.63)

First discovered by Ramsay (1904), the Zaton site (Site 0 of Houmark-Nielsen et al., 2001) was examined by Devyatova and
Loseva (1964) and Devyatova (1982). As described most recently by Grøsfjeld et al., (2006), the entire LIG marine sequence
spans 2 MASL to 11 MASL. At the base of the section, from 2 to 7.5 MASL are marine clays with a gradual transition into
silty sands, interpreted as an offshore to shoreface sediment succession (>45 m to <12 m water depth). At the top of the
section, from 7.5 to 10 MASL are laminated sand and silt, separated by gravel horizons. These sediments are interpreted as
deposited in a strong coastal/tidal environment with channel erosion and infilling (foreshore environment; <12 m depth).
Capping this sediment succession, from 10 to 11 MASL, is cross-bedded fluvial sand. Marine molluscs, dinoflagellate cysts
and benthic foraminifera are present throughout (Grøsfjeld et al., 2006). The stratigraphy observed at this site is laterally
continuous for ~800 m.

Several geochronological attempts have been made at the Zaton site. Early TL dating on sediments overlying the marine
sediments yielded an age of 93 ka, which supports a LIG-age for the underlying marine sediments (Hütt et al., 1985). Amino
acid dating on shells from the marine unit yielded mean D/L ratio of 0.051±0.006, which is "slightly higher than expected"
but "probably Eemian" according to Miller and Mangerud (1985). Subsequent ESR dating confirms a LIG age for this
marine deposit (120 ka to 82 ka; Molodkov and Raukas, 1988; Molodkov and Bolikhovskaya, 2002). A LIG-age is also
suggested by pollen analyses (Devyatova, 1982); according to the established pollen-based Eemian climate for western
Europe (Zagwijn, 1996), marine sediments at this site span the entire interval from ~133 ka to ~119.5 ka (Grøsfjeld et al.,
2006). Moreover, molluscs from this site suggest warmer-than-present day conditions (notably, *Corbula gibba* and *Balanus
improvises*), which also supports a LIG age (Grøsfjeld et al., 2006).

### 6.15.2   Bychye site (65.79, 45.06)

The Bychye site (sometimes spelled "Bychie"; Site 1 of Houmark-Nielsen et al., 2001 and Grøsfjeld et al., 2006) was first
described Devyatova and Loseva (1964) and again by Devyatova (1982). As described most recently by Grøsfjeld et al.
(2006), the stratigraphy at this site is similar to the Zaton site. The entire LIG marine sequence spans 12.5 to 23 MASL. At
the modern-day river level (~12 MASL) is a till unit, which is overlain by marine clays which gradually coarsen into clayey
silt (12.5 to 19 MASL; >45 m water depth). Theses sediments are terminated by an erosional horizon ( at 19 MASL),
indicating falling sea level (Grøsfjeld et al., 2006), followed by laminated sand and silt separated by gravel horizons with
channel incisions (from 19 to 23 MASL), interpreted as a regression sequence (shoreface/tidal environment; <12 m water
depth). Marine molluscs, dinoflagellate cysts and benthic foraminifera (Grøsfjeld et al., 2006) are present throughout. The
stratigraphy at this site is laterally continuous for ~500 m.





The marine sediments at the Bychye site are assigned to the LIG based on palaeoenvironmental and stratigraphic inferences. Pollen data (examined by Devyatova, 1982) suggest correlation to the ~133 to ~124-ka interval (Grøsfjeld et al., 2006) LIG climate for western Europe (Zagwijn, 1996). Various marine molluscs, dinoflagellate cysts and benthic foraminifera suggest a transition from cooler-than-present to warmer-than-present, which supports the capturing of the MIS 6 deglaciation,
followed by establishment of warmer marine conditions (Grøsfjeld et al., 2006).

### 6.15.3    Viryuga W (65.82, 46.00)

Located on the northern side of the Pyoza River, the Viryuga W site (Site 4 of Houmark-Nielsen et al., 2001 and Grøsfjeld et al., 2006) was first described by Devyatova and Loseva (1964). Our descriptions are derived from the more recent examination by Houmark-Nielsen et al. (2001) and Grøsfjeld et al. (2006). At the base is a marine unit (spanning 21to 39
MASL) containing stratified sands with shells. A till is present between 39 and 45 MASL. Capping the stratigraphic section is an upper marine unit (spanning 46 to 49 MASL), consisting of a clayey diamict containing abundant *Mya truncata* and *Macoma calcarea* shells, often paired and thus suggesting *in-situ* preservation. Dinoflagellate cysts and benthic foraminifera are present throughout (Grøsfjeld et al., 2006).

The lower marine unit yielded OSL ages ranging from 237 to 194 ka, suggesting an ice-free interval during MIS 6 (Houmark-Nielsen et al., 2001). Accordingly, Grøsfjeld et al. (2006) interpreted the upper marine sediments as the earliest interval of the LIG. Correlation of pollen data from this site with the climate for western Europe (Zagwijn, 1996) place the upper marine sediment unit between ~133 ka and ~130 ka (Grøsfjeld et al., 2006). Marine molluscs and benthic foraminifera suggest cooler-than-present conditions during this time, which suggests deposition during the deglacial phase at the
beginning of the LIG (Grøsfjeld et al., 2006). However, because the upper marine sediments themselves remain undated and the stratigraphic context is unclear, earlier workers (Devyatova and Loseva, 1964; Houmark-Nielsen et al., 2001) interpreted this upper unit as Weichselian-aged (e.g. late MIS 5).

### 6.15.4    Viryuga E (65.80, 45.99)

The Viryuga E site (Site 6 of Houmark-Nielsen et al., 2001 and Grøsfjeld et al., 2006) was first described by Devyatova and
440 Loseva (1964) and again by Grøsfjeld et al., (2006). Following Grøsfjeld et al., (2006), the basal interval at this site is a clayey till between 56 to 58 MASL, suggested to have been emplaced during the MIS 6 glaciation. The till is, with a sharp boundary, overlain by clayey glaciomarine sediment between 58 and 60 MASL, interpreted as deposited during marine inundation and sea-level high stand following ice sheet collapse. Capping the stratigraphy is marine, mollusc-bearing cross-bedded sand with gravel and clay bedding (60 to 63 MASL), interpreted as shoreface deposits. Marine molluscs,
dinoflagellate cysts and benthic foraminifera are present throughout (Grøsfjeld et al., 2006). Marine molluscs and benthic foraminifera suggest cooler-than-present conditions during deposition, followed by warmer-than present molluscs; together, these data suggest deposition during the deglacial phase at the beginning of the LIG and then the early LIG (Grøsfjeld et al.,



2006). Water depths were likely >45 m at time of deposition. No direct dates are available at this site. However, correlation of pollen data from this site with climate for western Europe (Zagwijn, 1996), along with the stratigraphic context, place the marine sediments between ~133 and ~130 ka (Grøsfjeld et al., 2006).

### 6.15.5 Kalinov (65.79, 46.22)

The Kalinov site (Site 8 of Houmark-Nielsen et al., 2001 and Grøsfjeld et al., 2006) displays at the base marine clays interbedded with sands (situated from 28 and 37 MASL), the sediment succession interpreted as deposited in a lower shoreface environment (Grøsfjeld et al., 2006). The marine sediments are capped by fluvial sands from 38 to 40 MASL. The marine clay interval contains marine molluscs, dinoflagellate cysts and benthic foraminifera (Grøsfjeld et al., 2006). Correlation of pollen data from this site with climate for western Europe (Zagwijn, 1996) place the marine sediment succession between ~133 ka and ~130 ka (Grøsfjeld et al., 2006), although there are no direct dates at this site. Marine molluscs and benthic foraminifera suggest cooler-than-present conditions during this time, which suggest deposition during the deglacial phase at the beginning of the LIG (Grøsfjeld et al., 2006).

### 6.15.6 Yatsevets (65.70, 46.52)

At the base of the Yatsevets site (Site 10 of Houmark-Nielsen et al., 2001 and Grøsfjeld et al., 2006) is a till situated between 32 and 34 MASL. Overlying this till is a sequence of marine clay between 33 and 38 MASL, containing marine molluscs, dinoflagellate cysts and benthic foraminifera (Grøsfjeld et al., 2006). Molluscs and benthic foraminifera suggest cooler-than-present conditions during sediment deposition and with an increasing water depth (>45 m depth), which support deposition during the deglacial phase at the beginning of the LIG (Grøsfjeld et al., 2006). Correlation of pollen data (Devyatova and Loseva, 1964) from this site with the climate for western Europe (Zagwijn, 1996) place the marine sediment deposition between ~133 and ~130 ka (Grøsfjeld et al., 2006), although there are no direct dates at this site.

### 6.15.7 Site 11 Orlovets (65.71, 46.84)

The Orlovets site displays almost entirely laminated marine silts that coarsen upward (from 38 and 43.5 MASL), the sediment succession interpreted as deposited in a lower shoreface environment (Grøsfjeld et al., 2006). The laminated silts are capped by sand and gravel, interpreted as deposited in a foreshore environment (Grøsfjeld et al., 2006). Numerous marine molluscs, dinoflagellate cysts and benthic foraminifera are present throughout (Grøsfjeld et al., 2006), all suggesting slightly warmer marine conditions than present-day conditions (Grøsfjeld et al., 2006). No direct dates are available for the marine sediments; however, according to the established Eemian climate for western Europe (Zagwijn, 1996), marine sediments at this site are suggested to span an age interval from ~128 to ~124 ka (Grøsfjeld et al., 2006).



### 6.15.8   Site 12 Orlovets (65.69, 46.93)

The stratigraphic record at Site 12 Orlovets is identical to that of Site 11 Orlovets. Laminated marine silts that coarsen upward (from 38 to 43.5 MASL) were interpreted as deposited in a lower shoreface environment (Grøsfjeld et al., 2006). At this site, however, the marine interval is capped by sand, interpreted to be deposited in a fluvial environment (Grøsfjeld et al., 2006).

### 6.15.9   Site 13 Yolkino (65.68, 47.60)

Site 13 Yolkino was first described by Devyatova and Loseva (1964) and again by Houmark-Nielsen et al. (2001) and Grøsfjeld et al. (2006). At the base of the section is marine sand containing molluscs, dinoflagellate cysts and benthic foraminifera, between 48 and 51 MASL. There are no direct age constraints on the sediments at this site. However, correlation of pollen data from this site with the climate for western Europe (Zagwijn, 1996) place marine sediment deposition between ~130 and ~128 ka (Grøsfjeld et al., 2006). The marine molluscs suggest warmer than present-day conditions (Grøsfjeld et al., 2006). The marine sediments are overlain by a series of organic-bearing lacustrine sediments, dated by OSL to 89 ka (Houmark-Nielsen et al., 2001), as well as tills and fluvial sediments.

### 6.15.10  Site 14 Yolkino (65.68, 47.60)

The stratigraphic record at Site 14 Yolkino is identical to that of Site 13 Yolkino. Importantly, marine sand containing molluscs, dinoflagellate cysts and benthic foraminifera are situated at the base of this stratigraphic section between 48 and 51 MASL. An OSL age on these marine sediments suggests deposition at 124 ka (Houmark-Nielsen et al., 2001).

### 6.15.11  Burdui (65.67, 48.06)

Marine sand at the Burdui site (Site 24 of Houmark-Nielsen et al., 2001 and Grøsfjeld et al., 2006) is situated between 59
and 60 MASL. These sediments are overlain by a 1-m interval of proglacial sand, dated by OSL to 97 ka (Houmark-Nielsen et al., 2001). Marine molluscs, dinoflagellate cysts and benthic foraminifera are present throughout (Grøsfjeld et al., 2006). Analysis of the marine molluscs suggest warmer conditions than present-day (Grøsfjeld et al., 2006). According to the established Eemian climate for western Europe (Zagwijn, 1996), the marine sediments at this site seemingly span an age interval from ~131 to ~130 ka (Grøsfjeld et al., 2006).

## 6.16   Ponoi River, Kola Peninsula, Russia (67.078, 41.131)

The Ponoi site is in the Ponoi River Valley on the eastern part of the Kola Peninsula. The most complete section in the area is about 20-m thick and has been studied by Lavrova (1960), Nikonov (1966), Gudina and Yevzerov (1973), Ikonen and Ekman (2001) and Korsakova et al. (2016). According to the latter two studies, a till (~MIS 6) at the base of the section is overlain by a marine clay unit (denoted unit 2) with sand and gravel interbeds. This clay unit is situated between 7 and 11



MASL and contains sporadic unbroken mollusc shells and shell fragments. The mollusc, foraminifera, and diatoms, together
with palynological and lithostratigraphical evidence, indicate that the marine clay can be correlated with the Eemian
interglacial stage (i.e. LIG; MIS 5 e). Overlying this marine unit are stratified sands and gravels with shell detritus and
unbroken shells. This upper unit is 8 m thick and has a sharp lower contact with the underlying marine clays (Korsakova et
al., 2016). Mollusc shells and sand in this upper unit have been dated with ESR and OSL to between 96±8 ka (ESR) and
71.9±8.2 ka (OSL) indicating that this upper sand was deposited during the Early Weichselian substage (approximately MIS
5 a–c; see Korsakova et al., 2016). This age assignment supports and earlier LIG age for the underlying marine clays.
Capping the entire stratigraphic section is a glacial unit consisting of glaciolacustrine silt with clast and a till unit (Korsakova
et al. 2016).

### 6.17    Svyatoi Nos, Kola Peninsula, Russia (68.016, 39.874)

The Svyatoi Nos site is located on the northeastern coast of the Kola Peninsula. As described by Korsakova (2019, *in press*),
the Moscovian (~MIS 6) glaciomarine sediments are overlain by marine mollusc-bearing silty sands that are present between
11 and 16 MASL. The mollusc species indicate sublittoral faunal assemblage (e.g. Ikonen and Ekman, 2001) and temperate
saline water conditions (e.g. Korsakova, 2019). Based mainly on mollusc, foraminifera, and pollen results, the silty sands are
thought to have been deposited during the LIG (MIS 5e) (Gudina and Yevzerov, 1973; Ikonen and Ekman, 2001; Korsakova,
2009, 2019) despite the slightly younger IRSL age of 109.9 ± 10.9 ka obtained from these sands.

### 6.18    Chapoma, Kola Peninsula, Russia (66.114, 38.970)

In the southeastern part of the Kola Peninsula, exposures at the Chapoma site occur in the river terraces on the bank of the
River Chapoma, about 3.4 km from the river mouth (Gudina and Yevzerov, 1973). The exposure is approximately 25 m
high. A till bed at the base is suggested to have been deposited during the Moscovian glaciation (~MIS 6) (Gudina and
Yevzerov, 1973; Korsakova et al., 2004; Korsakova, 2019). Relevant to the present study, the till bed is overlain by clay, silt
and sandy silt beds that spans 6.5 to 10 MASL, and with abundant mollusc shells (Korsakova, 2019). Foraminifera fauna
identified from these marine sediments include a rich and relatively warmwater-indicating fauna with species such as
*Bulimina aculeata, Bolivina pseudoplicata and Hyalina baltica* (e.g. Gudina and Yevzerov, 1973; Ikonen and Ekman, 2001).
Pollen data from the marine sediments indicate a succession from a closed to open *Betula–Pinus* forests. ESR dating
obtained from molluscs at around 9 MASL yielded ages of 128 ± 7.5 ka and 138.5 ± 9.6 ka, which supports deposition
during the LIG. An additional marine bed at this section post-dates the LIG and was likely deposited during MIS 5a (see
Discussion).

### 6.19    Strelna River, Kola Peninsula, Russia (66.011, 33.637)

Located just south of the Kola Peninsula, the Strelna site is situated approximately 7 km upstream from the mouth of the
River Strelna. Many workers, for example Grave et al. (1969) have studied the litho- and biostratigraphy for this >16-m thick



sediment succession. The most recent summary of the previous work is given in Korsakova (2019). At the base of this site is a 2.5 m-thick unit of mollusc-bearing fine sand and silt situated between 33 and 35.5 MASL. The ESR-dated molluscs from this unit yielded an age of 111.5±11.2 ka. Taken together, the mollusc and diatom assemblages suggest that the basal marine sediments were deposited in a coastal environment, possibly indicating a contemporary sea level above 36 MASL

(Korsakova et al., 2016). Pollen evidence suggest that a birch and pine forests existed in the area (Grave et al., 1969). Accordingly, this unit is most likely correlative with the LIG (MIS 5e) (e.g. Korsakova et al., 2004; Korsakova, 2019).

### 6.20 Varzuga, Kola Peninsula, Russia (66.399, 36.641)

On the southern Kola Peninsula, exposures along the banks of the Varzuga River between the Koytolov and Kletnoy rapids, south of Varzuga village, have been studied by Lavrova (1960), Gudina and Yevzerov (1973), Apukthin (1978) and Lunkka

et al. (2018). An exposure located on the right bank of the Kletnoy rapid consists of a marine mollusc-bearing clayey silt at the base of the section (situated between 10 and 14 MASL), conformably overlain by glaciolacustrine silt and sand-rich silt (Site 'S1' of Lunkka et al., 2018). Pollen, diatom and foraminifera evidence indicate that the marine silt was deposited in a sublittoral zone at water depths of 40–50 m during an interglacial stage, correlated with the Eemian interglacial (MIS 5e) (Ikonen and Ekman, 2001, and references therein; Lunkka et al., 2018). Although the sedimentary sequence along the banks

of the Varzuga River is glaciotectonized in places (Apukthin, 1978), the marine clay/silt situated between 10 and 14 MASL is *in situ*, where it is conformably overlain by glaciolacustrine silt at around 14 MASL. These overlying glaciolacustrine silts are thought to have been deposited during the Early Weichselian prior to 88 ka (Lunkka et al., 2018). Capping the stratigraphic sections are two till units interbedded with sand and silt (Lunkka et al., 2018).

### 6.21 Petrozavodsk, Western Russia (61.77, 34.40)

Marine sediments assigned to the LIG were first noted in a borehole record from Petrozavodsk, western Russia, described by Wollosovich (1908). These sediments are located at 40 MASL and conformably overlie glaciolacustrine clays (Wollosovich, 1908; Lukashov, 1982). A marine origin for these sediments is confirmed by saline diatom taxa (Ikonen and Ekman, 2001) and several mollusc species, suggesting saline conditions on the order of 10 to 15 ‰ (Funder et al., 2002). No geochronological data are available for these sediments; instead, the LIG-age is based on correlation of the local pollen

record to an established Eemian (MIS 5e; LIG) pollen assemblage (Lukashov, 1982; Ikonen and Ekman, 2001). Based on the pollen taxa and its correlation to the saline diatom taxa, Ikonen and Ekman (2001) showed that the marine phase prevailed in the area for a long time during the LIG, i.e. from Pinus-Betula PAZ to Picea-Alnus-Carpinus PAZ (possibly between 130 ka to 124 ka).

### 6.22 Peski, Western Russia (60.15, 29.29)

Miettinen et al. (2002) describe a 32-m borehole record from the Peski site in western Russia. Sediments between 9 and 13.5 MASL were interpreted to correlate to the LIG. The base of the borehole is a till (interpreted as deposited during the MIS 6





glaciation), followed by a dark bluish, organic-bearing clay and silt deposit containing *Portlandia arctica* molluscs between 10.5 and 11.5 MASL, interpreted as deposited during relatively deep-water marine conditions. The marine deposit is assigned to MIS 5e based on the palaeoecological and stratigraphic context. Pollen data from this interval suggest warm conditions, especially by the occurrence of *Corylus* and *Carpinus*, which are associated with the climatic optimum of MIS 5e (Miettinen et al., 2002). Diatoms contained in the marine unit suggest relatively deep-water, planktonic conditions, possibly representing the maximum of the marine transgression (Miettinen et al., 2002). Marine conditions were confirmed by the subsequent identification of dinoflagellate cysts and foraminifera in these sediments (Miettinen et al., 2014). The LIG unit gradually transitions to Early Weichselian deposition, which is distinguished by a change to grey colour and cooler conditions deduced from pollen and diatoms. This likely indicates that sea level remained above 13.5 m for all of the LIG. Stratigraphically, the marine unit is overlain by a 16-m thick till associated with the advance of MIS 5d/b ice sheets (Miettinen et al., 2002). No geochronological attempts were made at this site.

### 6.23   Põhja-Uhtju, Estonia (59.68, 26.51)

Miettinen et al. (2002) describe a Quaternary sediment sequence in a 70-m deep borehole from the Põhja-Uhtju island in Estonia. The base of the record showed sands and silty clay, interpreted as deposited during the deglaciation of MIS 6 ice. Next, located at 51 to 49 MBSL, was clay and silt, interpreted as marine in origin, suggesting that contemporary sea level could have been above present sea level. Overlying these marine sediments were ~35 m of silty clays and tills associated with later ice advances. The interval of 51 to 49 MBSL is assigned to the MIS 5e marine incursion based on palaeoecological and stratigraphic context. Pollen data suggest warm conditions, especially *Corylus* and *Carpinus*, which are associated with the climatic optimum of MIS 5e (Miettinen et al., 2002). The diatom assemblage in these sediments suggest shallow, marine to brackish water conditions. No geochronological attempts were made at this site.

### 6.24   Suur-Prangli, Estonia (59.62, 25.01)

Similar to the nearby island of Põhja-Uhtju, marine sediments assigned to MIS 5e have also been reported from a borehole record at Suur-Prangli, Estonia. Liivrand (1987) reported a silt/clay sediment succession between 61 and 75 MBSL that is both overlain and underlain by tills. The diatom record suggests brackish conditions followed by a shallow marine environment (Liivrand, 1991). These marine sediments are assigned to MIS 5e based on stratigraphic position (bracketed by tills assigned to MIS 6 and MIS 4) and paleoclimate succession recorded in pollen at that site, notably, a maximum *Picea* and *Carpinus* interval (Liivrand, 1991). As further justification for the MIS 5e age assignment, the pollen spectra observed at Suur-Prangli shows a similar succession to nearby pollen successions (Forsström and Punkari, 1997) and a well-dated Eemian pollen record from Germany (Field et al., 1994)





## 6.25 Lower Vistula Region, Poland (2 sites)

Extensive late Pleistocene stratigraphic records in the Lower Vistula region of Poland were first described by Roemer (1864). We describe two representative sites that later have been subject to extensive stratigraphic, sedimentological and palaeoecological analyses. However, we note that other occurrences of the MIS 5e marine record are also present in this area
(see Makowska, 1986).

### 6.25.1 Obrzynowo (53.78, 19.27)

A 212-m borehole record from Obrzynowo, Poland, was first described in an unpublished report from the Polish Geological Institute at Warsaw University (Janczyk-Kopikowa and Marks, 2002), subsequently published in Knudsen et al. (2012). This borehole record (ground surface is at 104.5 MASL) contains a sequence of glacial and interglacial deposits and intersects
marine sediments between 108 and 115 m depth, which situates the entire marine sequence between 10.5 and 3.5 MBSL (Janczyk-Kopikowa and Marks, 2002; Knudsen et al., 2012). This record captures a beach/shoreline environment from 10.5 to 6.5 MBSL (fine-grained sand and silt with mollusc shells) that gradually transitions into deeper-water marine deposit from 6.5 to 3.5 MBSL (clays, silts and very fine sands). Dinoflagellate cysts and diatoms confirm saline conditions (Knudsen et al., 2012). These marine sediments are both overlain and underlain by tills, fluvial sediments, varved clays and organic-rich
sediments and this site likely documented several glacial-interglacial transitions over the late Quaternary. The marine sequence at Obrzynowo has been assigned to MIS 5e based on stratigraphic position and comparison to the Licze core (discussed next; Head et al., 2005), as well as comparison to other regional pollen successions that have been assigned to the Eemian (an interpretation especially based on the presence of *Picea* and *Carpinus*; Mamakowa, 1989, 1988).

### 6.25.2 Licze (53.75, 19.13)

A borehole record at the Licze site (ground surface at 87 MASL) encountered marine sediments between 95 m and 105 m depth; therefore, the marine sediments are situated between 8 and 14.5 MBSL (Makowska et al., 2001; Zawidzka, 1997). Several inferences can be made regarding the local sea level from these marine sediments. Notably, the interval between 14.5 and 10.6 MBSL contains sands and clay balls that suggest a beach environment (Makowska et al., 2001). This interval´s sediments host freshwater molluscs that are gradually replaced by a marine assemblage, with the first marine molluscs
present at 13.5 MBSL and they remain a common occurrence until 8 MBSL (Mamakowa, 1989). Dinoflagellate cysts and diatoms confirm marine conditions through this stratigraphic interval and suggest warmer waters than present-day (Head et al., 2005; Knudsen et al., 2012). Throughout this interval with sediment transition from silty sand to silt and clay suggests an environmental change (transgression) from a beach environment to deeper marine conditions (Makowska et al., 2001). These marine sediments are both overlain and underlain by extensive Quaternary deposits (tills, fluvial deposits, varved clays and
organic-rich sediments). These marine sediments are assigned to MIS 5e owing to their stratigraphic position as well as due



to comparison with the pollen record at the Obrzynowo site (Head et al., 2005) and to other regional pollen successions, assigned to the Eemian (especially due to the presence of *Picea* and *Carpinus*; Mamakowa, 1989, 1988)

### 6.26    Rewal coastline, Poland (3 sites)

Marine sediments bracketed by tills were described in several borehole records along the Rewal coastline, Poland, by
Krzyszkowski et al. (1999) as well as Krzyszkowski (2010). The marine sediments consist of fine-grained sand (known as the Rewal Sand) that contains a rich assemblage or plant detritus and marine shells (largely boreal *Cardium* sp. (now *Cerastoderma)*, but also *Astarte borealis*, and the boreal *Thracia popyracea;* Krzymińska, 1996). Krzyszkowski et al. (1999) suggest these sediments represent either a lagoon or beach environment and they are assigned to MIS 5e based on their stratigraphic position. Both sites are overlain by the Ninikowo Till (assigned to the last glacial cycle) and underlain by the
Pustkowo Till (assigned to MIS 6 glaciation; Krzyszkowski et al., 1999). Here, we describe the position of LIG marine sediments in three representative boreholes from this region.

#### 6.26.1    Rewal borehole (54.09, 15.03)

The Rewal borehole (WH5) encountered a marine sand unit at an elevation between 5.5 and 13 MBSL (Krzyszkowski et al., 1999).

#### 6.26.2    Ciećmierz borehole (53.99, 15.03)

The Ciećmierz borehole intersected a thin lens of the Rewal Sands situated between 6.5 and 8 MBSL. Additional pollen work at this site suggests boreal conditions at the time of deposition of this marine unit (Krzyszkowski, 2010).

#### 6.26.3    Sliwin borehole (54.08, 15.01)

The Sliwin borehole (WH4) encountered the marine unit between 6.3 and 9 MBSL (Krzyszkowski et al., 1999).

### 6.27    Ollala, Finland (64.18, 25.35)

Forsström et al. (1988) described a sediment exposure and four boreholes at Ollala in central Finland on their lithology, pollen, plant macrofossil and diatom content. At borehole F, there is a till (>4 m thick) at the base above the Proterozoic crystalline bedrock. Overlying this till are < 1 m of glaciofluvial sands, which then changes into silt (situated between 116.25 and 116.5 MASL) and then thick greenish gyttja (situated from 116.5 to 117.5 MASL). The silt and gyttja intervals (together
comprising 116.25 to 117.5 MASL) contain diatom taxa indicating a shallow marine (saline and brackish water) basin passing upwards into fresh-water diatom taxa, typical for a lake basin. This change from saline to fresh-water taxa takes place in the lower part of the gyttja layer at 117 MASL, indicating that the fresh-water lake was isolated from the marine basin during the latter part of the MIS 5e (LIG), likely the result of glacio-isostatic uplift. Pollen data from this interval also





clearly suggest these sediments were deposited during interglacial conditions (Grönlund, 1991a, 1991b; Nenonen, 1995).
There are no finite absolute dates from the Ollala sediments. The gyttja unit is overlain by 0.5-m of glacially sheared and
compressed organic-rich material that is, in turn, overlain by up to 5 m thick Weichselian till (Forström et al., 1987;
Grönlund, 1991a). This marine gyttja unit is also present between 116.6 and 116.8 MASL at nearby borehole B.

### 6.28    Ukonkangas, Finland (63.916, 25.851)

The Ukonkangas site is a till-covered esker, located 8 km southeast of Kärsämäki, central Finland. According to Grönlund
(1991b) the Ukonkangas gravel pit exposed gravel at its base, overlain by ~0.5-metre-thick bluish, organic-bearing silt
(situated between 105.4 and 105.95 MASL). This silt bed is laterally continuous for 15 meters in the exposure, suggesting it
is *in situ*. Grönlund (1991b) interpreted the silt bed as deposited in a littoral zone of the Eemian (LIG) Baltic Sea (Fig. 3),
with time becoming more shallow and less saline. Pollen content indicates that the regional vegetation was forested and
composed of broad-leaved trees and other pollen (e.g. *Osmunda*) indicating Eemian interglacial temperate climatic
conditions (Eriksson, 1993). The diatom taxa are typical to the Eemian Baltic Sea taxa, which also occur at the Eemian sites
in adjacent areas (Grönlund 1991a, b). The  silt is overlain by parallel bedded sand, interpreted as beach sand (situated
between 105.95 and 108 MASL; Grönlund 1991b). The top 2 metres of the section is composed of till (MIS 2?). No
geochronological data are available for this site.

### 6.29    Viitala, Finland (62.598, 23.002)

At the Viitala site, located in western Finland, three borehole records documented a clay beneath the MIS 2 (Weichselian)
till (Nenonen et al., 1991). The clay unit is ~5 m thick. Diatom taxa indicate that the lower ~2 m of the clay was deposited in
a large cool and freshwater basin during the early part of the Eemian interglacial, and that the upper ~3 m was deposited in a
brackish/saline littoral zone of the Eemian Baltic Sea (e.g. Grönlund, 1991a). These marine sediments are situated between
81.5 and 84.5 MASL. Pollen data from the entire clay unit suggest a birch zone (lower ~2 m; freshwater sediments) and a
birch-pine-oak and hazel zone (upper ~3m; marine sediments). The interglacial-type pollen taxa and the stratigraphical
position of the clay indicates that it was deposited during the Eemian interglacial (Nenonen et al., 1991; Eriksson, 1993).

### 6.30    Mertuanoja, Finland (64.113, 24.586)

The Mertuanoja region is in the outskirts of Ylivieska, Finland, and is the stratotype area for the Eemian interglacial (LIG;
MIS 5e) for southern Finland (Nenonen, 1995; Eriksson et al., 1999). We present a composite of several sediment exposures
studied in the Mertuanoja area. At the base of the stratigraphy is a till, deposited during the Saalian glaciation (MIS 6) (e.g.
Nenonen, 1995; Lunkka et al., 2016). Overlying this till is the Mertuanoja Clay (average thickness ~1.6 m), consisting of
laminated silt and organic-bearing silty clay in its basal part overlain by a sand layer (from 2 to 20 cm thick), which is, in
turn, overlain by laminated silt. Diatom data suggest a fresh-water environment in the basal part of the Mertuanoja Clay,
changing to marine/brackish water, and then back to fresh-water conditions in laminated silt and clay above the sand layer





(Nenonen, 1995; Eriksson et al., 1999). Relevant to the present study is the transition from the fresh-water diatom taxa at the basal part of the Mertuanoja Clay into the marine taxa that takes place at around 59 MASL. Coastal marine conditions are recorded in this stratigraphic record until the sand layer at 60 to 61 MASL, interpreted as littoral sand. The pollen spectra from the Mertuanoja Clay suggest that it was it was deposited during interglacial conditions correlated to the Eemian interglacial (MIS 5e) (e.g. Grönlund, 1991a; Nenonen, 1995). No geochronological data are available for the Mertuanoja

Clay. However, a sand and gravel unit overlying the Mertuanoja Clay was OSL dated to 110±7 ka (Lunkka et al., 2019), followed by tills and an upper-most sand and gravel unit dated by and $^{14}$C to between 70 ka and 35 ka (Lunkka et al., 2016).

### 6.31  Norra Sannäs, Sweden (61.7830, 16.6931)

Sub-till marine sediments were intersected in boreholes in the vicinity of Lake Dellen, Sweden, during field work carried out in the late 1980s and early 1990s. Robertsson et al. (1997) described a representative borehole record from Norra Sannäs

(surface at 45 MASL) that shows a marine (littoral) sequence at 17.35 m depth in the core, which translates to 27.65 MASL. The diatom assemblage in clay sediments suggests marine to brackish conditions, which abruptly change to fresh-water diatoms at 17.1 m depth, suggesting the end of marine conditions (Robertsson et al., 1997). Stratigraphically, the marine sediments are overlain and underlain by tills that have been assigned to MIS 4 and MIS 6, respectively, thus suggesting a MIS 5e age for the marine sediments. The MIS 5e age assignment is also supported by pollen data from this site (Robertsson

et al., 1997), which bear a similar succession to other Eemian records from the region. The sole geochronological attempt at this site was a radiocarbon dating of the marine sediments that gave 25,480 ± 520 BP (Ua-1666); Robertsson et al. (1997) considered this age unreliable owing to a very low organic content at the sampled interval.

### 6.32  Fjøsanger, Norway (60.3431, 5.330)

The Fjøsanger site, located in the outskirts of Bergen City, is situated along a small fjord, well inside the extent of the

Scandinavian Ice Sheet during large Quaternary glaciations, including MIS 6 and MIS 2. The marine limit was 55 MASL during the last deglaciation, dated to 11,500 cal years BP (Mangerud et al., 2019). Relevant to the present study, at Fjøsanger there is a continuous sequence of shallow marine sediments, presently situated between 0 and 15 MASL, where molluscs, foraminifera, and pollen show that the climate changed from cold to warmer-than-present and back to cold (Mangerud et al., 1981). The sequence is covered by a till. The fauna, especially the boreal *Littorina littorea*, and the coarse-grained sediments

(gravel) suggest very shallow-water sediments, but beach sediments were not found; thus, sea level has been slightly higher. Correlation of the pollen stratigraphy with sites in the Netherlands, Denmark and Germany shows clearly that the warm period represents the Eemian, a conclusion supported by the occurrence of the marine gastropod *Bittium reticulatum*, which is known only from the Eemian in Europe (Mangerud et al., 1981), and by thermoluminescence ages (Hütt et al., 1983). Amino-acid stratigraphical correlation with classical European Eemian sites yielded slightly high D/L values for Fjøsanger,

but within uncertainty (Miller and Mangerud, 1985). A surprising, and indeed important, conclusion is that the relative sea level at Fjøsanger was above 15 MASL from late MIS 6, through the entire MIS 5e (≈ Eemian) and into MIS 5d (Fig. 4)



(Mangerud et al., 1981). We consider this as a secure conclusion because the sequence shows a complete Eemian and 15 MASL is the elevation of the top of the *in situ* marine beds. Fjøsanger was one of the first sites where it was demonstrated that the Eemian should be correlated with the MIS 5e, and not with the entire MIS 5 (Mangerud et al., 1979).

**6.33    Bø, Norway (59.3622, 5.276)**

The Bø site is located on the inland side of the large island Karmøy, on the extreme south-west coast of Norway. Like the Fjøsanger site, Bø is situated well inside the extent of the Scandinavian Ice Sheet during large Quaternary glaciations, including MIS 6 and MIS 2. The marine limit was about 16 MASL during the last deglaciation, dated to 17,500 cal years BP (Vasskog et al., 2019). Relevant to the present study, interglacial sediments were found between 1 MBSL and 6 MBSL in an
excavation, but only the upper 3 m could be sampled (Andersen et al., 1983). The interglacial sequence was covered by till and other sediments. The pollen stratigraphy (Andersen et al., 1983; Høeg, 1999) and amino-acid stratigraphy clearly shows that the interglacial is the Eemian (Miller and Mangerud 1985; Sejrup, 1987). The very early part of the interglacial is missing in the samples, but molluscs, foraminifera (Sejrup, 1987) and pollen show that the warmest part is present. Andersen et al. (1983) concluded that MIS 5e relative sea level was more than 15 to 20 MASL because they postulate an open sea-
connection across the island. Sejrup (1987) described paired shells of the boreal molluscs (*Lucinoma borealis* and *Pecten maximus*); these presently live-in water depths of 20 to 50 m and he postulated that sea level was this high during MIS 5e, however at lowest at the end of the interglacial. While we cannot determine a precise relative sea level at Bø, the results support the conclusion from Fjøsanger, namely that sea level in western Norway during the entire MIS 5e was at least 15 m higher than the current sea level.

**6.34    Hidalen, Svalbard, Norway (78.90, 28.13)**

The Hidalen site is in eastern Svalbard, on the island of Kongsøya. The marine limit from the last deglaciation is about 100 MASL (Salvigsen, 1981). The stratigraphy at Hidalen contains three coarsening-upward sequences from marine silt to littoral gravel, separated by till beds (Ingólfsson et al., 1995). The youngest coarsening-upward sequence (Unit F) is of Holocene age, whereas the two older have non-finite [14]C ages. The oldest (Unit B) obtained a TL age of 148 ka and
combined with the amino acid D/L ratio, Ingólfsson et al. (1995) considered Unit B to be of pre-Eemian age. Based on the amino acid D/L values they argued that the younger Unit D is of Eemian (MIS 5e) age. Mangerud et al. (1998) presented two alternative interpretations, the first being the one by Ingólfsson et al. (1995). However, they argued that it was more probable that the oldest Unit B is of Eemian age. Kongsøya is a national park with strong regulations, and no Quaternary geologist has later been allowed to go ashore to solve this problem. For the present interest, we state that MIS 5e marine sediments are
present on Kongsøya and they span ~50 to >60 MASL. If Unit D is indeed of MIS 5e age, then sea level was up to ~80 MASL. In our database, we have added both units, and note the controversy in age.





### 6.35 Kapp Ekholm, Svalbard, Norway (78.55, 16.55)

The Kapp Ekholm site (Fig. 5) is in central Svalbard, ~14 km outside the large Nordenskiöldbreen glacier, which occupies the head of Isfjorden-Billefjorden. This implies that if Kapp Ekholm was ice free, glaciers on Svalbard were not much larger

than at present. The marine limit during the last deglaciation was 90 m a.s.l. (Salvigsen, 1984) and this site was repeatedly covered by ice during the Quaternary. The site was first described by Lavrushin (1967, 1969) who found the subarctic mollusc *Mytilus edulis* within one of the marine formations (Fig. 6). *Mytilus* requires warmer climate on Svalbard than during the 20th century (Mangerud and Svendsen, 2018). Kapp Ekholm is the only pre-LGM site on Svalbard where *Mytilus* or other "warm-water" molluscs are found.


Several marine and glacial units are found in superposition at Kapp Ekholm (Fig. 5). The entire exposure is described in detail by Mangerud and Svendsen (1992); here we will only describe the formation containing *Mytilus* ('Formation B'), which is situated between 5 and 22 MASL. The lower ~5 m of Formation B contains thick marine mud with numerous floating stones (called 'marine diamicton'). The stones have probably rolled down the steep slope from the shore. Molluscs,

including thick *Mya truncata*, in living position, are common and paired shells of subarctic *Mytilus edulis* are also found in this unit (Fig. 6). The mud is overlain by thin and almost horizontal sand beds, capped by 12 m thick, steeply dipping gravel foresets. The latter are interpreted as formed by long-shore drift, and the top (22 MASL) therefore reflects sea level during the formation. The entire Formation B suggests a shallowing during deposition and thus sea level was considerably higher than 22 MASL when the mud and sand were deposited. Chronologically, Formation B is dated with five OSL dates that

yielded an average of 118 ka (Mangerud et al., 1998), suggesting an Eemian age (MIS 5e). Moreover, the frequent occurrence of the "warm-water" demanding *Mytilus edulis*, suggests that Formation B should be correlated with an interglacial in northern Europe when sea water was warmer than at present. Amino-acid D/L values also strongly suggest that Formation B cannot stem from an older interglacial.

### 6.36 Skilvika, Svalbard, Norway (77.57, 14.44)

In southwestern Svalbard, Skilvika is located on the shore of Bellsund, western Spitsbergen. The site was first described by Semevskij (1967), and later studied by several scientists. A detailed description was provided by Landvik et al. (1992). The entire Quaternary exposure is one km long and stretches up to 35 MASL. Most of the exposure consists of Formations 3 and 4, which are the candidates for MIS 5e age and together span 15 to 30 MASL (Fig. 7). Formation 3 is a coarsening upward sequence from glacimarine silt, through sand (shoreface) to cross-bedded gravel (foreshore). The overlying Formation 4

consists of a strange sediment, namely foresets of boulders and containing marine fossils in sand lenses between the up to 2-m large boulders. The foresets of Formation 4 interfinger with the beds in Formation 3 (Figs 7 and 8). The interpretation is that Formations 3 and 4 together represents a prograding beach showing a sea level about 30 m above the present (Landvik et al., 1992). Formation 4 was formed by an advance of the local glacier. There were found neither foraminifera (Lycke et





al., 1992) nor molluscs (Landvik et al., 1992) that require as warm water as at present and based on TL ages and amino acid
D/L values Landvik et al. (1992) concluded a MIS 5c age, which was also accepted by Mangerud et al. (1998), although with
the comment that "an Eemian age cannot be ruled out". Alexanderson and Landvik (2018) then performed an extensive
dating program, obtaining 20 OSL ages from Formation 3. The result was large spread in ages from 66 to 263 ka, although
when excluding these two outliers the range was 99±7 to 149±17 ka, with a mean of 119±5 (n=18). After another quality
screening they obtained 118±7 (n=10). The conclusion is that Formations 3 and 4 probably are of MIS 5e age, although
warm-water species are missing, and a MIS 5c age cannot be excluded. The relative sea level was about 30 MASL.

### 6.37    Kongsfjordhallet, Svalbard, Norway (79.03, 11.88)

The site Kongsfjordhallet is located on the north shore of Kongsfjorden, western Spitsbergen. Marine limit during the last
glaciation was about 40 m a.s.l. (Lehman and Forman, 1992). These exposures have been studied by several scientists,
apparently first by Boulton (1979). The oldest Quaternary marine units are probably about one million years old (Houmark-
Nielsen and Funder, 1999). In this review we mainly rely on Alexanderson et al. (2018). Relevant to the present study, units
4 and 5 are considered to be of MIS 5e age. The altitude of these units varies along the ~700 m studied section of the
coastline, but they are largely present from 27 to 34 MASL. The base of these sediments is ~1.5 m thick glacimarine mud.
This is overlain by up to three-meter-thick littoral sand and gravel, suggesting a shallowing due to glacial isostatic uplift. The
assumed MIS 5e age is based on two OSL dates with an average of 132 ± 7 ka, supported by a diverse foraminifera fauna
(Alexanderson et al., 2018). This age is also consistent with the chronology of the entire sequence. We find it probable that
these units are of MIS 5e age, although based on two OSL ages only. The relative sea level was about 34 MASL when the
littoral sands and gravels were deposited. However, the units are parts of an emergence cycle; thus, sea level was initially
higher than indicated by the littoral sediments.

### 6.38    Poolepynten, Svalbard, Norway (78.45, 11.66)

The Poolepynten site is located on the island Prins Karls Forland on the west coast of Spitsbergen. The marine limit from the
last deglaciation is about 40 MASL (Forman, 1990). The site was first described by Miller (1982) and later in more detail by
Bergsten et al. (1998), Andersson et al. (1999) and Alexanderson et al. (2013). The site has attracted much interest because
the oldest known bones of polar bear (*Ursus maritimus*) was discovered here (Ingólfsson and Wiig, 2009; Lindqvist et al.,
2010). In the present review, we mainly rely on Alexanderson et al. (2013). The exposure shows a sequence of several
marine and glacial units with ages 10-130 ka BP as dated with different geochronological methods. The lowermost unit
(named A1 in Alexanderson et al., 2013) is considered to be of MIS 5e age, and is the only unit discussed here. It is exposed
between 2 and 4 MASL and consists of thin sand beds with scattered pebbles, shell fragments and kelp, and it is interpreted
as shallow marine or sublittoral deposits. The polar bear jaw was found in this unit.



The MIS 5e age of unit A1 is based on several lines of evidence: the stratigraphical position, amino acid D/L values (Miller, 1982), slightly warm foraminifera fauna (Bergsten et al., 1998), the phylogenic position of the polar bear mandible (Lindqvist et al., 2010), and two robust OSL ages giving 118 ± 13 and 105 ± 10 ka (Alexanderson et al., 2013). We consider it very probable that Unit A1 is of MIS 5e age and that it indicates a relative sea level of 2 to 4 MASL. However, as with all other sites on Svalbard, the unit is interpreted as part of an emergence cycle caused by glacio-isostatic uplift. Thus, the

mentioned sea level does not show the MIS 5e high stand and possibly neither the low stand.

### 6.39    Leinstranda, Svalbard, Norway (78.88, 11.55)

The Leinstranda site is located on the eastern shore of Forlandsundet on the west coast of Spitsbergen. The site was discovered by Troitsky et al. (1979) and later studied by several scientists, in most detail by Miller et al. (1989) and Alexanderson et al. (2011b). A synthesis and critical review is given by Alexanderson et al. (2011a). The marine limit from

the last deglaciation is 46 m a.s.l. (Forman, 1990). The sequence consists of several formations of marine, beach and glacial sediments dated from almost 200 to 10 ka BP with different geochronological methods. The units considered to be of MIS 5e age are found between 16 and 19 MASL. The lower part consists of two-meter thick glaciomarine silt considered to reflect a sea level about 80 m higher than at present. The sediments are coarsening upwards to shore-face sand reflecting a sea level of 20 MASL at its formation. Almost all dated samples are collected from the shore-face sand. Marine mollusc shells, some

being paired, are found in all units. All species are common on Svalbard today.

Miller et al. (1989) found a foraminifera fauna that suggested slightly warmer conditions than at present and named the unit for the Leinstranda interglacial. Their amino-acid results suggested an Eemian age, which subsequently was supported by IRSL dating (Forman, 1999). We consider four recent OSL dates, with an average of 129 ± 10 ka, to provide the most

reliable age (Alexanderson et al., 2011a; Alexanderson et al., 2011b), but we appreciate that several methods agree on a MIS 5e age. The conclusion is that Leinstranda represents a MIS 5e site. The sea level was above 20 MASL when the dated bed was deposited and sea level was thus considerably higher before. Whether sea level later in MIS 5e dropped even lower due to glacial isostatic uplift is unknown.

### 6.40    Galtalækur site in Jökulhlaup valley, Iceland (63.99, -19.96)

Van Vliet-Lanoë et al. (2018) describe, in central southern Iceland, marine sediments exposed in deeply incised valleys (Rangá, Þjórsá and Hvítá rivers) that formed because of late glacial jökulhlaups (flooding events) from nearby glaciers. The Rangá Formation is around 30-m thick and records two separate marine transgressions following MIS 6 deglaciation of the region. The marine unit (represented by regional stratigraphic unit R-C2) consists largely of sands draped by thin silt sediment, representing a coastal fluvial setting, where a tidal delta formed in an estuary (Van Vliet-Lanoë et al., 2018). It is

best preserved at the Galtalækur site, where it is located at approximated 120 MASL (extracted from a stratigraphic plot; exact measurements at that site were not provided). The Rangá Formation sediments are assigned to MIS 5e based on



stratigraphic context, absolute dating and tephrachronology. From a stratigraphic standpoint, the Rangá Formation is overlain by extensive deglaciation deposits (tills, glaciofluvial and glacial marine sediments) that are dated to the Holocene via tephrachronology (presence of Vedde ash; Van Vliet-Lanoë et al., 2018). Contained in these marine sediments is a tephra
layer from the Grimsvötn volcano which is dated to MIS 5e based on its position in North Atlantic marine cores (Davies et al., 2014). Both underlying and overlying units that bracket the Rangá Formation have absolute ages. Van Vliet-Lanoë et al. (2018) presented three K-Ar ages on the glacio-volcanic unit that underlies the Rangá Formation that yielded ages between 155 ka and 129 ka. Two other ages from this unit (using the $^{39}Ar/^{40}Ar$ method) also yielded ages in this range (Clay et al., 2015; Flude et al., 2008).

**6.41    Region of Scoresby Sund, East Greenland (8 sites)**

Last Interglacial sites in the region of Scoresby Sund, eastern Greenland, were the subjects of intensive research in the early 1990s as part of the PONAM (Polar North Atlantic Margins; Late Cenozoic Evolution) project, which resulted in several papers published in the mid-1990s. A common indicator of LIG age is the presence of warmer-water than today fauna, e.g. *Balanus crenatus.* Present are also members of the Astarte genus, some arctic and some subarctic to their biogeography
(Funder et al., 2002). The presence of *Astarte borealis*, generally regarded as an arctic species, is, however, thought only possible in this region under conditions of increased advection of Atlantic water, which occurred only during the LIG and the Holocene (Vosgreau et al., 1994). Some sites are also constrained to the LIG based on >20 luminescence ages (Mejdahl and Funder, 1994). Below, we present 9 locations where these LIG sediments were documented in the vicinity of Scoresby Sund. The coordinates for these sites are estimated from maps.

**6.41.1    Kikiakajik section (70.038, -22.2467)**

At Kikiakajik there is a 700 m long and 10-15 m high coastal cliff, but it is partly covered by perennial snowbanks. Mangerud and Funder (1994) cleaned and described two exposures. The first exposure has a coarsening-upwards sequence from horizontally bedded marine silts at 3.5 MASL through sand to gravel foresets, which are cut by a till at 7.5 MASL. The second exposure consists only of the gravel foresets at 9 to 13 MASL, covered by till. The silt and sand contain a molluscs
fauna similar to the one described at Kap Hope, including *Balanus crenatus* and the warmer than today-indicating *Astarte borealis*. A single shell was $^{14}C$ dated to >44 ka BP, and the sand was dated with thermoluminescence to $132 \pm 10$ ka. The distinct fauna strongly suggests a correlation between Kap Hope and Kikiakajik, and the warm water requirements of the fauna together with the dates from both sites indicates a MIS 5e age.

**6.41.2    Kap Hope (70.459, -22.318)**

At Kap Hope there is a 20-m high coastal section where Mangerud and Funder (1994) described a marine silt between 2 and 3 MASL, directly overlying bedrock. The silt contains a rich *Astarte* fauna including the warmer-water species *Balanus crenatus* and *Astarte borealis,* which are not living in East Greenland today. The fauna suggests an offshore environment



warmer than at present. From the silt there is a coarsening up-wards sequence through sand (well-sorted, ripple laminations with gravel lenses) to gravel (includes well-rounded boulders), representing a regression from off-shore through shoreface to
beach environment at 10 m MASL. The beach sediments are covered by 5 m sandy gravel interpreted to suggest a rising relative sea level, eventually capped by a till. Two [14]C dates from individual shells gave non-finite ages (>42 ka) and two TL/OSL dates from the shoreface sand yielded 97 ± 10 ka and 75 ± 8 ka, respectively. Mangerud and Funder (1994) assigned the entire clay-sand-gravel sequence to the LIG based on the fauna and supported by the TL/OSL dates, although the latter yielded too young ages. In the database, we have added separate entries for both the regressive and transgressive
units.

### 6.41.3    Kap Stewart composite (70.44, -22.78)

Marine sediments assigned to the LIG are present from 0.5 to 40 MASL around Kap Stewart (Tveranger et al., 1994). In that study, workers examined several sites to better understand the sedimentology and to find as many fossils as possible. However, ultimately, all sedimentary structures, dates and fossils were assumed to represent one single site, which we follow
here. To provide additional context on the LIG marine event, below we provide a brief overview of each individual site.

At Loc. 471 marine sediments interpreted as a tidal, shallow marine delta environment were found between 9 and 12 MASL and at Loc. 473 similar sediments at 0.5 to 8 MASL (Tveranger et al., 1994). These sediments contain marine molluscs suggesting warmer-than-present water. Based on these molluscs and the stratigraphic position below diamicton and more
recent sediments, these deposits were assigned a LIG (MIS 5e) age. From Loc. 473 the interglacial unit could be mapped on the surface some 500 m inland (towards N) to Loc. 468 where the unit consists of tidally influenced channels and estuarine/lagunal deposits up to about 40 MASL. The unit was mapped further to Loc. 470 where it consists of trough cross-bedded sand and gravel located 30 to 38 MASL and interpreted as fluvial deposits with paleocurrents towards the fjord. The interpretation is that the transition from marine to a coarsening upwards fluvial sequence reflects a progradation of the paleo-
Ostraelv river delta. The vertical stacking of tidal channels and the thickness of fluvial deposits indicate that progradation took place during a relative sea-level rise up to about 40 MASL.

### 6.41.4    Hesteelv composite (70.44, -23.10)

Similar to the Kap Stewart site, in the Hesteelv area, workers examined several sites to better understand the stratigraphic record (Tveranger et al., 1994). Ultimately, all sedimentary structures, dates and fossils were assumed to represent one single
site, with the LIG marine unit situated between 5 and 35 MASL. Below, to provide additional context on the LIG marine event, we provide a brief overview of some individual sites.

At Hesteelv 13 outcrops, located in an area stretching five km along the fjord and one km inland, were cleaned and sedimentological logged in detail (Tveranger et al., 1994). Sediment units could several places be traced on the surface





between the outcrops and the correlations between the outcrops mentioned here are reliable. At Loc. 153 a continuous
outcrop was cleaned from 5 to 71 MASL and this is a key exposure for the full stratigraphy of the area but here we will only
describe the MIS 5e sequence (Unit A in Tveranger et al., 1994), which are found 5 to 35 m MASL. The lower 15 m consist
of 2 to 50-cm thick sandy turbidites interbedded with massive or weakly laminated mud layers containing molluscs,
dropstones and synsedimentary slumps. The turbidites are overlain by low-angel, cross-stratified sand, with layers of gravel,

and sand with climbing ripples and planar cross-bedding. Depositional direction was towards the fjord (i.e. south) throughout
the unit. The unit is interpreted as a prograding delta sequence with transition from offshore turbidites to shoreface deposits
with migrating bars. The altitude and facies indicate a sea-level 35 to 40 MASL. The interglacial sediments are overlain by a
till, interstadial marine sediments and another till and deglacial sediments, not described here. The MIS 5e unit can be
followed from Loc. 153 to Loc. 152 (18 to 30 MASL) and Loc. 154 (9 to 33 MASL) by surface mapping. Both localities

have a coarsening up-wards sequence like that at Loc. 153 and thus support the interpretation there. MIS 5e sediments were
described also from some of the other outcrops and fossils collected. At some places erosional, regressive facies are mapped
and interpreted to stem from the subsequent falling relative sea level. Several terrestrial plant macrofossils (e.g. *Betula
pubescens*), beetles and marine molluscs (e.g. *Mytilus edulis*) collected from the MIS 5e unit in different outcrops show
warmer-than-at-present climate, and mean July temperature was estimated to have been about 5 °C (Tveranger et al., 1994).

Thus, an interglacial origin is clear. The average of 9 OSL ages is 115 ± 29 ka and suggests a MIS 5e age, which is
consistent with the full stratigraphy at the site and indeed by correlation with other sites along Scoresby Sund (Vosgerau et
al., 1994).

### 6.41.5    Site 443d, Fynselv area (70.464, -23.320)

At Site 443d, marine sediments were identified between 12 and 21.5 MASL, dated by TL/OSL to 120 ± 10 ka, 121 ± 10 ka,

and 122 ± 10 ka (Hansen et al., 1994). The marine sediments were interpreted as a deltaic to shallow marine succession
(units Ib, Ic, Id) and are overlain by a till and marine sediments, representing a later incursion.

### 6.41.6    Langelandselv composite (70.546, -23.644)

Like the Kap Stewart and Hesteelv sites, workers in the 6.40.7 Langelandselv  region examined several sites to better
understand the stratigraphic record (Landvik et al., 1994). The LIG marine unit (amalgamated from several sites) spans 0 to

70 MASL. Below, to provide additional context on the LIG marine event, we provide a brief overview of some individual
sites.

At sites 77A and 77D, a layer of marine silts and sands are situated between 0 and 5 MASL (Landvik et al.,1994). OSL/TL
dating of the uppermost silt sediments yielded 122 ka and 121 ka (Mejdahl and Funder, 1994). Nearby, at sites 74 and 76A,

marine silts and sands were documented between 4 and 11 MASL. These marine sediments are overlain by ~10m of sand,





interpreted to be of fluvial origin. These were OSL/TL dated to 100 ka, 106 ka, 117 ka and 129 ka, the whole sediment succession suggesting an environmental change from marine conditions into a fluvial setting (Landvik et al.,1994).

At four sites in the Langelandselv region (sites 95, 96, 97 and 113), LIG marine sediments are situated at an anomalously high elevation. Landvik et al. (1994) suggested these marine sediments were deposited early during the LIG during isostatic recovery from the MIS 6 glaciation and were believed to represent an earlier interval than the other sites from the Scoresby Sund area, typically situated much closer to present-day sea level. At sites 95, 96 and 97, Landvik et al. (1994) described a sand unit containing ripple and herring-bone cross-beds, situated between 44 and 57 MASL. These sands were interpreted as a shallow marine environment and were OSL dated to 117 ka (Landvik et al., 1994). The mollusc assemblage in these sands suggests deposition during a warm-water interval, which is also indicative of the LIG (Vosgreau et al., 1994). Finally, at site 113, Landvik et al. (1994) described marine sand (Unit $A_0$) situated between 67 and 70 MASL.

### 6.41.7    Location 72, Aucellaelv River (70.585, -23.764)

On the north shore of Scoresby Sund, along the Aucellaelv River, are marine sediments at the base of the stratigraphic record which consist of mollusc-bearing (largely *Astarte borealis*) sand and silt, situated between 12 and 16 MASL, and interpreted as deposited in a sublittoral shallow marine environment (Unit 2; see Israelson et al., 1994). The sediments were OSL and TL dated to 122 ka and 144 ± 15 ka (sample R9110004). A broad LIG age is also suggested by the presence of warm-water mollusc fauna. The marine sediment unit is both underlain and overlain by till and on top is sediment from a later marine event.

### 6.41.8    Lollandselv-Falsterselv region, Greenland (70.873, -24.179)

Ingólfsson et al. (1994) noted LIG marine sediments at 23 stratigraphic sites between the Lollandselv and Falsterselv rivers. A composite diagram of these sites shows shell-bearing marine silts situated between 0 and 8 MASL in this region (Ingólfsson et al., 1994). The maximum elevation of marine sediments in this location is 20 MASL. The authors interpreted these marine silts as deposited during the LIG, representing a transition from glaciomarine to tidal and eventually a littoral depositional setting. Macrofossils (plants, molluscs) suggest temperatures warmer than the Holocene. These LIG marine sediments are overlain by a series of cross-bedded sands (fluvial), till (ice advance), topped by sediments suggesting a later marine incursion.

### 6.42    Thule, Western Greenland (7 sites)

Kelly et al., (1999) described three separate marine events in sediment sequences from western Greenland. The middle marine sediment succession (known as the Qarmat event) contains mollusc shells from *Mya truncata*, *Hiatella arctica* and *Chlamys islandica*. The age of these marine sediments is assigned to the LIG, based on TL ages between 91 to 154 ka (mean age at 127 ka). The Qarmat sediments coarsen upwards, and the uppermost sediments have wave influenced sedimentation.





We outline 8 sites below, as described by Kelly et al. (1999). Some of the chronological data were described in Sejrup (1990).

### 6.42.1   Iterlak K (76.710, -69.410)

At this site, sands and muds are situated from 25 to 27.5 MASL. Marine shells are present, and two amino acid dates suggest deposition during the LIG, which is supported by a non-finite radiocarbon age.

### 6.42.2   Iterlak L (76.713, -69.417)

At this site, sands and muds are situated from 11 to 14 MASL. No marine shells or foraminifera were noted, but a TL age suggests deposition at 118 ka.

### 6.42.3   Saunders Ø B (76.597, -69.736)

At this site, the Qarmat marine event is situated between 3 to 20 MASL. Glaciomarine muds are present from 3 to 6 MASL. Overlying the glaciomarine sediments is a ~1-m interval of sands, followed by clast-supported gravels with varying amounts of mud and sand, which was interpreted as a beach deposit. Marine shells are present. Three TL ages from the lower-most sediments yielded 154 ka, 153 ka and 119 ka.

### 6.42.4   Saunders Ø C (76.597, -69.736)

At this site, the marine interval spans 5 to 12 MASL. The lower part of the section consists largely of sands which gradually coarsens to sandy gravels at ~8 MASL, which was interpreted as a beach deposit. Marine shells are present and the uppermost sediments, at ~12 MASL, were dated with a TL to 91 ka.

### 6.42.5   Narsaarsuk D (76.453, -69.287)

Sand with occasional coarse gravel beds are situated from 0 to 6 MASL. The sediments have signs of being wave influenced. Marine shells and foraminifera are present, and amino acid dating of shells suggest deposition during the LIG.

### 6.42.6   Narsaarsuk E (76.453, -69.287)

Clay and silt, transitioning to sands are situated from 11 to 17 MASL. Marine shells and foraminifera are present. A TL age from the upper sediments suggests deposition at 119 ka, which is supported by amino acid dating suggesting a LIG 990   timeframe, as do two non-finite radiocarbon ages.



### 6.42.7 Narsaarsuk F+G (76.453, -69.287)

At this site, sand with occasional coarse gravel beds is situated from 3 to 9 MASL. Marine shells, foraminifera and plant matter are present. Amino acid dating suggests deposition of these sediments broadly within the LIG, which is supported by a non-finite radiocarbon age as well as a TL age of 123 ka.

### 6.43 Iles de la Madeleine, Quebec, Canada (3 sites)

In eastern Canada, the Iles de la Madeleine contains sediment exposures bearing an extensive stratigraphic record that details the interplay between sea level and Pleistocene ice sheets. Marine sediments assigned to MIS 5e have been described at three sites, shortly described below.

### 6.43.1 Camping site (47.346, - 61.880)

Rémillard et al. (2017) described marine sediments along the cliffs of the Iles de la Madeleine that overlie local bedrock. These sediments consist of fine to medium red sand with gravel beds, moderately to poorly-sorted, and 10 to 30 cm in thickness. This unit was located at 14 MASL with a maximum thickness of 4 m. The sediments were interpreted as a barrier beach sediment displaying alternating low and high energy fluctuations and were assigned to the early part of MIS 5 (likely 5e), based on an IRSL age of 115±8 ka (OSL51; Rémillard et al., 2017). The

unit was originally interpreted as being marine limiting, which we have kept, despite a beach barrier being more strictly a terrestrial limiting deposit.

### 6.43.2 Portage du Cap (47.2458, -61.9058)

The Portage du Cap site was first described by Prest et al. (1976) and subsequently by Dredge et al. (1992). At this site, a sequence of sub-till silts, pebbles and gravel are exposed within a gravel pit. The entire marine sequence spans 14 to 17 MASL (Dredge et al., 1992). Underlying the entire marine sequence is sandstone bedrock containing borings from *Ziphraea* sp., a species that lives in the modern intertidal zone in the region. The lowest unit, situated at 14 MASL, is a grey marine silt, rich in dinoflagellate cysts, containing a pollen, beetle and diatom assemblage suggesting conditions warmer than present-day in the region (Dredge et al., 1992; Prest et al., 1976). Within this unit are clast supported beach gravels with well-rounded pebbles. Overlying the gravel unit is a 1 to 2 m thick organic sand unit with a woody organic horizon. For this reason, the lowermost section of this stratigraphic sequence has been assigned to a MIS 5e marine incursion that immediately followed the removal of MIS 6 ice from this area. On top of this unit is a gravel unit with a pollen assemblage similar to present-day, likely the result of relative climatic cooling following peak MIS 5e conditions, paired with local sea level rise and beach formation (Dredge et al., 1992). Three radiocarbon ages are available at this site. Plant detritus and wood have been dated to >38 ka and >35 ka BP (GSC-2313 and BGS-259; Prest et al., 1976) and a marine shell was dated to 42.9





± 0.72 ka BP (GSC-4633 HP; McNeely and McCuaig, 1991). The finite age on the marine shell is possibly beyond the reliability limit for dating marine shells (Pigati, 2002), and all radiocarbon ages considered minimum such.

### 6.43.3  Le Bassin site (47.2333, -61.8983)

Dredge et al. (1992) described a stratigraphic section of sub-till peat overlain by clays, sands and gravels present at the Bassin site, Iles de la Madeleine, Canada. At the base of the section overlying bedrock is 0.5 m sand interval, followed by a

0.2 m peat layer. This was interpreted as nearshore fluvial and lacustrine environments. Overlying the peat layer is 0.9 m silt and clay unit, followed by 0.1 m sand, both interpreted as marine in origin based on the presence of warm-water oyster shells (*Ostrea virginica*). The altitude of these marine sediments is unclear; we assume the base of the stratigraphic section is 0 MASL ('just below tide level' according to Dredge et al., 1992), which would situate the marine silt, clay and sands between 0.7 and 1.7 MASL. Pollen data from these sediments suggest conditions similar to present-day (Mott and Grant, 1985). The

sub-till sediments suggest that, following deglaciation of this region, the local area became a peatland and, as sea levels rose, it transitioned to a lagoon environment, beach ridge and coastal barrier (Dredge et al., 1992). Three U/Th ages on these sediments range from 89 ka to 106 ka (UQT-182, UQT-183, UQT-184). Two diamictons (interpreted as tills deposited during ice advances) truncate the sand unit. Based on pollen and stratigraphic position, Dredge et al. (1992) suggested an MIS 5e age assignment (Dredge et al., 1992); the U-Th ages are interpreted as minimum age constraints by the original

authors.

### 6.44  Clyde Foreland, Baffin Island, Canada (3 sites)

First described by Løken (1966) and Feyling-Hanssen (1967), sub-till mollusc-bearing marine sands are present along cliffs of the Clyde Foreland, Baffin Island. These are known as the Cape Christian marine sediments and several radiocarbon dating attempts on organic material from this unit yielded infinite ages or finite ages (e.g. QL-188) that can be considered as

minimum ages. Feyling-Hanssen (1967) suggested that these sediments were deposited during MIS 5e owing to its content of relatively warm-water-indicating foraminifera (*Cassidulina teretis*). Warm conditions were confirmed via pollen analyses. Later, these sites were re-visited by Miller et al. (1977) who reported and a [230]Th age of 130 ka (Miller et al., 1977). Below, we document 3 of these sites, as described by Miller et al. (1977). These sites are correlated to each other on the basis of stratigraphic position, and amino acid ratios.

### 6.44.1  Profile 6 (70.694, -68.946)

At this site, the Cape Christian marine sediments are situated between 27 and 28 MASL. They are underlain by a series of sands and tills and overlain by a thin layer of buried soil along with till from later Pleistocene ice advance.



### 6.44.2   Profile 9 (70.596, -68.413)

At this site, Cape Christian marine sediments (coarse sand and cobbles) are found between 9 and 11 MASL. These marine
sediments are overlain by two tills and younger marine sediments. This is the type section for the Cape Christine marine
sediments.

### 6.44.3   Profile 10 (70.575, -68.347)

Miller et al. (1977) documented coarse sands and gravels between 1.8 MASL and 2.5 MASL, which they correlate to the
LIG Cape Christian marine sediments.

### 6.45   Ile aux Coudres, Quebec, Canada (47.4089, -70.4161)

Occhietti et al. (1995) described a core taken on Ile aux Coudres, located in the Saint Lawrence Estuary, Canada, that
showed a succession of Pleistocene-aged deposits. A till was noted at the base of this record (known as the Baie-Saint-Paul
glacial complex), followed by a series of clays, rhythmites and deltaic sediments, known as the "Guettard Sea" sediments,
present from 125 to 2 MBSL. Occhietti et al. (1995) described, at the base of this stratigraphic unit, a very compact, finely
stratified grey clay between 125 and 102 MBSL. A dinoflagellate cyst at 119 MBSL suggests a marine origin for these
sediments, and the authors interpreted it as a transgressive prodeltaic (deep marine) deposit. Next, between 102 and 71
MBSL, are grey rhythmites, hypothesized to be a high-level prodeltaic deposit resulting from the gradual shallowing of
marine waters. Lastly, the sediment core shows silts and sandy silts between 71 and 2 MBSL, interpreted as prodeltaic,
transitioning to deltaic sediments. This uppermost unit contains benthic foraminifera common to brackish marine water.
Based on stratigraphic context (i.e. heavily isostatic depression following large-scale glaciation), Occhietti et al. (1995)
aligned this record with the transition period between MIS 6 and MIS 5e. They infered a large-scale, long-lasting glaciation
followed by ice recession, rapid marine inundation into the isostatically depressed landscape, and shallowing of marine
waters owing to subsequent rebound. This accumulation of marine sediments may represent ~3500 years of deposition
(Occhietti et al., 1995).  Since the age of the deposit is based on regional correlations and environmental conditions from
pollen assemblage composition, rather than direct dating, we assign the lowest age quality score.

### 6.46   Bridgehampton, New York, United States (40.9374, -72.3064)

Marine sediments, typically sandy clay that is brown or grey/green in colour, were first encountered at depth on Long Island,
New York in the early 1900s (Fuller, 1914). The so-called "Gardiners Clay" is present at depth in several borehole records
between 20 and 45 MBSL in various well records in this region (Nemickas and Koszalka, 1982). Paleontological work
revealed a variety of warm-water fauna, including foraminifera, coelenterata, bryozoa and mollusca (Gustavson, 1972),
confirming a marine origin for these sediments. At the Bridgehampton site, the Gardiners Clay was encountered between 23
and 46 MBSL (Nemickas and Koszalka, 1982).





The Gardiners Clay is provisionally assigned to MIS 5e, based on stratigraphic context and amino acid racemization. Stratigraphically, this marine unit is overlain by a series of Pleistocene deposits, notably the Montauk till, associated with the most recent (MIS 2) Laurentide Ice Sheet maximum extent (Nemickas and Koszalka, 1982). Moreover, at some sites, the Gardiners Clay is underlain by a gravel deposit (Jameco Gravel; Fuller, 1914) that has been associated with fluvial or glaciofluvial deposition immediately following retreat of the MIS 6 ice sheet and prior to the marine incursion. The interpretation of the age is complicated by the fact that no amino acid racemization had been conducted on shells from the Bridgehampton site. However, shells contained in the Gardiners Clay from a nearby outcrop (40002; the stratigraphy of which was not described in detail) yielded an age assignment of early MIS 5 (Wehmiller et al., 1988; Wehmiller and Pellerito, 2015). The significant difference in elevation and depositional context (at-depth in a drill core vs. sub-aerial outcrop) implies the Gardiners Clay might represent two marine deposits of different ages.

### 6.47    Kwataboahegan River, Ontario, Canada (51.1389, -82.1167)

The Hudson Bay Lowlands contain a rich stratigraphic record consisting of tills interspaced by non-glacial sediments. Along the Kwataboahegan River, *in situ* marine sediments were first discovered underlying till by Bell (1904), and later described in detail by Skinner (1973). These so-called "Bell Sea" sediments were first reported to be at 75 MASL (Skinner, 1973), however subsequent geochronological work on samples originally obtained by Skinner (1973) report the elevation as 90 MASL (McNeely, 2002). The reason for this discrepancy is unknown. These marine sediments consist of 0.8 m compact bluish grey sand and silt that are deformed owing to subsequent Wisconsinan ice advance over the region. An "undulating bed of marine mollusc shells" is present near the mid-point of the sediment package (Skinner, 1973).

The Bell Sea sediments are assigned to MIS 5e based on stratigraphic context, amino acid dating and minimum limiting radiocarbon ages. Stratigraphically, this marine deposit is overlain by tills that are assigned to the Wisconsin Glaciation, thus it preceded the build-up of ice during the Wisconsinan. Amino acid techniques also support a MIS 5e assignment for the Bell Sea sediments; Andrews et al. (1983) show that isoleucine epimerization ratios from shells in the Bell Sea sediments (average of ~0.2) are significantly older than ratios from assumed mid-Wisconsinan marine deposits in this region (average of ~0.14), which are even older than shells from the post-LGM marine incursion (average of ~0.03). In calculating the amino acid dates, the average regional diagenetic temperature (0.6°C) can only be reconciled with the isoleucine epimerization ratios if the smallest isoleucine epimerization ratio (Bell Sea sediments), is assigned to MIS 5e (Andrews et al., 1983). For that reason, the Bell Sea sediments were assigned to ~130 ka (Andrews et al., 1983). Finally, two radiocarbon ages on marine shells (*Hiatella arctica*) from the Bell Sea deposit yielded >37 ka (GSC-1475; Blake, 1988) and 47.85 ± 1.09 ka (TO-2503; McNeely, 2002). The finite age is unreliable because shells samples are highly susceptible to modern-day contamination, the date is very close to the limit of radiocarbon dating (McNeely, 2002). Thus, both are considered as minimum limiting radiocarbon ages.





### 6.48    East of Nicholson Peninsula, Northwest Territories, Canada (69.894, -128.521)

Along the westernmost coastline of the Northwest Territories, Canada, sediments representing the Liverpool Bay Interglaciation (MIS 5e; LIG) are exposed as part of the Ikpisugyuk Formation. As described by Rampton (1988), a layer of organic-bearing marine sand and silt is present between 0 and 2 MASL (Locality VH-83-050), which was interpreted as an intertidal beach complex. Fossils from a nearby exposure of the Ikpisugyuk Formation (VH-83-040; does not contain a marine unit) suggest a climate warmer-than-present, which supports LIG deposition (Rampton, 1988). Radiocarbon age attempts on driftwood contained in this unit were non-finite (GSC-3722) and amino acid ratios 0.1 to 0.15, both supporting a LIG age assignment

## 7    OTHER LIG MARINE SITES

During our search for MIS 5e/LIG marine sites, we located several sites containing marine sediments that are not *in situ*, mostly due to post-depositional, i.e. later, glacial deformation/relocation. These sites are thus unsuitable as indicators of RSL but merit inclusion in this manuscript. We describe ten of these sites below, ordered east to west.

### 7.1    Lower Ob sites, West Siberian Plain, Russia

As shown in Fig. 3, the stratigraphic record present at the Lower Ob sites, West Siberian Plain, contains a well-dated MIS 5e section consisting of alluvium with OSL ages of 125 to 138 ka and thick peat with U/Th ages 133 and 141 ka. These terrestrial sediments occur at the same altitudes or slightly higher than the corresponding marine sediments of the Arctic. However, marine sediments at the base of this stratigraphic sequence (separated from the overlying terrestrial beds by a till) yielded a youngest age of $153 \pm 15$ (sand with rich boreal foraminifers of the Kazantsevo assemblage in Hashgort borehole at 65.42, 65.67; Arkhipov et al., 1992). This marine unit possibly represents an earlier marine incursion and is therefore excluded from our database.

### 7.2    More-Yu, Pechora Lowland, Russia

At the More-Yu site, Pechora Lowland (67.867, 60.183), OSL ages of 112 ka and 120 ka were obtained from large sand blocks with shells of boreal molluscs included into the glacially deformed diamict sequence (Fig. 3). The uranium/thorium date of 130±8 ka from this section on boreal *Arctica islandica* shell confirms an Eemian age of the detached marine sand and a Weichselian age of the encompassing till with fossil glacial ice (Astakhov and Svendsen, 2002). We exclude this site from our database owing to the likely glacial translocation of the marine unit.



### 7.3 Malaya Kachkovka, Kola Peninsula, Russia

The Malaya Kachkovka site is in the tributary valley of the Malaya Kachkovka River on the eastern coast of the Kola Peninsula, Russia (67.40, 40.90; 140 MASL). The entire sediment exposure is ~10 m thick (Gudina and Yevzerov, 1973). Mollusc fauna, foraminifera and pollen content of the marine sediment interval (127.5 to 134 MASL) were studied by Lavrova (1960) and Gudina and Yevzerov (1973). The marine part of the section comprises 0.5 m thick clay unit at the base of the exposure overlain by mollusc-bearing sands. Lavrova (1960) considered that the mollusc fauna in the marine sequence represents the upper sublittoral zone, and according to Gudina and Yevzerov (1973) the faunal assemblage represents the warmest fauna of the interglacial marine sediment faunas discovered on the Kola Peninsula. 230Th/234U ages on *Astarte borealis* shells from coarse and medium marine sands at around 132 MASL yielded ages of 102±4 ka and 114±4 ka (Arslanov et al., 1981) suggesting that the sediments formed during the Eemian interglacial (MIS 5e). However, Gudina and Yevzerov (1973) pointed out that the characteristics of faunal assemblages and the high altitude of the marine interglacial sediments might suggest that the sediments at the Malaya Kachkovka site do not represent the Eemian interglacial sediments but rather an older interglacial. Even if that was not the case, the marine sequence is overlain by glacial till and therefore, it is probable that the marine sequence is not *in situ* and has been glacially thrust as a block to its present high-altitude position.

### 7.4 Ludyanoi, Kola Peninsula, Russia

Grave et al. (1969) described a riverbank section of the Ludyanoi creek (66.33, 39.92; 70 MASL), a small tributary to Pulonga River, southeastern costal area of the Kola Peninsula. They described a sediment exposure where the top 10.7 m of section is glaciofluvial sands and gravels overlying 1.5 m thick till. Below the till is a 3.7-m thick greenish grey marine clay with shells and shell fragments recognised as *Astarte borealis, A. elliptica, A. crenata and A. montaqui*, of which the latter two are considered subarctic to their biogeography. The top of the marine clay unit is at around 60 MSAL, but it is not precisely known if this marine unit is in situ. It is therefore excluded from our database.

### 7.5 Lovozero, Kola Peninsula, Russia

There is a thick sediment sequence at the Lovozero site located in the inner part of the Kola Peninsula, Russia (67.08, 38.970; 210 MASL). The lower part of the drill hole sections contains a 44-m thick sand and silt unit between 140 and 184 MASL. A proportion of the diatom flora in the lower part of the sediment core is marine and the pollen taxa is dominated with pine and birch (Ikonen and Ekman 2001). However, it is generally thought that diatoms and pollen are mostly reworked in this lower sand-rich sediment unit and therefore, the correlation of this unit to Eemian interglacial (MIS 5e) is not possible and the position of the sea level cannot be reconstructed (e. g. Ikonen and Ekman, 2001). We do not include the Lovozero site in our database.



### 7.6 Evijärvi, Finland

The Evijärvi site is situated in central Ostrobothnia, western Finland (63.434, 23.322; 67 MASL). At this site, the LIG sediments occur on the proximal flank of a drumlin (Eriksson et al., 1980). In the top 6.4 m of the borehole record, several till and sand beds occur. The interglacial sediments consist of silt (9.0 to 9.5 m depth) and gyttja (6.4. to 7.4 m depth) interbedded with till. Samples for pollen and diatoms were analysed from borehole sediments (Grönlund, 1991; Eriksson, 1993). The pollen spectra of the silt layer (9.0 to 9.5 m depth) and gyttja layer (6.4. to 7.4 m depth) indicate only one local pollen assemblage zone consisting of *Betula*, *Alnus*, *Picea* and *Corylus,* which is correlated to the Eemian interglacial (Erikson, 1993). Diatoms in the silt bed (9.0 to 9.5 depth) above the till at the base of the core are exclusively marine while the diatom taxa in the gyttja is dominated with marine lagoonal types (Grönlund, 1991a). We do not include the Evijärvi LIG site in our database because these marine sediments are not at their original position of deposition but have been transported for an unknown distance and altitude by ice (e.g. Eriksson, 1993).

### 7.7 Norinkylä, Finland

The Norinkylä site is situated in southern Ostrobothnia, western Finland (62.58, 22.020, 114 MASL). At this site, on the flanks of a till covered esker, sediments suggested to be Eemian consist of glacially deformed gyttja, organic-rich silt and sand (Niemelä and Tynni, 1979; Donner, 1988). Boreholes made in the Rahkaneva mire next to the till-covered esker display silt, clay and gyttja in between two till beds with a maximum thickness of 4.5 m. The inter-till sediments have been studied for their lithology and pollen and diatom content (Grönlund, 1991b; Erikson, 1993). The pollen assemblage shows a succession typical for the Eemian interglacial in western Finland (Eriksson, 1993). Fresh-water diatom taxa dominate the lowermost part of the inter-till sediments (biostratigraphically belonging to the early Eemian Betula regional zone; Grönlund, 1991b) while there is a transition to marine diatom taxa that takes place at around 97 MASL. Saline diatom taxa dominate the interglacial sediment between 97 and 102 MASL (Grönlund, 1991 a, b). We do not include the Norinkylä LIG sediments in our database because these sediments are glaciotectonised and have been transported an unknown distance from their original place of deposition.

### 7.8 Svartenhuk Halvø, west-central Greenland

Kelly (1986) and Bennike et al (1994) describe a raised marine deposit in west-central Greenland, which was correlated to the LIG based on AAR. They found no evidence of a Holocene highstand in this location. We were unable to add this site to the database, as there was no information on the elevation, and limited information on the geology.

### 7.9 Nantucket Island, Massachusetts, United States

Marine sediments, bracketed by two tills, on Nantucket Island, Massachusetts, United States, were first described in the 1800s (Desor and Cabot, 1849; Verrill, 1875). These marine sediments are located between 0 to 20 MASL (Oldale et al.,



1982). These sediments consist of a gravel base, overlain by several meters of cross-bedded so-called "Sankaty Sands" that contain stratified silt and clay along with abundant marine shells. Oldale et al. (1982) assigned the Sankaty Sands to MIS 5e based on stratigraphic context, palaeoenvironmental data, absolute chronology (U-Th, minimum radiocarbon ages) and relative age determinations (amino acid racemization). The lowermost till was assigned to MIS 6 and the upper till to MIS 2

ice advances, and the Sankaty Sands were stratigraphically constrained to the time interval immediately following retreat of the MIS 6 ice sheet. Palaeoenvironmentally, well-preserved warm-water oyster shells (*Crassostrea virginica*) and clam shells (*Mercenaria mercenaria*) are present at the base of this marine unit, and cooler-water clams (*Mercenaria campechiensis*) and mussels (*Mytilus edulis*) are present near the top (Verrill, 1875). These marine shells capture the transition from warm-water MIS 5e conditions to more temperate MIS 5d conditions (Oldale et al., 1982). In terms of

absolute chronology, corals located within the Sankaty Sands were dated via Uranium-series methods to 133±7 ka, which the authors believed to be a maximum age for this deposit (SHIO/80-9; Oldale et al., 1982). Moreover, seven radiocarbon ages were obtained from wood and shells located at this site. Four yielded non-finite ages and three were finite. However, the latter were suspected to be affected by modern carbon contamination (Oldale et al., 1982). Finally, amino acid racemization analyses (isoleucine epimerization ratios) of 10 shells from this marine unit suggest deposition between 140 ka and 120 ka

(Oldale et al., 1982). In summary, the palaeoenvironmental data, stratigraphic context and available age assignments all suggest a MIS 5e age for the Sankaty Sands. Despite the clear evidence of MIS 5e age assignment, the section is glaciotectonized, and is therefore unreliable as an estimate of palaeo-sea level.

### 7.10  Western Banks Island, Canada

Lakeman and England (2014) describe a glaciotectonized marine deposit on Phillips Island, located off the west coast of

1215 Banks Island, which dated to the LIG based on OSL dates and non-finite radiocarbon dates. The exposure has interbedded sand, silt and clay with abundant mollusc fossils. The sample was taken at 7 MASL. Several other raised in-situ marine deposits were sampled in western Banks Island but returned finite ages that would place the age of those deposits to MIS 3. They cautioned that the results from the Phillips Island OSL dates means that those radiocarbon dates may be minimum ages. If regarded in this way, these sites may be from the LIG, but the ambiguity means we have not added these sites to the

1220 database.

### 8  DISCUSSION

This manuscript brings together 82 MIS 5e/LIG sea level proxies from the glaciated Northern Hemisphere. In general, the sites reported in this paper do not offer constraint on the MIS 5e high stand, but rather any evidence of sea level position after the end of the MIS 6 glaciation. The motivation behind some of the research described herein was to constrain the

1225 extent of the Barents Sea, and the (possible) connection between the White Sea and the Baltic Sea caused by the glacio-isostatic depression of western Russia during the MIS 6 glaciation and into the LIG (Ikonen and Ekman, 1991; Funder et al.,



2002; Miettinen et al., 2014). Last Interglacial sites in Svalbard, Iceland and Greenland are largely located along the modern-day coastline. Comparatively fewer LIG sites are present in North America, likely owing to more erosion or their position in only borehole records.

## 8.1    Uncertainty and data quality

The primary challenge of assessing past sea level in formerly glaciated areas in that the sea level position likely did not remain at a fixed position for any length of time. The dating techniques that can be applied to LIG deposits lack the precision to determine when sea level was at a specific elevation (with the possible exception of the correlation with European pollen records, if such long-distance correlations are valid). Therefore, we chose to describe the proxies in terms of how much information they give to show sea level variations in the LIG. Most of the studies we looked at lack information on how elevation was determined. We were able to include this information in the database for sites that involved the authors of this study. Since most of the sites described here were overriden by ice sheets during the last glacial period, most of the sections are incomplete due to erosion. This means that it is usually not possible to determine is subaerial exposure happened.

Last Interglacial marine sites in the Northern Hemisphere have poor or average chronological constraint (quality scores 2, 3 or 4, see Table 5 and Fig. 9), due to conflicting geochronological data, a dependence on chronostratigraphic inference, large dating uncertainties and a fragmented stratigraphic record. Higher scores in WALIS are generally only assigned for U/Th dated corals that can have precision of less than a few thousand years, which is not the case in the techniques applied to glaciated region sea level proxies. Despite this, based on stratigraphy and environmental proxies, there is reasonable confidence that most of the proxies documented in our database fall either in the LIG, the latest part of MIS 6 when the ice sheets were retreating, or during the early parts of MIS 5d when the ice sheets began to grow again. Without the environmental indicators, it can be difficult to distinguish deposits that might instead date to the sea level highstand that happened after the MIS 5d glaciation (e.g. Mangerud et al., 1998). For the sites in North America, a high rating assignment from environmental conditions alone is not applied, as the MIS 5e stratigraphy is not as well established as in Europe (Otvos, 2015). The quality of RSL data is more varied, with some sites ranking very low (e.g. Kwataboahegan and Pyoza River sites), and others ranking very high (e.g. Fjøsanger and Bø sites).

## 8.2    Pre- and post-LIG sea level oscillations

Northern Hemisphere ice sheets waxed and waned (Batchelor et al., 2019) many times during the Quaternary, allowing for marine incursions prior to the LIG (e.g. during MIS 7) and during later intervals (e.g., during MIS 5c, MIS 5a, MIS 3 and the Holocene), with maximum high-stands preferentially during the transitional stages between glacials and following interglacials/interstadials. However, given difficulties inherent to dating sediments beyond the range of radiocarbon dating, chronological constraints remain a key challenge for these deposits. Here, we highlight some locations where pre-LIG and post-LIG sea level oscillations are recorded. This list is not exhaustive; rather, it is meant to provide examples of other sea



level oscillations that are preserved in the stratigraphic record, and to provide context for the difficulties in assigning ages to
marine deposits in the glaciated Northern Hemisphere.

Several pre-LIG marine units are known at Kongsfjordhallet, Svalbard (Houmark-Nielsen and Funder, 1999), although not
described in the present paper. The oldest probably as much as 1 million years old. Marine sediments dating to MIS 7 have
been documented at several sites in Russia. At the Bol'shaya Kheta River sections (67.97, 83.10), in addition to LIG marine
sediments (see Section 6.9), a lower marine bed contains shells of the extinct mollusk *Cyrtodaria jenisseae* Sachs (today
generally considered as *Cyrtodaria angusta*, see e.g. Möller et al., 2019a) and OSL ages at 225 ± 16, 226 ± 21, 162 ± 23, 231
± 17 ka indicate a MIS 7 age (Nazarov et al., 2020; Astakhov and Semionova, 2021). A pre-LIG age is also suggested for
numerous sites on the Taimyr Peninsula. Kind and Leonov (1982) claimed that >60 outcrops on central Taimyr Peninsula,
south of Lake Taimyr, expose MIS 5e-aged marine sediments (Kazantsevo in their terminology). In the Bol´shaya Balakhnya
river valley (73.5, 104.6) such sediment should reach 30 to 40 MASL, while further to the NE, in the Bol´shaya Rassokha
river valley (73.8, 107.7), such sediment should reach 60 to 70 MASL. However, as demonstrated in Möller et al. (2019a)
from a large number of sections along the Bol´shaya Balakhnya River, these sediments are of an older age as suggested from
numerous ESR and OSL dates (Möller et al., 2019a, 2019b). The sediments that Kind and Leonov (1982) presumed MIS 5e-
aged are often overlain by a till; as the sediment sections are south of the Severokokora Ice Marginal Zone (IMZ), marking
the southern maximum boundary of the MIS 5d (Early Zyryanka) glaciation over Taimyr (Möller et al., 2019a), this till must
postdate the MIS 5d glaciation (MIS 6 or older) and the marine sediments are consequently of a pre-MIS 5e age.

Post-LIG marine sediments are far more commonly preserved in the stratigraphic record. For example, at the Chapoma site
on the Kola Peninsula, in addition to an LIG marine sediments (see Section 6.16), an upper marine silt is present between 11
and 13 MASL (Korsakova, 2019). Two ESR dates on shells at around 12 MASL in this marine unit yielded ages of 85.5 ±
3.2 ka and 86.0 ± 3.9 ka (Arslanov 1981), placing it into MIS 5a. Multiple sea level oscillations are also present at the Lower
Agapa River sections (Gudina et al., 1968) as well as the Strelna River site (Section 6.17) where a sequence of upper marine
sediments were dated via IRSL and ESR to between 102 ka and 84 ka (Korsakova et al., 2004; Korsakova, 2019). Möller et
al. (2015, 2019a) describe several marine sequences in central and southern parts of the Taimyr Peninsula, Russia. All these
marine sediment successions were dated with ESR and OSL; with six outliers excluded the mean age is ~86 ka (*n* = 62) with
an age scatter of ± 15 ka, an age that firmly put these sediments in an Early Zyryanka deglaciation (MIS 5c–b). Finally,
marine sediments at the Anjeliko River site (77.368, 102.730) have been constrained to an ESR age cluster of 80 to 93 ka
(mean age 86 ka), suggesting deposition at marine inundation during MIS 5a following a Kara Sea Ice Sheet glaciation
during MIS 5d–5c.

Marine sediments dating to MIS 3 have been described in the glaciated region of Russia and Europe. For example, at the
Ozernaya River on October Revolution Island, offshore of Taimyr Peninsula (section 6.6), Marine unit IV sediments were





exposed in two sections on opposite sides of the river valley, ~2.5 km apart (Oz 4 and Oz 5; Möller et al., 2007). These sediments contained a rhythmical sedimentation pattern, several *in situ*-positioned paired mollusc shells and almost complete

skeletons of at least 9 narwhals (*Monodon Monoceros*) that were in the process of eroding out at the top of the sediment succession. The Marine IV sediments were interpreted as formed in a marine setting at a water level >65 MASL within a deeply embayed estuary behind a valley-mouth bar/barrier system in the Ozernaya valley (Möller et al., 2007). Radiocarbon ages on a narwhal tusk and molluscs yielded ages of ~50, 45.4 and 46.8 cal. ka BP ka (all at the upper limit of radiocarbon dating), ages supported by a single mollusc shell ESR dated to 59 ka BP. The dating of Marine IV sediments thus suggests

that they are of a MIS 3 age, formed at a marine inundation of the island that probably was glaciated during the whole time span of MIS 5d to MIS 4.

Marine sediments dated to between MIS 5e and MIS 2 are widespread in Svalbard (Mangerud et al., 1998; Alexanderson et al., 2018). At Kapp Ekholm there are in fact two pre-Holocene marine units situated stratigraphically above the LIG beds,

one considered to be of a MIS 5c age and the younger of MIS 3 age. Beds of basal till are found between these marine units.

Marine sediments dating to MIS 3 are uncommon in North America and are often less reliably constrained than similar sites in Russia and Europe. Shallow marine and beach sediments in Eastern Canada constrained to MIS 3, using OSL and radiocarbon dating, are situated at 30, 37 and ~15 MASL (Rémillard et al., 2016, 2017). Also, in the Saint Lawrence

Lowlands, marine sediments dated by radiocarbon to MIS 3 are situated at 30 MASL (Dionne and Occhietti, 1996). Finally, the Severn Marine site in the Hudson Bay Lowlands was dated to MIS 3, using OSL techniques (55 MASL; Dalton et al., 2016). However, whether Hudson Bay was ice-free or remained glaciated through MIS 3 remains a topic of ongoing discussion (Dalton et al., 2019; Miller et al., 2019).

### 8.3 Holocene sea level databases

Holocene marine sediments are widely preserved in the glaciated Northern Hemisphere and are documented in several databases, including the Russian Arctic (Baranskaya et al., 2018), the Baltic Sea (Rosentau et al., 2021), European west coast (García-Artola et al., 2018), Greenland (Long et al., 2011) and eastern Canada (Vacchi et al., 2018).

## 9 CONCLUSIONS AND FUTURE RESEARCH

Reconstructing sea level change through the MIS 5e/LIG is critical for understanding the sensitivity of the Earth System to

future change. We here contribute 82 LIG sea level proxies from the glaciated Northern Hemisphere to the World Atlas of Last Interglacial Shorelines (WALIS) database. Given their position in the envelope of Northern Hemisphere ice sheets, these data should also be useful for inferring the relative size of MIS 6 ice sheets. Obtaining an accurate chronology remains one of the most significant challenges for LIG deposits in the glaciated region. When geochronological data are lacking,

often the only way to distinguish between interglacial deposits is via palaeoecological inferences, which have their own set
of uncertainties. Therefore, key areas of future research should focus on revisiting these sites to vet the stratigraphic record
and testing new geochronological methods (especially U-Th and OSL techniques).

## DATA AVAIABILITY

The database on LIG sites in the glaciated Northern Hemisphere is available here: https://doi.org/10.5281/zenodo.5602212
(Dalton et al., 2021). A detailed description of database fields in the WALIS database is available here:
https://doi.org/10.5281/zenodo.3961544 (Rovere et al., 2020).

## AUTHOR CONTRIBUTIONS

ASD and EJG conceptualised this paper. ASD wrote the first draft with substantial input from JM, PM, JPL and VA. EJG
managed data entry into the WALIS database and developed the methodological framework. All authors reviewed/edited
subsequent drafts of the manuscript.

## 1335 COMPETING INTERESTS

The authors declare no competing interests.

## DISCLAIMER

The information contained in this database was the result of studies from many scientists over the course of several decades.
Please cite the original sources of the data in addition to this database.

## 1340 ACKNOWLEDGEMENTS

Specific project funding for research on Taimyr/Russia was provided through grants from the Swedish Natural Science
Research Council (VR) to P. Möller (contract nos. G-650-199815671/2000 and 621-2008-3759) and logistics were mainly
arranged and, to a large extent, funded by the Swedish Polar Research Secretariat (SPRS). Field work for Jan Mangerud was
funded by several consecutive grants from the Research Council of Norway. Overview of Russian sites by Valery Astakhov
was performed according to the research plan of the St. Petersburg State University, Russia. E.J.G. was funded by a Japan
Society for the Promotion of Science Fellowship, Helmholtz Exzellenznetzwerks "The Polar System and its Effects on the
Ocean Floor (POSY)" and Helmholtz Climate Initiative REKLIM (Regional Climate Change), a joint research project at the
Helmholtz Association of German research centres (HGF). This study was also supported by the PACES-II program at the



Alfred Wegener Institute and the Bundesministerium für Bildung und Forschung funded project, PalMod. The WALIS
database was developed by the ERC Starting Grant "Warmcoasts" (ERC-StG-802414) and PALSEA. PALSEA is a working
group of the International Union for Quaternary Sciences (INQUA) and Past Global Changes (PAGES), which in turn
received support from the Swiss Academy of Sciences and the Chinese Academy of Sciences.  The database structure was
designed by A. Rovere, D. Ryan, T. Lorscheid, A. Dutton, P. Chutcharavan, D. Brill, N. Jankowski, D. Mueller, M. Bartz,
E.J. Gowan and K. Cohen. Thanks go to Svend Funder for helpful discussions.

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

inundated the isostatically depressed landscape of western Russia and Europe.



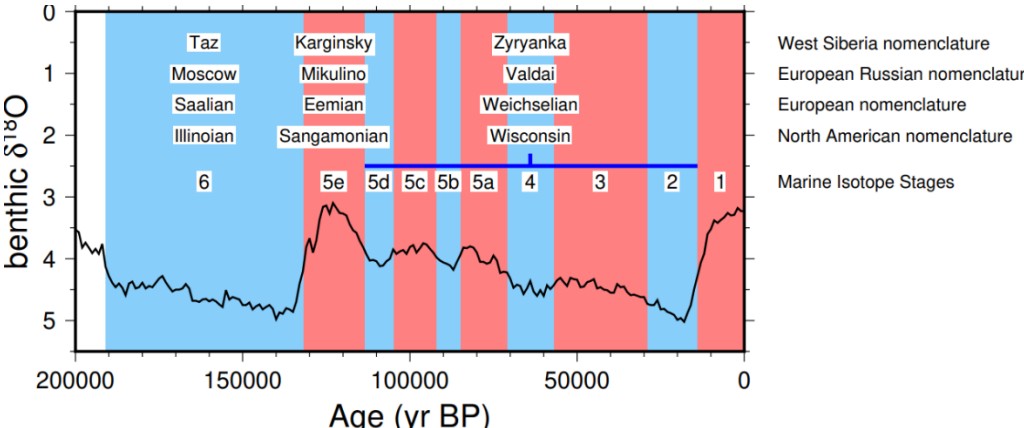

**Figure 2.** The timescales covered in this manuscript, with regional nomenclature.


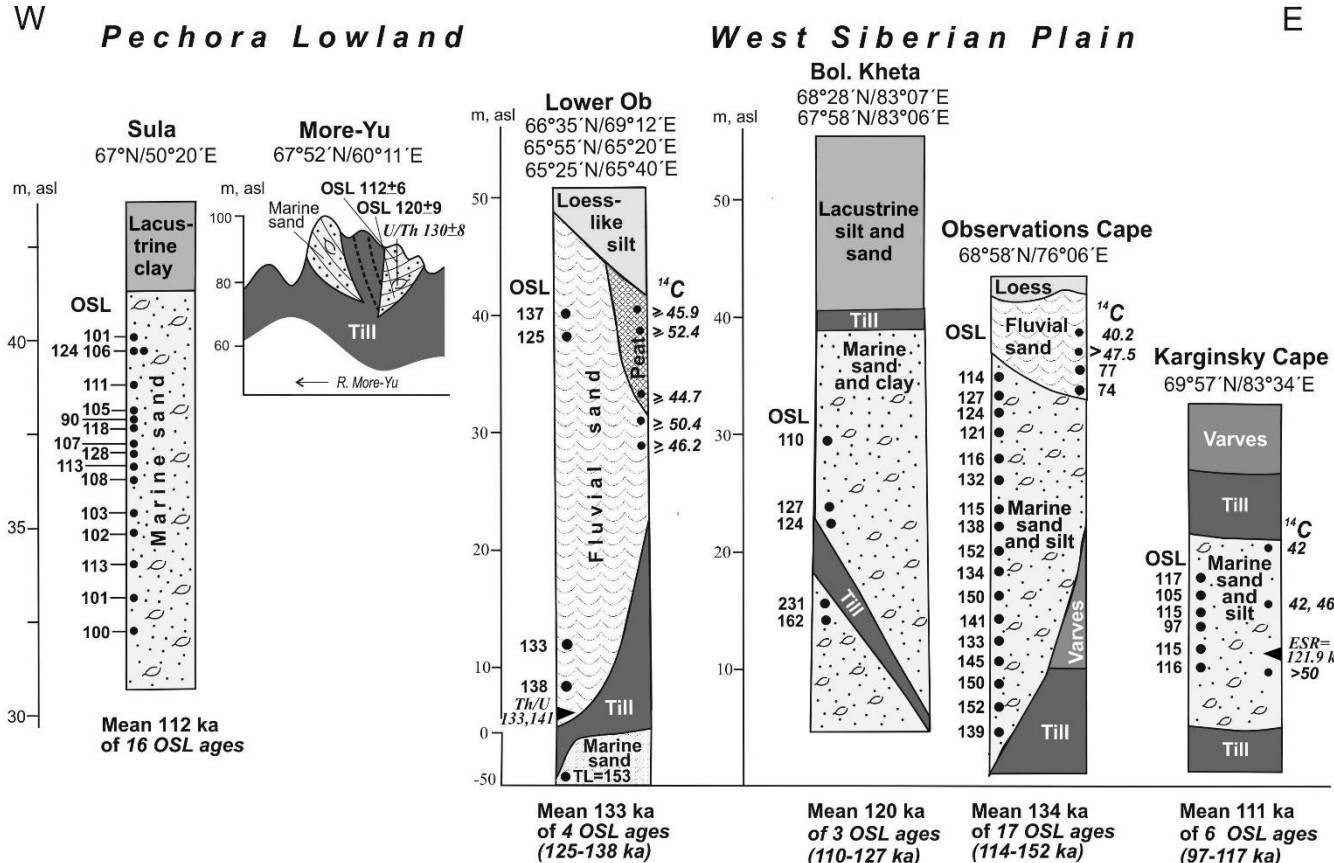

**Figure 3.** Best dated sections of the last interglacial in the Russian Arctic mainland. Numbers along the columns are geochronometric ages in kilo years. Relative positions of samples are shown by black dots: large ones – dated by optical luminescence, smaller ones – by radiocarbon; black triangles indicate dates by uranium series and electron-spin resonance methods. Sources of information: Sula (Murray et al., 2007), More-Yu (Astakhov and Svendsen, 2002), Nadym Ob (Astakhov et al., 2004, 2005), Observations Cape (Astakhov and Nazarov, 2010a, b), Bol. Kheta (Nazarov et al., 2020), Karginsky Cape (Astakhov, 2013; Nazarov et al., 2020). These stratigraphic sections are sometimes summaries of several sites; see text for details.



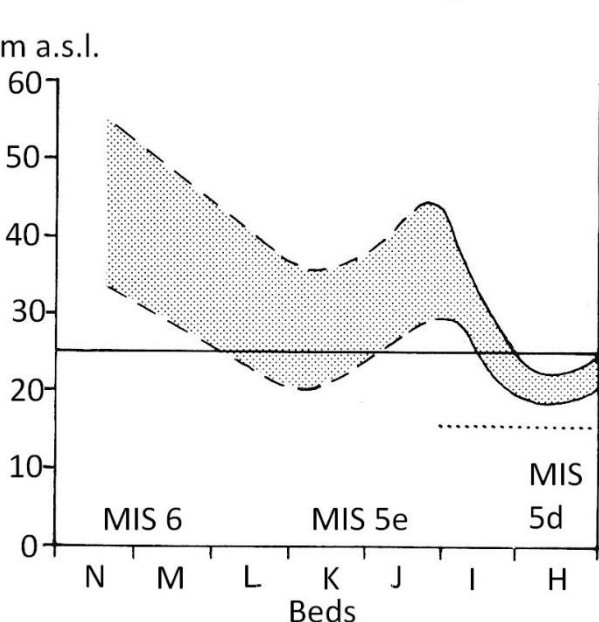

**Figure 4.** Relative sea-level curve for Fjøsanger during the entire (Eemian ≈) MIS 5e, shown as a dotted uncertainty band.
The dashed part of the curve is considered most uncertain. The line at 25 m a.s.l. shows the threshold in the Bergen Valley; when sea level was higher than this threshold the fjord turned into a sound that was open in both ends. The dotted line at 15 m a.s.l. shows the top of the marine MIS 5e sediments and thus the minimum altitude of the relative sea level during the entire MIS 5e.






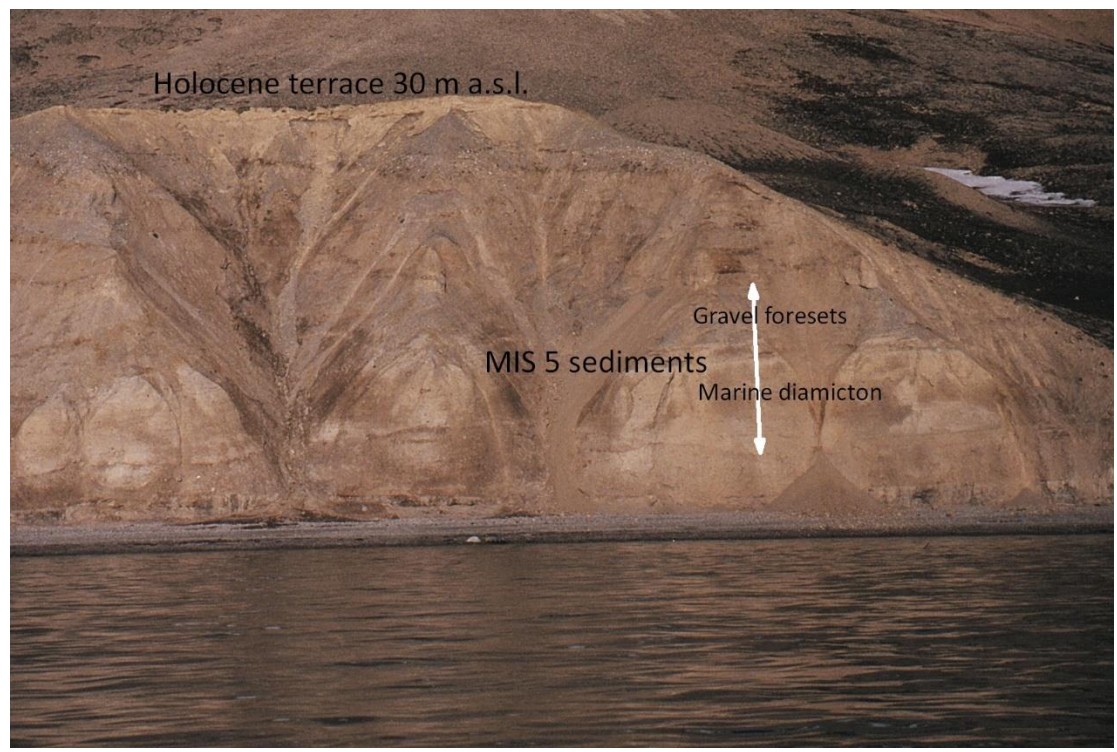

**Figure 5.** Photo of Section II at Kapp Ekholm, Svalbard. The MIS 5 sediments (white arrow) are about 10 m thick. Most *Mytilus edulis* were found in the lower part of the marine diamicton. Photo J. Mangerud 1988.




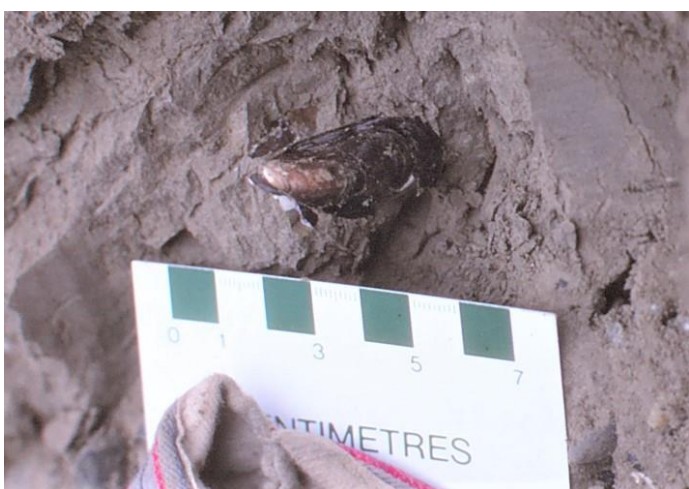

**Figure 6.** Preserved *Mytilus edulis* in the MIS 5e sediments at Kapp Ekholm, Svalbard. This mollusc requires warmer water than around Svalbard for the last millennia, until it immigrated in 2004. Photo J. Mangerud 1988.





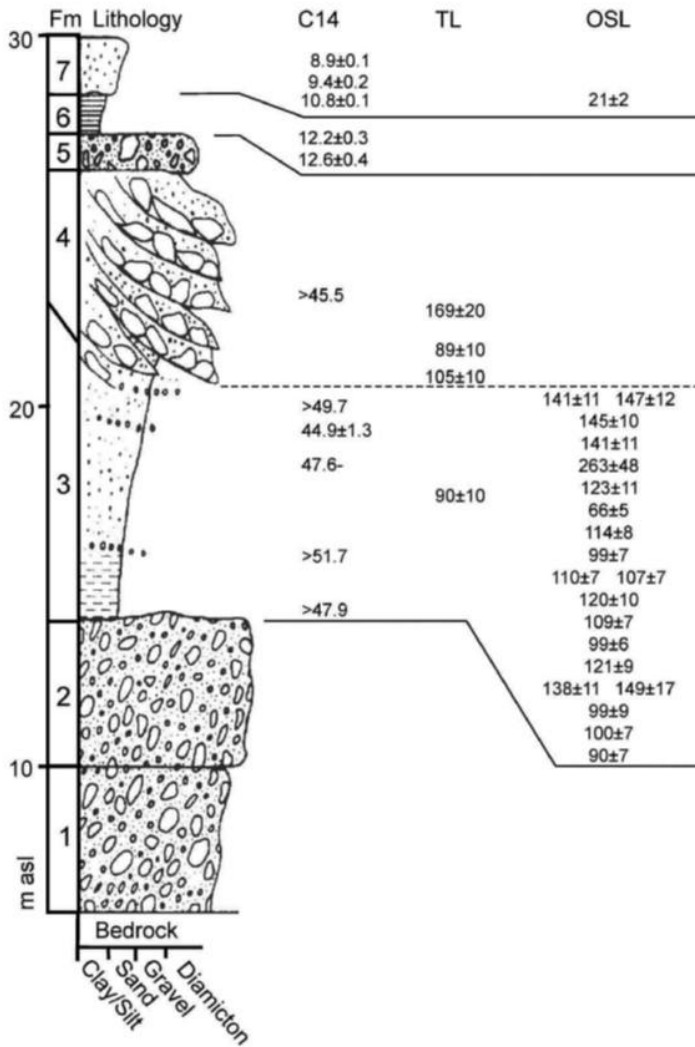

**Figure 7.** Composite log of the Skilvika exposure with luminescence (TL and OSL in ka) and radiocarbon ages (in cal. ka BP for ages > 40 ka BP, un-calibrated for younger ages). Formations (Fm) 3 and 4 are the candidates for MIS 5e. From Alexanderson and Landvik (2018).






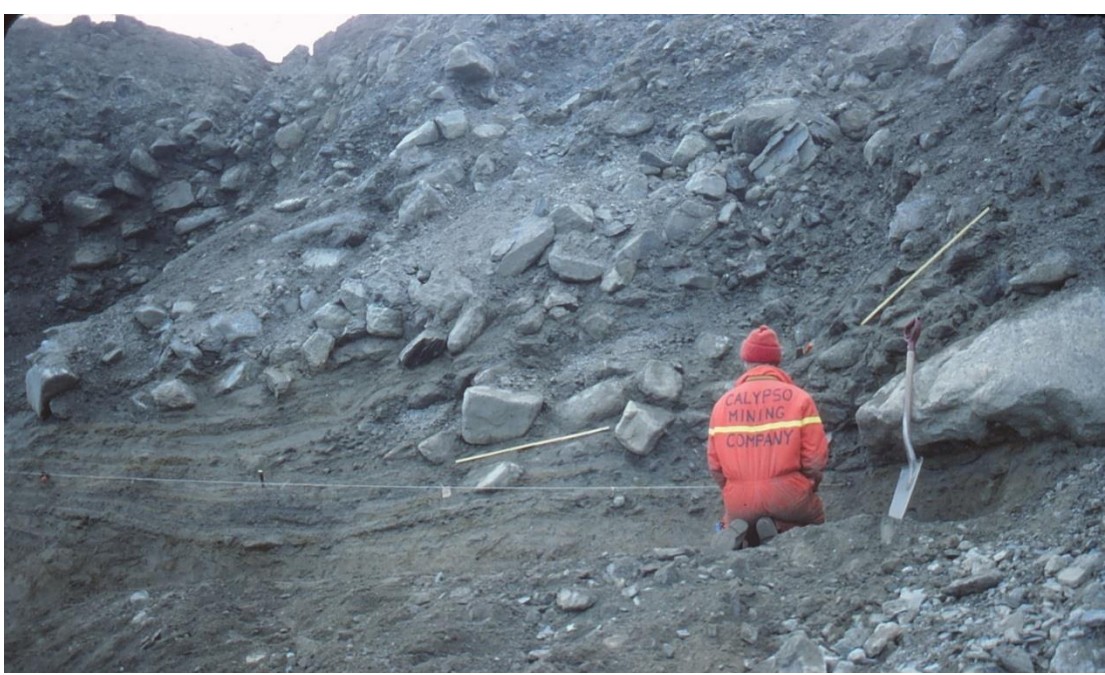

**Figure 8.** The upper half of the photo shows the extremely coarse forests in Formation 4 at Skilvika. Note how the foresets interfinger with the sand and gravel beds in Formation 3. Photo J. Mangerud 1984.





**Figure 9.** Summary of quality scores for the RSL data (upper panel) and chronology data (lower panel).






TABLES


**Table 1: Measurement techniques used to establish the elevation of MIS 5e marine deposits (Rovere et al., 2020).**

| Measurement technique | Description | Typical accuracy |
|---|---|---|
| Barometric altimeter | Difference in barometric pressure between a point of known elevation (often sea level) and a point of unknown elevation. Not accurate and used only rarely in sea-level studies | Up to ±20% of elevation measurement |
| Cross-section from publication | The elevation was extracted from a published sketch/topogrpahic section. | Variable, depending on the scale of the sketch or topographic section. |
| Differential GPS | GPS positions acquired in the field and corrected either in real time or during post-processing with respect to the known position of a base station or a geostationary satellite system (e.g. Omnistar). Accuracy depends on satellite signal strength, distance from base station, and number of static positions acquired at the same location. | ±0.02/±0.08 m, depending on survey conditions and instruments used (e.g., single-band vs dual-band receivers) |
| Distance from top of drill core | Distance from the top of drill core. | Depending on coring technique and sampling procedures |
| Handheld GPS | Commercial hand-held GPS | Dependent on model and satellite coverage but could be as low as 1-2 m. |
| Inclinometer | Elevation measured with inclinometer starting from a point of known altitude. | Variable depending on the distance between reference and measured point. |
| Metered tape or rod | The end of a tape or rod is placed at a known elevation point, and the elevation of the unknown point is calculated using the metered scale and, if necessary, clinometers to calculate angles. | Up to ±10% of elevation measurement |
| Not reported | The elevation measurement technique was not reported, most probably hand level or metered tape. | 20% of the original elevation reported added in root mean square to the sea level datum error |
| Theodolite and rod | Elevation derived from triangulation with a theodolite. | Usually very precise, centimetric accuracy, depending on distance. |
| Topographic map and digital elevation models | Elevation derived from the contour lines on topographic maps. Most often used for large-scale landforms (i.e. marine terraces). Several meters of error are possible, depending on the scale of the map or the resolution of the DEM | Variable with scale ofmap and technique used to derive DEM. |



**Table 2: Sea level datums reviewed in this study (Rovere et al., 2020).**

| Datum name | Datum description |
|---|---|
| Mean Sea Level / General definition | General definition of MSL, with no indications on the datum to which it is referred to. A datum uncertainty can be established on a case-by-case basis. |

**Table 3. Quality scores for sea level proxies.**

| Description | Quality rating |
|---|---|
| Sedimentary sequence confidently shows a regression from deep (> 10 m) marine to terrestrial conditions, or the deposit provides, via age constraints, a limiting elevation for a long portion of the interglacial. Elevation measurements are well documented to be within 3 m. | 5 (excellent) |
| Same as above, but details of elevation measurements are not well documented. | 4 (good) |
| Sedimentary sequence is regressive, but it is less certain that sea level fell below the elevation of the top of the sequence (*e.g.* beach sediments). | 3 (average) |
| Sedimentary sequence is marine limiting, but there is evidence that sea level was within 10 m of the top of the sequence (*e.g.* shoreface and wave influenced littoral sediments). | 2 (poor) |
| Sedimentary sequence is marine limiting, but there is evidence of sea level fluctuations. | 1 (very poor) |
| The sequence is marine or terrestrial limiting, and there is no information on potential sea level position. | 0 (minimal) |


**Table 4. Quality scores for age, from the WALIS documentation (Rovere et al., 2020)**

| Description | Quality rating |
|---|---|
| Very narrow age range, e.g. a few thousand's of years, that allow the attribution to a specific timing within a substage of MIS 5 (e.g. 117±2 ka) | 5 (excellent) |
| Narrow age range, allowing the attribution to a specific substage of MIS 5 (e.g., MIS 5e) | 4 (good) |
| The RSL data point can be attributed only to a generic interglacial (e.g. MIS 5) | 3 (average) |
| Only partial information or minimum age constraints are available | 2 (poor) |
| Different age constraints point to different interglacials | 1 (very poor) |
| Not enough information to attribute the RSL data point to any Pleistocene interglacial. | 0 (rejected) |




**Table 5. Summary of LIG sites from the glaciated Northern Hemisphere that contain RSL indicators. Further details are available in Dalton et al., (2021). AAR = Amino Acid Racemization; Lum.= Luminescence; ESR = Electron Spin Resonance; ChrStrat.= Chronostratigraphy.**

| Section in manuscript | WALIS RSL ID | Site name | Subsite | Nation | Lat. (DD) | Long. (DD) | RSL indicator elevation (m) | RSL indicator elevation error (m) | Age attribution | Quality of RSL data | Quality of age data |
|---|---|---|---|---|---|---|---|---|---|---|---|
| 6.1 | 4123 | Novorybnoye 2 | Unit F | Russia | 72.83 | 105.79 | 24 | 2 | Lum.; ESR | 2 | 4 |
| 6.2 | 4124 | Bol'shkaya Balakhnya River (BBR 17) | Unit A | Russia | 73.62 | 105.36 | 13 | 2 | ESR; ChrStrat. | 0 | 3 |
| 6.3 | 4091 | Kamennaya River | | Russia | 76.53 | 103.52 | 132.6 | 10 | ChrStrat. | 4 | 4 |
| 6.4 | 4088 | Kratnaya River | KR1 | Russia | 77.51 | 103.21 | 43.3 | 2 | Lum.; ESR; Other dating | 1 | 3 |
| 6.4 | 4089 | Kratnaya River | KR2 | Russia | 77.51 | 103.20 | 39.1 | 2 | Lum.; ESR; Other dating | 1 | 3 |
| 6.4 | 4090 | Kratnaya River | KR3 | Russia | 77.50 | 103.20 | 36.2 | 2 | Lum.; ESR; Other dating | 1 | 3 |
| 6.5 | 4085 | Anjeliko River | AR3 | Russia | 77.35 | 102.73 | 58.5 | 2 | Lum.; ESR; Other dating | 3 | 3 |
| 6.5 | 4086 | Anjeliko River | AR4 | Russia | 77.36 | 102.68 | 59.2 | 2 | Lum.; ESR; Other dating | 3 | 3 |
| 6.5 | 4087 | Anjeliko River | Bolotniy River BR1 | Russia | 77.39 | 102.66 | 48.8 | 2 | Lum.; ESR; Other dating | 4 | 3 |
| 6.6 | 4000 | October Revolution Island | Ozernaya River, highest beach ridge | Russia | 79.12 | 96.92 | 140 | 5 | AAR; Lum.; ESR; Other dating | 4 | 3 |
| 6.7 | 4139 | Lower Agapa River | | Russia | 71.60 | 88.30 | 63 | 5 | ChrStrat. | 1 | 4 |
| 6.8 | 4140 | Karginsky Cape | | Russia | 69.95 | 83.57 | 21 | 5 | Lum.; ESR; ChrStrat. | 0 | 4 |
| 6.9.1 | 4147 | Tanama | Tanama 1 | Russia | 70.24 | 79.76 | 65 | 5 | Lum.; ChrStrat. | 3 | 4 |
| 6.9.2 | 4148 | Tanama | Tanama 2 | Russia | 69.83 | 79.00 | 65 | 5 | Lum.; ChrStrat. | 3 | 4 |
| 6.10.1 | 4255 | Bol'shaya Kheta | Site 7251 | Russia | 68.47 | 83.12 | 30 | 0.5 | Lum.; ChrStrat. | 2 | 4 |
| 6.10.2 | 4256 | Bol'shaya Kheta | Site 7248 | Russia | 67.97 | 83.10 | 30 | 0.5 | Lum.; ChrStrat. | 2 | 4 |
| 6.10.3 | 4257 | Bol'shaya Kheta | Site 7249 | Russia | 68.00 | 83.13 | 30 | 0.5 | Lum.; ChrStrat. | 2 | 4 |
| 6.10.4 | 4258 | Bol'shaya Kheta | Site 7246 | Russia | 67.96 | 83.21 | 30 | 0.5 | Lum.; ChrStrat. | 2 | 4 |
| 6.11 | 4151 | Observation Cape | | Russia | 68.97 | 76.10 | 35 | 5 | Lum. | 2 | 3 |
| 6.12 | 4152 | Sula | Sula 21/22 | Russia | 67.00 | 50.34 | 50 | 5 | Lum.; ChrStrat. | 4 | 4 |
| 6.13 | 4153 | River Yangarei | Yangarei-1 | Russia | 68.70 | 61.83 | 70.5 | 5 | Lum.; ChrStrat. | 0 | 4 |
| 6.14 | 4154 | Vorga-Yol | | Russia | 66.70 | 56.75 | 91 | 5 | Lum. | 2 | 4 |
| 6.15.1 | 4188 | Pyoza River | Zaton site | Russia | 65.58 | 44.63 | 10 | 2 | AAR; ESR; ChrStrat. | 4 | 4 |
| 6.15.2 | 4189 | Pyoza River | Bychye site | Russia | 65.79 | 45.06 | 23 | 4.6 | ChrStrat. | 2 | 4 |
| 6.15.3 | 4190 | Pyoza River | Viryuga W. | Russia | 65.82 | 46.00 | 49 | 9.8 | ChrStrat. | 0 | 3 |
| 6.15.4 | 4191 | Pyoza River | Viryuga E. | Russia | 65.82 | 46.00 | 63 | 10 | ChrStrat. | 2 | 4 |
| 6.15.5 | 4192 | Pyoza River | Kalinov | Russia | 65.79 | 46.22 | 37 | 7.4 | ChrStrat. | 3 | 4 |
| 6.15.6 | 4193 | Pyoza River | Yatsevets | Russia | 65.70 | 46.52 | 38 | 7.6 | ChrStrat. | 1 | 4 |



| 6.15.7 | 4194 | Pyoza River | Site 11 Orlovets | Russia | 65.71 | 46.84 | 43.5 | 8.7 | ChrStrat. | 3 | 4 |
|---|---|---|---|---|---|---|---|---|---|---|---|
| 6.15.8 | 4195 | Pyoza River | Site 12 Orlovets | Russia | 65.69 | 46.93 | 43.5 | 8.7 | ChrStrat. | 3 | 4 |
| 6.15.9 | 4196 | Pyoza River | Site 13 Yolkino | Russia | 65.68 | 47.60 | 51 | 10 | ChrStrat. | 2 | 4 |
| 6.15.10 | 4197 | Pyoza River | Site 14 Yolkino | Russia | 65.68 | 47.60 | 51 | 10 | Lum.; ChrStrat. | 2 | 4 |
| 6.15.11 | 4198 | Pyoza River | Burdui | Russia | 65.67 | 48.06 | 60 | 10 | ChrStrat. | 2 | 4 |
| 6.16 | 4155 | Ponoi River | Unit 2 | Russia | 67.08 | 41.13 | 11.5 | 2.3 | Lum.; ESR; ChrStrat. | 2 | 4 |
| 6.17 | 4156 | Svyatoi Nos | | Russia | 68.02 | 39.87 | 16 | 3.2 | Lum.; ChrStrat. | 0 | 4 |
| 6.18 | 4157 | Chapoma | | Russia | 66.11 | 38.97 | 10 | 2 | ESR; ChrStrat. | 4 | 4 |
| 6.19 | 4158 | Strelna River | | Russia | 66.01 | 33.64 | 35.5 | 7.1 | ESR; ChrStrat. | 2 | 4 |
| 6.2 | 4159 | Varzuga | S1 | Russia | 66.40 | 36.64 | 14 | 2.8 | ChrStrat. | 0 | 4 |
| 6.21 | 4160 | Petrozavodsk | | Russia | 61.77 | 34.40 | 40 | 8 | ChrStrat. | 1 | 4 |
| 6.22 | 3712 | Peski | | Russia | 60.15 | 29.29 | 13.5 | 8 | ChrStrat. | 4 | 4 |
| 6.23 | 3711 | Põhja-Uhtju | | Estonia | 59.68 | 26.51 | -49 | 1 | ChrStrat. | 2 | 4 |
| 6.24 | 3985 | Suur-Prangli | | Estonia | 59.62 | 25.01 | -61 | 1 | ChrStrat. | 2 | 4 |
| 6.25.1 | 3987 | Lower Vistula Region | Obrzynowo | Poland | 53.78 | 19.27 | -3.5 | 10 | ChrStrat. | 4 | 4 |
| 6.25.2 | 3986 | Lower Vistula Region | Licze | Poland | 53.75 | 19.13 | -8 | 10 | ChrStrat. | 4 | 4 |
| 6.26.1 | 3990 | Rewal coastline | Rewal borehole | Poland | 54.09 | 15.03 | -5.5 | 0.4 | ChrStrat. | 3 | 4 |
| 6.26.2 | 3991 | Rewal coastline | Ciećmierz borehole | Poland | 53.99 | 15.03 | -6.5 | 4 | ChrStrat. | 3 | 4 |
| 6.26.3 | 3989 | Rewal coastline | Sliwin borehole | Poland | 54.08 | 15.01 | 6.3 | 0.8 | ChrStrat. | 3 | 4 |
| 6.27 | 4161 | Ollala | Borehole F | Finland | 64.18 | 25.35 | 116.5 | 1 | ChrStrat. | 4 | 4 |
| 6.28 | 4162 | Ukonkangas | | Finland | 63.91 | 25.85 | 105.7 | 1 | ChrStrat. | 5 | 4 |
| 6.29 | 4163 | Viitala | | Finland | 62.60 | 23.00 | 84.5 | 1 | ChrStrat. | 2 | 4 |
| 6.3 | 4164 | Mertuanoja | | Finland | 34.11 | 24.59 | 60 | 1 | Lum.; ChrStrat. | 4 | 4 |
| 6.31 | 3988 | Norra Sannäs | | Sweden | 61.78 | 16.69 | 27.65 | 9 | ChrStrat.; Other dating | 4 | 4 |
| 6.32 | 3708 | Fjøsanger | | Norway | 60.34 | 5.33 | 15 | 0.1 | AAR; Lum. | 5 | 4 |
| 6.33 | 3709 | Bø | | Norway | 59.36 | 5.28 | -1 | 0.1 | AAR; ChrStrat. | 5 | 4 |
| 6.34 | 3833 | Hidalen | Unit D | Svalbard and Jan Mayen | 78.90 | 28.13 | 83 | 2 | AAR | 3 | 2 |
| 6.34 | 3834 | Hidalen | Unit B | Svalbard and Jan Mayen | 78.90 | 28.13 | 64 | 2 | AAR; Lum. | 2 | 2 |
| 6.35 | 3787 | Kapp Ekholm | Formation B | Svalbard and Jan Mayen | 78.55 | 16.55 | 22 | 0.1 | AAR; Lum.; ChrStrat. | 3 | 4 |
| 6.36 | 3810 | Skilvika | Formation 3 | Svalbard and Jan Mayen | 77.57 | 14.44 | 28 | 0.5 | Lum. | 3 | 3 |
| 6.37 | 3799 | Kongsfjordhallet | | Svalbard and Jan Mayen | 79.03 | 11.88 | 34 | 1 | Lum. | 3 | 3 |
| 6.38 | 3809 | Poolepynten | Unit A1 | Svalbard and Jan Mayen | 78.45 | 11.66 | 5 | 2 | AAR; Lum.; Other dating | 3 | 4 |
| 6.39 | 3788 | Leinstranda | | Svalbard and Jan Mayen | 78.88 | 11.56 | 19.2 | 0.5 | Lum.; ChrStrat. | 3 | 4 |
| 6.4 | 3710 | Galtalækur | | Iceland | 63.99 | -19.96 | 120 | 10 | ChrStrat.; Other dating | 2 | 3 |
| 6.41.1 | 4168 | Kikiakajik | Beach | Greenland | 70.04 | -22.25 | 9.75 | 3.4 | Lum.; ChrStrat.; Other dating | 3 | 4 |
| 6.41.2 | 4166 | Kap Hope | Regressive sequence | Greenland | 70.46 | -22.32 | 8 | 2.23 | Lum.; ChrStrat.; Other dating | 5 | 4 |





| 6.41.2 | 4167 | Kap Hope | Transgressive sequence | Greenland | 70.46 | -22.32 | 15 | 1 | Lum.; ChrStrat.; Other dating | 3 | 4 |
|---|---|---|---|---|---|---|---|---|---|---|---|
| 6.41.3 | 4169 | Kap Stewart | | Greenland | 70.44 | -22.78 | 40 | 8 | Lum.; ChrStrat. | 4 | 4 |
| 6.41.4 | 4170 | Hesteelv | | Greenland | 70.44 | -23.10 | 35 | 7 | Lum.; ChrStrat. | 3 | 4 |
| 6.41.5 | 4172 | Fynselv | 443d | Greenland | 70.46 | -23.32 | 21.5 | 4.3 | Lum.; ChrStrat. | 3 | 4 |
| 6.41.6 | 4254 | Langelandselv | composite section | Greenland | 70.55 | -23.64 | 70 | 10 | AAR; Lum.; ChrStrat. | 4 | 4 |
| 6.41.7 | 4181 | Aucellaelv River | Location 72 | Greenland | 70.59 | -23.76 | 16 | 3.2 | AAR; Lum.; ChrStrat. | 1 | 4 |
| 6.41.8 | 4182 | Lollandselv-Falsterselv | | Greenland | 70.87 | -24.18 | 20 | 4 | ChrStrat. | 4 | 2 |
| 6.42.1 | 4199 | Thule | Iterlak K | Greenland | 76.71 | -69.41 | 27.5 | 5.5 | AAR; ChrStrat.; Other dating | 2 | 4 |
| 6.42.2 | 4200 | Thule | Iterlak L | Greenland | 76.71 | -69.42 | 14 | 2.8 | Lum.; ChrStrat. | 2 | 4 |
| 6.42.3 | 4201 | Thule | Saunders Ø B | Greenland | 76.60 | -69.74 | 20 | 4 | Lum.; ChrStrat. | 3 | 4 |
| 6.42.4 | 4202 | Thule | Saunders Ø C | Greenland | 76.60 | -69.74 | 12 | 2.4 | Lum.; ChrStrat. | 3 | 4 |
| 6.42.5 | 4203 | Thule | Narsaarsuk D | Greenland | 76.45 | -69.29 | 6 | 1.2 | AAR; ChrStrat. | 2 | 4 |
| 6.42.6 | 4204 | Thule | Narsaarsuk E | Greenland | 76.45 | -69.29 | 6 | 1.2 | AAR; Lum.; ChrStrat. | 2 | 4 |
| 6.42.7 | 4205 | Thule | Narsaarsuk F+G | Greenland | 76.45 | -69.29 | 9 | 1.8 | AAR; Lum.; ChrStrat. | 2 | 4 |
| 6.43.1 | 3661 | Iles de la Madeleine | Camping | Canada | 47.35 | -61.88 | 14 | 4 | Lum. | 2 | 3 |
| 6.43.2 | 3640 | Iles de la Madeleine | Portage du Cap | Canada | 47.25 | -61.91 | 17 | 4.88 | ChrStrat.; Other dating | 3 | 2 |
| 6.43.3 | 4183 | Iles de la Madeleine | Le Bassin | Canada | 47.23 | -61.90 | 2 | 1 | ChrStrat.; Other dating | 4 | 2 |
| 6.44.1 | 4185 | Clyde Foreland | Profile 6 | Canada | 70.69 | -68.95 | 28 | 5.6 | ChrStrat.; Other dating | 1 | 4 |
| 6.44.2 | 4186 | Clyde Foreland | Profile 9 | Canada | 70.60 | -68.41 | 11 | 5.6 | U-Series; AAR; ChrStrat. | 0 | 4 |
| 6.44.3 | 4187 | Clyde Foreland | Profile 10 | Canada | 70.58 | -68.35 | 2.5 | 0.5 | AAR; ChrStrat. | 1 | 4 |
| 6.45 | 3639 | Ile aux Coudres | Site du Forage | Canada | 47.41 | -70.42 | -2 | 0.4 | ChrStrat.; Other dating | 2 | 2 |
| 6.46 | 3680 | Long Island | Bridgehampton - core S59793 | United States of America | 40.94 | -72.31 | -23 | 1.4 | AAR; ChrStrat. | 1 | 2 |
| 6.47 | 3637 | Kwataboahegan River | Marine unit | Canada | 51.14 | -82.12 | 90 | 18 | AAR; Other dating | 0 | 2 |
| 6.48 | 4184 | East of Nicholson Peninsula | VH-83-050 | Canada | 69.89 | -128.52 | 2 | 0.4 | AAR; ChrStrat.; Other dating | 3 | 2 |