# Peer review of "Last interglacial sea-level proxies in the glaciated Northern Hemisphere"

_Earth System Science Data, 2021_

## Author Response (AR1)

**Replies to Anonymous Referee #1**

Last interglacial (MIS 5e) sea level proxies in the glaciated Northern Hemisphere

In this review the authors thoroughly researched the sea level proxies from a variety of settings within the glaciated Northern Hemisphere (Russia, northern Europe, Greenland and North America). This compilation is a a valuable contribution to the World Atlas of Last Interglacial Shorelines database and should prove an extremely useful resource for sea-level scientists and those in other fields in future.  After defining the types of sea level proxies, elevations measurements, dating techniques and quality assessment, the authors do an excellent job in describing in detail each site.

> *Response: We thank the Reviewer for their thorough review of our manuscript. They are clearly well-versed in the topic of the last interglacial, and we have adopted many of their suggestions, as detailed below.*

**General comments**

I understand the authors' point that given the scale of this database is rather difficult to map all the locations described in Figure 1 and the reader is referred to the original publications for additional site information. However, since the main component of the manuscript is the detailed description of each site, I think it would be very useful to include, close up maps for all these locations. For example, one suggestion would be to consider including the photos from figures 5 and 7 in one single figure which also includes a map indicating the location of this site and the ones nearby. Similar recommendation for Figures 7 and 8.

> *Response: We agree this is a good idea. In the revised manuscript, we add 6 regional maps to show locations where LIG sites are clustered. In these maps we also label local geographic features that are mentioned in the text to better situate the reader.*

I would also recommend the authors to expand on the Conclusions section and future research directions that could merit additional work, and to highlight how this compilation will help those interested in inferring the size of the MIS 6 ice sheets through GIA modeling.

> *Response: We agree this is a good idea. In the revised manuscript, we add additional text to the Conclusion that highlights the utility of this dataset for reconstructing MIS 6 ice sheets. We also emphasize this point in the Abstract and Introduction.*

In my opinion, it is not necessarily to include Tables 2 and 4 but instead the authors can include the information from Table 2 in the text (since it is only one line) and to refer the readers to Rovere et al (2020) for information in Table 4.

> *Response: We agree that the information from these tables is repeated in the text (especially Table 2), but the inclusion of Tables 2 and 4 follow the formatting of the articles in this Special Issue. We make no changes to the revised manuscript.*

In the Relative sea level proxies section I would like to make some suggestions to avoid repeating the same wording too many times:

Removing lat/long from each subsection title since they are already listed in Table 5.

> *Response: We agree, this is a good ideal. In the revised manuscript, we remove the lat/long from the subsection titles. We further clarify at the beginning of Section 6: "Additional information (including site coordinates, elevation of marine sediments, quality scores for both RSL and age determinations) is summarized in Table 5 and detailed in the database of Dalton et al., (2021)."*

Perhaps it would be easier to follow the sites list organizing them by country, for example, 6.1. Rusia, 6.1.1. Novorybnoye 2, Taimyr peninsula, etc., without having to mention the country again in each subsection title.

> *Response: We agree this is a good suggestion, but during the development of this manuscript we experimented with several methods for organizing the site descriptions. There are several pros/cons to each technique, and we ultimately decided to be flexible with the site labelling and be flexible with the ordering of the sites to keep the descriptions flowing east to west. For example, Svalbard is technically part of Norway, but grouping the Svalbard sites with those from mainland Norway caused significant 'geographical jumping around' in the text. Therefore, these sites were described separately from the mainland Norway sites.*

I suggest merging sub-sections 6.9.1 and 6.9.2 in 6.9. and also rephrasing to something along the line: "the unit in site 1 and 2 is associated with terraces that reach an elevation…"

> *Response: We agree this is redundant, but we keep these as separate sections because they are separate entries in the database. We aim to make each site description a "stand-alone" piece of text in this manuscript.*

Section 6.10, to avoid repeating "(marine sand, and clays (constrained to MIS 5e; LIG) are located between …), maybe use instead something like: "marine sand, and clays (constrained to MIS 5e; LIG) are located …" and then list the 4 sites.

> *Response: A noted above, we agree this is somewhat redundant, but we use this notation because these are separate sites in the database. Moreover, some of the details (eg. Elevation of marine sediments and age constraints) are different at these sites. Therefore, it would be difficult to merge this into a single site description. We aim to make each site description a "stand-alone" piece of text in this manuscript.*

Section 6.15, I suggest deleting "site" from the first 2 subsections titles. It seems that at each site description there is the same sentence that: "A LIG-age is also suggested by correlation of pollen data from this site with the climate for western Europe (Zagwijn, 1996). To avoid

repeating this, the authors could mention this once at the beginning of the section and to mention that this applies to all sites in 6.15.

> *Response: As suggested, we removed "site" from the two sub-section titles. Regarding the pollen data: we agree the same references are used in most sub-sections. However, the age inferred from pollen data is different throughout the 11 sections described here. For example, pollen data suggest a broad age assignment of 133-119.5 ka for the Zaton site, and pollen data offer a more constricted age assignment for the Bychye site (133-124 ka). We therefore find it appropriate to cite the pollen studies and provide additional site-specific context in each subsection.*

Section 6.26, I do not see it necessary to create subsections, but instead I suggest list the 3 boreholes. This way, the word "borehole" is not repeated as often as it is now.

> *Response: As part of the standard formatting used in this manuscript, we created a separate subsection for each site where LIG marine sediments were located. We feel this format is best suited for the reader who is interested in knowing the location of individual occurrences of the LIG marine sediments. Therefore, in this section, we describe 3 separate boreholes.*

To avoid confusion, I recommend to clarify the difference between "a date" and "an age" and use it correspondingly throughout the manuscript. I also suggest merging the subsections 4.1.5.- 4.1.7. into Luminescence dating methods.

> *Response: We use the 'date' and 'age' notation interchangeably, depending on the context of the site. We broke down "Luminescence dating methods" into 3 separate subsections to reflect 3 very different dating methods and to permit discussion of the pros/cons and key characteristics associated with each.*

For consistency throughout the manuscript, please consider the following:

Use properly "sea-level" and "sea level"

> *Response: Thank you to the Reviewer for pointing this out. We have made several corrections of this in the revised manuscript, including the title.*

Choose one of U-Th, U/Th, 230Th, uranium/thorium, 230Th/234U when you refer in the text to the U-series age. Same for radiocarbon dating vs $^{14}C$ .

> *Response: In the revised manuscript, we adopt the "U/Th" notation. We also refer to "radiocarbon" thorough the revised manuscript, except for section 4.1.2 where we use "14C" in cases where we specifically refer to the decay of the 14C isotope.*

Once it has been clarified in the text that the Eemian is LIG in western, central and northern Europe, there is no need to repeat it later in the manuscript.

> *Response: We agree with this suggestion. In the revised manuscript, we define the LIG in the Introduction (equivalent to MIS 5e; Eemian…etc) and then we refer only to "LIG" afterward. We further remove "MIS5e" from the title and instead state "Last interglacial".*

Perhaps you could mention once in the description of the relative sea level proxies that: "The geochronological results are reported for all the sites where data is available." and no need to mention at each site where there are no ages that "No geochronological data are available for x site."

> *Response: We agree this is somewhat repetitive. In the revised manuscript, we adopt the suggestion of the Reviewer.*

Refer to the journal citation style and use it consistently.

> *Response: We use the citation style of the ESSD journal. Nevertheless, we noticed a few errors, and we have made some minor changes to update the text accordingly.*

Express a sea level range in the same way, i.e., 2-11 masl instead of 2 masl to 11 masl

> *Response: We have updated the manuscript accordingly and made this change throughout to standardize the notation.*

Use the same time unit "ka"

> *Response: We have updated the manuscript accordingly.*

"up-wards" or "upwards"

> *Response: We clarify 'upward' in the text.*

Some ages have uncertainties included in the text and some don't - is this based on the original publications?

> *Response: Yes, in some cases the original publication does not refer to errors. In other cases, we report the average age of a stratigraphic unit and refer the reader to the original text for more information.*

Use MIS-5e or LIG consistently

> *Response: We agree with this suggestion. In the revised manuscript, we define the LIG in the Introduction (equivalent to MIS 5e; Eemian…etc) and then we refer only to "LIG" afterward. We further remove "MIS5e" from the title and instead state "Last interglacial".*

Consider rewording some of the paragraphs so that you don't cite the same reference so many times within the same paragraph (for example, lines 471; 479; 520)

> ***Response:*** *We have carefully reviewed the manuscript and made the appropriate changes, where necessary, to clarify the text and reduce repetitive references.*

**Specific comments by line number:**

30 and 31: more up to date references for the temperature and the sea level during LIG

> ***Response:*** *We add a newer reference here, and we update to clarify that sea levels could have reached 5 m higher. New reference:*
>
> *Dyer, B., Austermann, J., D'Andrea, W. J., Creel, R. C., Sandstrom, M. R., Cashman, M., Rovere, A., Raymo, M. E.: Sea-level trends across The Bahamas constrain peak last interglacial ice melt, Proc. Natl. Acad. Sci. U.S.A., 118, e2026839118, 2021.*

39: define what MIS stands for

> ***Response:*** *We update the manuscript accordingly.*

55: I think this should read "In the first part…"

> ***Response:*** *We update the manuscript accordingly.*

62: MIS 7 time interval is not correct here

> ***Response:*** *Thank you to the reviewer for their careful read of the manuscript. We update the manuscript to correct the time range for MIS 7 (243-191 ka).*

64: delete "and" before MIS 3

> ***Response:*** *We update the manuscript accordingly.*

69: define GIA

> ***Response:*** *We update the manuscript accordingly.*

74: delete one of "the"

> ***Response:*** *We update the manuscript accordingly.*

96: "whichever"; delete "is" before "it is". I suggest rephrasing this sentence that start with "The later constraint is it is unlikely the elevation uncertainty…"

> ***Response:*** *We update the manuscript accordingly.*

118: use ka unit for the age

> ***Response:*** *We update the manuscript accordingly.*

122: "purpose"

> ***Response:*** *We are unclear what is being referred to here. We reviewed this paragraph and can find no grammatical concerns.*

127: I suggest rephrasing the sentence starting with "conversely,.."

*Response:* *We update the manuscript accordingly and replace with "Moreover".*

172: U/Th method is often applied to determine the age of other LIG deposits, so maybe specify that is not a common method to date mollusks shells in the LIG deposits.
*Response:* *This point is clarified in the next sentence of the manuscript.*

178: missing "." at the end of the sentence
*Response:* *We update the manuscript accordingly.*

189: use another word for "precisely" to avoid using the same word twice in one sentence
*Response:* *We delete the 2nd 'precise' since this word is not necessary in this sentence.*

194: I think there is a word missing in here: "The rating decreases when there is less geological evidence sea level position changes and proximity to sea level."
*Response:* *We update the manuscript accordingly.*

232: check citation style: Moller et al 2019a; 2019b?
*Response:* *We update the manuscript accordingly.*

236: consider rephrasing this: "However, two molluscs in the above-lying fluvial sediments (OSL-dated to a MIS 3 age), the molluscs redeposited from erosion of the marine sediment, yield ESR ages of 122 ka and 123 ka."
*Response:* *We re-structure this sentence to make it clearer.*

238: no need to use both MIS e and LIG
*Response:* *We update the manuscript accordingly.*

243: recommend rephrasing to: "site 373 located at the highest altitude at 133 mapl"
*Response:* *We update the manuscript accordingly.*

249: suggest: "however, the authors do no present any numerical age data."
*Response:* *We update the manuscript accordingly.*

254: check the required citation style
*Response:* *We update the manuscript accordingly.*

295: GSL ages - do you mean OSL here?
*Response:* *We clarify "Optically stimulated luminescence".*

325: replace "gave" with "turned out"?
*Response:* *We clarify "yielded ages in the range of MIS 4-3".*

341, 345: delete comma after "sand"
*Response:* *We update the manuscript accordingly.*

365: space missing after "31"
> ***Response:*** *We update the manuscript accordingly.*

388: delete MASL after "2"
> ***Response:*** *We update the manuscript accordingly.*

396: delete "at the Zaton site" - not needed because the section is dedicated to this site
> ***Response:*** *This is the beginning of a second paragraph that describes this site. Thus, we feel it is proper (and in line with the rest of the manuscript) to refer to the name of this site in the introductory sentence.*

398: define "D/L"ratio either here or in the AAR dating section
> ***Response:*** *In the revised manuscript, we clarify in the AA dating section "For the LIG, the epimerization of D-alloisoleucine to L-isoleucine is most used (known as the D/L ratio; Oldale et al., 1982; Miller and Mangerud, 1985)."*

400; 402: remove ka after 120 and after 133 respectively
> ***Response:*** *We update the manuscript accordingly.*

407: add "by" after "described". Find a synonym for "described' to avoid repetition
> ***Response:*** *We update the manuscript accordingly. We agree there is some repetition of the word "described" in this manuscript, and we have made an effort to use synonyms where appropriate.*

408: "spans from 12.5…"
> ***Response:*** *We update the manuscript accordingly.*

410: delete the space before "at"
> ***Response:*** *We update the manuscript accordingly.*

416: remove "at the Bychye site"
> ***Response:*** *This is the beginning of a second paragraph that describes this site. Thus, we feel it is proper (and in line with the rest of the manuscript) to refer to the name of this site in the introductory sentence.*

419: "cooler-than-present to warmer-than-present" - temperatures?
> ***Response:*** *We update the manuscript accordingly.*

424: add space after 21
> ***Response:*** *We update the manuscript accordingly.*

433: remove "ka" after 133
> ***Response:*** *We update the manuscript accordingly.*

456: I suggest adding "the" before "correlation" and to replace "place" with "places"
> ***Response:*** *We update the manuscript accordingly.*

482: replace "again" with "later"
> ***Response:*** *We update the manuscript accordingly.*

484: delete "at this site"
> ***Response:*** *We update the manuscript accordingly.*

507: no need to mention again LIG or MIS e
> ***Response:*** *We update the manuscript accordingly.*

509: delete ESR and OSL in brackets, and use instead 'respectively" after the second age
> ***Response:*** *We update the manuscript accordingly.*

520: add reference for the reported IRSL age.
> ***Response:*** *We update the manuscript accordingly.*

519: remove "MIS 5e"
> ***Response:*** *We update the manuscript accordingly.*

526: "span" instead of "spans"
> ***Response:*** *We update the manuscript accordingly.*

535: "workers"? Perhaps "researchers?
> ***Response:*** *We update the manuscript accordingly.*

547: "indicates" instead of "indicate"
> ***Response:*** *We update the manuscript accordingly.*

565: "described"
> ***Response:*** *We update the manuscript accordingly.*

575: what is "this" here referring to? This transition? Also, perhaps would be better to rephrase "for all of the LIG" with "Throughout the entire LIG"
> ***Response:*** *We clarify "transition", and we update the manuscript accordingly.*

592: remove ' at that site"
> ***Response:*** *We update the manuscript accordingly.*

594: spectra "show" instead of "shows"
> ***Response:*** *We update the manuscript accordingly.*

615: delete "m" after 95
>  ***Response:*** *We update the manuscript accordingly.*

627: "especially due to the presence of Picea and Carpinus; Mamakowa, 1989, 1988" - this
information has been already mentioned in the previous paragraph
>  ***Response:*** *We agree this is somewhat repetitive. However, in this sentence, we are
>  emphasizing that the specific presence of Picea and Carpinus are the reason for the LIG
>  assignment. In this manuscript, we aim to make each site description a "stand-alone"
>  piece of text, thus we add the pollen data here.*

633: "suggest that these sediments …"
>  ***Response:*** *We update the manuscript accordingly.*

642: remove  "at this site"
>  ***Response:*** *We update the manuscript accordingly.*

648: "change" instead of "changes"
>  ***Response:*** *We update the manuscript accordingly.*

649: remove "(together comprising 116.25 to 117.5 MASL)", it is clear from the context
>  ***Response:*** *We update the manuscript accordingly.*

653: remove "LIG"
>  ***Response:*** *We update the manuscript accordingly.*

654: add "that" after "suggest"
>  ***Response:*** *We update the manuscript accordingly.*

657: a reference is needed at the end of this sentence
>  ***Response:*** *We update the manuscript accordingly to clarify the appropriate references:
>  Forström et al., 1987; Grönlund, 1991a*

660: replace "metre" with "m"
>  ***Response:*** *We update the manuscript accordingly.*

662: remove "LIG"
>  ***Response:*** *We update the manuscript accordingly.*

664: remove "interglacial"
>  ***Response:*** *We update the manuscript accordingly.*

665: remove the second "the Eemian"
>  ***Response:*** *We update the manuscript accordingly.*

666: "Grönlund 1991a, b)" - or  "Grönlund, 1991a, 1991b" ? please check
    *Response: This is a citation style specific to Earth System Science Data.*

679: remove "(LIG; MIS 5e)"
    *Response: We update the manuscript accordingly.*

687: remove "it was"
    *Response: We update the manuscript accordingly.*

689: remove "(MIS 5e)"
    *Response: We update the manuscript accordingly.*

691: remove "and" before 14C
    *Response: We update the manuscript accordingly.*

697: remove "suggesting the end of marine conditions", is implied by change to freshwater
    *Response: Yes, this is correctly stated in the manuscript. Since the purpose of this manuscript is to document marine sites, we relate this transition to marine conditions (as opposed to freshwater conditions).*

699: remove "from this site"
    *Response: We update the manuscript accordingly.*

701, 706: use ka unit as in the rest of the manuscript
    *Response: We update the manuscript accordingly.*

708: add "conditions" after "warmer-than-present"
    *Response: We update the manuscript accordingly.*

716: remove the comma after MIS 6 and remove "(≈ Eemian)"
    *Response: We update the manuscript accordingly.*

721: what does "on the inland side of the large island" mean?
    *Response: We clarify that the site is located on the eastern side of a large island.*

724: remove "MBSL" after 1
    *Response: We update the manuscript accordingly.*

726: perhaps a reference is needed for the amino-acid stratigraphy
    *Response: The reference is amino acid stratigraphy is at the end of the sentence.*

750: choose either "m a.s.l." or "MASL" and use it consistently
    *Response: We update the manuscript accordingly using "MASL" throughout.*

765: remove "MIS 5e"
*Response: We update the manuscript accordingly.*

772: "1-km" instead of "one km"
*Response: We update the manuscript accordingly.*

777: "represent"
*Response: We update the manuscript accordingly.*

778: "There were found neither foraminifera…" - , I suggest rephrasing this sentence
*Response: We appreciate the suggestion. However, we feel the original text is clearer. We retain the original sentence.*

782: "The result was large spread in ages from 66 to 263 ka …" I suggest replacing with something along the line: "The resulted ages vary in a large range between 66 to 263 ka.."
*Response: We appreciate the suggestion. However, the suggested sentence is very similar to what we originally had in the manuscript. We retain the original sentence.*

783: which "these two outliers"? Add ka after 119±5
*Response: We update the manuscript accordingly. We also clarify 'two outliers'.*

784: the sentence seems to be incomplete, "they obtained a mean age of 118±7 ka"
*Response: We update the manuscript accordingly.*

788: m a.s.l. or MASL?
*Response: We update the manuscript accordingly using the MASL notation throughout.*

798: suggest using "1 million years old"
*Response: We update the manuscript accordingly.*

790: complete "we mainly rely on the results reported by Alexanderson et al. (2018)".
*Response: We update the manuscript accordingly.*

793: replace "three-meter-thick" with "3-m-thick"
*Response: We update the manuscript accordingly.*

803: replace "was" with "were"
*Response: We update the manuscript accordingly.*

804: similar suggestion as for line 790
*Response: We update the manuscript accordingly.*

805: add the word "between" before "10-130 ka BP"
*Response: We update the manuscript accordingly.*

812: replace "giving" with "of"; add "ka" unit after "118 ± 13"
> ***Response:*** *We update the manuscript accordingly.*

819: consider replacing "given" with" provided"
> ***Response:*** *We update the manuscript accordingly.*

822: replace "two-meter-thick" with "2-m-thick"
> ***Response:*** *We update the manuscript accordingly.*

847: delete "that yielded ages"
> ***Response:*** *We update the manuscript accordingly.*

848: delete "ka" after 155
> ***Response:*** *We update the manuscript accordingly.*

859: from which maps?
> ***Response:*** *We clarify that maps are from the original publications.*

861: add dash between 700 and m
> ***Response:*** *We update the manuscript accordingly.*

868: remove "s" from "indicates"
> ***Response:*** *We update the manuscript accordingly.*

875: delete "m" after 10; add dash after 5 "5-m"
> ***Response:*** *We update the manuscript accordingly.*

882: replace "In that study, workers examined" with "in their study, the authors"
> ***Response:*** *We update the manuscript accordingly.*

887: complete with "Loc. 473 similar sediments were found at…"
> ***Response:*** *We update the manuscript accordingly.*

892: "further mapped"; perhaps delete "through"?
> ***Response:*** *We update the manuscript accordingly.*

893: "located between 30 and 38 MASL"
> ***Response:*** *We update the manuscript accordingly.*

900: suggest rephrasing to "For additional context on the LIG marine, we provide below a brief overview of some individual sites to provide additional context on the LIG marine"
> ***Response:*** *As suggested, we update the manuscript to say "For additional context on the LIG marine sediment, we provide below a brief overview of some individual sites"*

902: '5-km and 1-km"
> *Response: We update the manuscript accordingly.*

903: this sentence "Sediment units could several places be traced…"needs rephrasing
> *Response: We clarify this in the revised manuscript: "Sediment units could (at several places) be traced on the surface between the outcrops"*

909: "low-angle"?
> *Response: We update the manuscript accordingly.*

912: add "between" before "35 to 40 MASL"
> *Response: We update the manuscript accordingly.*

930: same suggestion as for line 900
> *Response: As suggested, we update the manuscript to say "For additional context on the LIG marine sediment, we provide below a brief overview of some individual sites"*

932: "is situated"
> *Response: We update the manuscript accordingly.*

934: "yielded ages of 122 and 121 ka …"
> *Response: We update the manuscript accordingly.*

967: there are 7 sites, not 8
> *Response: We update the manuscript accordingly.*

970, 976, 981, 992: remove "at this site"
> *Response: We update the manuscript accordingly.*

1013: "within this unit there are class…"
> *Response: We update the manuscript accordingly.*

1021: what is meant by "and all radiocarbon ages considered minimum such."
> *Response: This was a typo. We clarify "and all radiocarbon ages considered as minimum".*

1034: remove "(Dredge et al., 1992)". I suggest moving "the U-Th ages are interpreted as minimum age constraints by the original" 2 lines above where the U-Th ages are presented
> *Response: We update the manuscript accordingly.*

1042: delete "and" after "reported"; delete "(Miller et al., 1977)" - it has been already mentioned in the same sentence that this are the authors who reported the ages
> *Response: We update the manuscript accordingly.*

1044: delete comma after "position"
> ***Response:*** *We update the manuscript accordingly.*

1046, 1049: remove "at this site"
> ***Response:*** *We update the manuscript accordingly.*

1068: use ka unit
> ***Response:*** *We clarify "3.5 kyrs"*

1109: add "and" before "the date is very close…"
> ***Response:*** *We update the manuscript accordingly.*

1117: can you explain: "and amino acid ratios 0.1 to 0.15, both supporting a LIG age assignment"?
> ***Response:*** *We clarify "based on the amino acid framework for nearby Banks Island". We also add 2 references: Vincent, 1982, 1983.*

1125: 'U-Th ages of 133 and 141 ka"
> ***Response:*** *We update the manuscript accordingly.*

1231: use "is" instead of "in"; remove "position"
> ***Response:*** *We clarify this sentence in the revised manuscript.*

1238: replace "is" with "if"
> ***Response:*** *We update the manuscript accordingly.*

1279: remove the first "an"
> ***Response:*** *We update the manuscript accordingly.*

1298: suggest rephrasing this sentence: "Radiocarbon ages ….yielded ages"
> ***Response:*** *We update the manuscript accordingly.*

**Figures and tables**
Figure 1: no need to define again LIG, it has been done so earlier in the text
> ***Response:*** *We update the manuscript accordingly.*

Figure 2: I suggest organize this figure in such way that you only mention once "nomenclature"; one option would be to color-code the 4 nomenclatures and explain in the figure legend. The measure unit on the Y axis is missing.
> ***Response:*** *We thank the Reviewer for this suggestion. We make this change to Figure 2 (along with other minor changes suggested by the other Reviewer). The measurement unit along the y axis is also clarified in the revised manuscript.*

Figure 3: I recommend replacing the small black dots with a different symbol to make it easier to differentiate them from the large ones. I think the figure is missing labels a) and b) mentioned in the legend. I suggest listing the location in the order they are presented in the figure from left to right

> *Response: We appreciate the suggestions for this figure; however, we make no changes at this time. The small 'dots' that are used to indicate the location of geochronological data are substantially larger than the other 'dots' in the image (indicating sands in the stratigraphic records). The letters "a" and "b" in the figure caption refer to two separate publications from the same authors (Astakhov and Nazarov, 2010); they are not labels for the figure. Moreover, all "dots" are labeled on the figure. Finally, we present the stratigraphic plots in a west-east manner to align with their position in the Russian Arctic mainland.*

Figure 4. MASL instead of m.a.s.l.. Add the name the y-axis (Sea level) and place the unit along the axis.

> *Response: We make these change to Figure 4 in the revised manuscript.*

Figure 5. I suggest adding the 10 m thickness of the MIS 5e sediments in the picture.

> *Response: We update Figure 5 in the revised manuscript. We also include 2 panels (original photograph and annotated photograph) to future clarify this site for the reader.*

Figure 7. ( from Alexanderson in brackets).

> *Response: We make this change to the citation style in the revised manuscript.*

Figure 8: replace "forests" with "foresets"

> *Response: Thank you AR1 for your careful review of our manuscript. We made this change to the revised version.*

Table 5. Replace "nation" with "country".

> *Response: We make this change in the revised manuscript.*

**Anonymous Referee #2**

**Overall comments**

This paper serves two important purposes – an extensive and detailed review of LIG sea level field sites and a usable database of relative sea level (RSL) and chronology datapoints for each site. This undertaking certainly required a great deal of effort and was very well designed. I think this paper should be accepted for publication after addressing a few minor comments. I will admit that I am a bit of outsider when it comes to knowing individual sites of RSL across formerly glaciated areas (especially in Eurasia), but with that in mind I found the paper to be clear, concise, and easy to follow. Below are my two comments of greatest concern.

> *Response: We thank Anonymous Referee 2 for seeing the value of this work and for providing a thorough review. We adopted many of the suggestions into the revised manuscript.*

First, the presentation of field site chronologic and elevation quality in table format will be helpful for those accessing the database for their own research use, but in the paper, it is difficult for the reader to synthesize all of this information (mostly since the database is so extensive!). What I suggest is making a series of maps that show the elevation/chronology quality scaled by color. I think this could be helpful for the paleo-sea level community as it may allow identification of field areas that need to be re-visited to improve data quality. This could also provide some spatial data that allows us to understand which geographic areas in general need better data coverage (like eastern Siberia).

> *Response: We thank the Reviewer for their thorough review of our work and their constructive feedback. Following their suggestions, we have made several updates to the presentation of data in the manuscript. As detailed below, we (1) added 6 additional regional maps, and (2) include a plot of RSL vs chronology quality with each dataset labeled. These new figures help to visualize the spatial extent and quality of the dataset. We feel it would be repetitive to include the RSL/chronology rankings on the maps because they are clearly detailed in Table 5.*

In addition to making these maps (or as an alternativ, as part of the quality assessment, I think that an additional metric or index could be designed to identify which sites have the best overall data quality. This could be a 2- dimensional plot where the x-axis is the RSL data quality, and the y-axis is the chronology data quality. This would at least identify which sites should be revisited to improve either chronology or rsl measurements (or both!).
Along the same thread of thinking, it may be worth more discussion as to why certain areas have a greater prevalence and preservation of LIG sediments than others. Why aren't there any LIG marine sites from the Cordilleran Ice Sheet? I assume it's due to some combination of erosion, the prevalence of fjord settings that have drowned many sites (even if they rebounded after the LGM), but some comment on the spatial distribution of records and identification of data gaps in the discussion would be helpful.

> *Response: We thank the Reviewer for their helpful and positive feedback. Following their suggestion, we update Figure 15 to include a comparison of the RSL/chronology ranks for each site. The new version of Figure 15 better visualizes the 'spread' of sites in the glaciated Northern Hemisphere. The Reviewer also suggests a more detailed discussion of why certain areas have greater preservation of LIG marine sites, and others do not. We agree this is a good idea, and we add a few sentences to the Discussion, especially as it relates to the relative scarcity of LIG marine sites in North America. Moreover, throughout the revised manuscript, we make note of areas that were not covered by MIS 2 glaciation (a likely reason for the preservation of LIG marine sites due to less erosion).*

My second comment is that I was surprised that cosmogenic burial dating was not included or discussed in this paper. Burial dating is increasingly common in glacial settings, and it is possible to use cosmogenic nuclide burial isochron dating to constrain an absolute age of sediment burial. If such datasets are not available or common, I think the authors should acknowledge this in the discussion to encourage its application in future studies. I thank the authors for their

hard work pulling together this extensive database. I enjoyed learning about these many field sites and I look forward to seeing the WALIS database continue to grow in the future.

*Response: We thank the Reviewer for their thorough review of our work and their constructive feedback. We agree that cosmogenic nuclide dating is common in glacial studies. However, as detailed below, we did not come across any that were dated via the cosmogenic nuclide method. Nevertheless, we are excited by the potential for this method to date LIG marine sites, and we include a citation to this paper in the "Conclusion" section. We also hope that this method will be applied in the future to LIG marine sites to add further certainties to these hard-to-date sediments.*

**Specific comments:**

**Intro**

Line 64-66: This detail about the different regional names for the LIG seems out of place. I would move it to paragraph 1 where you define the LIG.

*Response: We thank the Reviewer for this suggestion. However, this sentence on regional place names seems more out-of-place at the end of paragraph 1. We attempted to place this sentence in other areas of the introduction, but we feel it is best placed where it was originally.*

**Dating techniques**

This is a really helpful brief overview of dating methods and their limitations for interpreting ages.

*Response: We thank the Reviewer for their positive feedback.*

Line 140: I think naming interglacial deposits as "Eemian" is potentially confusing since it has a specific regional meaning. Why not call it a "LIG deposit"?

*Response: We agree this is potentially confusing, and a similar concern was raised by the first Reviewer regarding our use of "MIS 5e". Throughout the revised manuscript, we change the notation to "LIG deposit".*

Line 150-59: I would switch the order of IRSL and OSL so that readers understand what luminescence dating is first, then explain why IRSL has additional complications. I would also include in these sections the temporal limitations of OSL and IRSL. I think you also need to explain what anomalous fading means more fully, i.e., the electrons that accumulate in the crystalline lattice but leak over time, which leads to apparent IRSL ages that are too young. I would also include that fading rates can be measured in the lab and that fading corrections are specific to individual IRSL samples.

*Response: We read over this section and clarified the concerns of the reviewer "feldspar is susceptible to anomalous fading, which can lead to large uncertainties in the age estimation because the electrons trapped in the crystal lattice 'leak' over time"*

Are there any cosmogenic burial ages ($26Al/10Be$; $36Cl/10Be$) for LIG sediments? This method is increasingly common, and I was surprised that this was not included as part of the database. If such papers do not exist, I think it would be important to point this out at the end of the paper

in order to encourage the community to apply these methods to these types of sea level archives.

> *Response: In our search for LIG marine sites, we did not come across any that were dated via this method. Nevertheless, we are excited by the potential for this method to date LIG marine sites, and we include a citation to this paper in the "Conclusion" section.*

I would specifically look for any papers that include cosmogenic burial isochron dating as this approach can provide an absolute age of burial of sediments regardless of nuclide inheritance. Here is an example from North America: Balco & Rovey, 2008. An isochron method for cosmogenic nuclide dating of buried soils and sediments. American Journal of Science December 2008, 308 (10) 1083-1114; DOI: https://doi.org/10.2475/10.2008.02

> *Response: We thank the Reviewer for this interesting suggestion. In our search for LIG marine sites, we did not come across any that were dated via this method. We emphasize that the purpose of this manuscript was to compile LIG* marine *sites, and therefore application of this method to dating terrestrial LIG deposits is not appropriate for this manuscript. Nevertheless, we are excited by the potential for this method to date LIG marine sites, and we include a citation to this paper in the "Discussion" section.*

**Quality assessment**

This is a great feature of this database. The rubric for how quality is assessed makes sense. My concern with the quality assessment is that the results of this evaluation effort are not presented spatially. It would be helpful to know where the best and worst elevation and chronology constraints are located. Would it be possible to include a metric or index value that considers both the RSL and chronology data quality? Or a plot of RSL vs chronology quality with each dataset labeled. This could be used to identify which sites have the best data overall.

> *Response: We thank the Reviewer for their positive feedback on the quality assessment, which had to be developed differently than most other papers in this Special Issue owing to the complexities associated with working on stratigraphic records in the glaciated region.*
>
> *The Reviewer recommends presenting the data quality assessment metrics spatially, so that the reader can know where the best/worse elevation and chronology data are located. This is a great suggestion; however, we feel it would be repetitive to include the RSL/chronology rankings on the maps because they are clearly detailed in Table 5. As a compromise (and to better illustrate the spatial extent of the sites), we add 6 additional regional maps to help the reader.*
>
> *The Reviewer recommends including a metric or index value that considers both the RSL and chronology data quality. In the revised manuscript, we update Figure 9 (now Figure 15) to include a plot of RSL vs chronology quality with each dataset labeled. This will help the reader to identify which sites have the best data overall.*

Line 196: Why aren't the total number of age constraints from an individual site considered as part of the quality assessment? Field sites with only a few ages don't have as robust of chronologies as those with many ages, right?

>*Response: In the revised manuscript, we clarify "Sites that have multiple age determinations increase the confidence in our LIG age assignment, though the precision of the techniques is not high enough to elevate the rating to the highest level."*

**Relative Sea Level Proxies & Other marine sites**

I did not comment on specific field sites because I am unfamiliar with much of this literature, but I will note that this is really helpful. I was able to learn the most salient information about many different field sites. Each section is concise and clear, and the dataset appears complete.

>*Response: We thank the Reviewer for their positive feedback and for seeing the value of our work.*

**Discussion**

Line 1228-29: Is this the only reason there are so few marine sites from North America? Are there places we could expect to find such records but have not identified them yet? Do borehole records include monitoring well and drinking water well logs? I am not suggesting that you look for all of these for this specific paper, but I was really surprised to see such a dearth of records from North America, particularly in the St. Lawrence River Valley.

>*Response: We agree that it was surprising to see so few records in North America. In the revised manuscript, we add a sentence to this effect in the "Conclusions and Future Research" section. We state:*

>*"Future work to locate new LIG sites (particularly in North America, which has a relative scarcity of LIG sites) should be focused on coastal regions that are known to contain extensive stratigraphic records (e.g. the Saint Lawrence valley)."*

**Conclusions**

In this section I would also include cosmogenic nuclides as an additional chronologic method that should be pursued for improving age control of these sediments.

>*Response: In the revised manuscript, we clarify:*

>*"key areas of future research should focus on revisiting these sites to vet the stratigraphic record and testing new geochronological methods (especially U-Th, OSL, and the potential for cosmogenic nuclide methods to constrain the age of buried sediments)."*

**Figures**

Figure 1: This is a helpful map, which shows how much work the authors have done to compile these records. It also shows the very large spatial gaps that still exist! However, I think you need to have additional accompanying maps, perhaps by region, that show your RSL elevation and

chronology data and quality ratings for all of these sites. It would be helpful at the community level to visually identify which field sites need better constraints for future work.

> *Response: This is an excellent suggestion. In the revised manuscript, we add 6 additional regional maps, and include a plot of RSL vs chronology quality with each dataset labeled. These new figures help to visualize the spatial extent and quality of the dataset. We feel the maps are cluttered with data; thus, we do not include the RSL/chronology rankings on the maps. Instead, they are clearly detailed in Table 5.*

Figure 3: The age constraints for OSL should have uncertainties reported as well. How were the mean ages reported at the bottom of the figure calculated?

> *Response: This figure provides an overview of the stratigraphy and geochronology at MIS 5e sites in the Russian Arctic mainland. Further details for each site is detailed in the Supplemental Table, which is available Open Access in Zenodo. We clarify in the caption:*
>
> *"Comprehensive documentation of each site (including detailed sea level measurements and chronological data) is available at https://doi.org/10.5281/zenodo.5602212 (Dalton et al., 2021)"*

**Tables**

Table 4: I can understand why all of the individual ages for each site are not reported, but could the total n of age constraints for each site be reported? Could a range of ages be reported in this table as well?

> *Response: In most cases were the age of LIG marine sediments is averages, we provide an 'n' number. Following the overall format of other WALIS manuscripts, the purpose of this table is to summarize the sea-level data and to provide a brief overview of the types of geochronological data at each site. Further details for each site is detailed in the Supplemental Table, which is available Open Access in Zenodo. We clarify in the caption for Table 5:*
>
> *"Comprehensive documentation of each site (including detailed sea level measurements and chronological data) is available at https://doi.org/10.5281/zenodo.5602212 (Dalton et al., 2021)"*

**Technical Comments**

Line 25: typo. "…regression due to sea level…"

> *Response: We made this change to the revised manuscript.*

Line 42: not sure about ESSD, but typically in prep manuscripts cannot be cited

> *Response: We knew about this 'in prep' manuscript because it was part of the same Special Issue. It has since been submitted, and we update the citation to reflect "in review".*

Line 55: grammar: "The first part of this manuscript (Sections 2-5) we defines…"

> **Response:** *We made this change to the revised manuscript.*

Line 143-44: grammar, "…although in recent years it has been largely replaced…"
> **Response:** *We made this change to the revised manuscript.*

Fig 2: Although it may be obvious to the paleoclimate community I would explain and slightly change the color scheme. The interglacial stages should be red with colder interglacial substages (i.e. MIS 5d, 5b) in a lighter hue of red since they were still warmer than the MIS 3 interstadial. Likewise, MIS 4 and MIS2 should be a darker blue and MIS 3 should be a lighter blue since it was a mild interstadial. Also include substage next to Marine Isotope Stages on upper right label.
> **Response:** *We make all suggested changes to Figure 2 in the revised manuscript.*

Figure 4: Is this figure from another paper? If so, cite the source.
> **Response**: *In the revised manuscript, we clarify "Modified from Mangerud et al., (1981)."*

Figure 5: These labels are slightly hard to see. Units in the figure are reported differently (m.a.s.l.) than in the paper (MASL). I would move the white arrow to the outside of the labels and also include some more annotations that help the audience see the strata better.
> **Response:** *Thank you for this suggestion. We update this figure following the suggestion by the reviewer in the revised manuscript. We also include 2 panels (original photograph and and annotated photograph) to future clarify this site for the reader.*

Figure 6: the scale is washed out. Reduce the brightness and/or highlights of this photo so that it is more visible.
> **Response:** *We reduce the brightness and sharpen this figure in the revised manuscript.*

Figure 8. Typo: forests is a typo. Change to foresets. An extremely coarse forest would indeed be a sight to see!
> **Response:** *Thank you AR2 for your careful review of our manuscript. We made this change to the revised version. This concern was also raised by the other Reviewer.*